# Cooperative insulation of regulatory domains by CTCF-dependent physical insulation and promoter competition

Thais Ealo[1,4], Victor Sanchez-Gaya [1,4] ✉, Patricia Respuela[1], María Muñoz-San Martín [1,2], Elva Martin-Batista[3], Endika Haro [1] ✉ & Alvaro Rada-Iglesias [1] ✉

The specificity of gene expression during development requires the insulation of regulatory domains to avoid inappropriate enhancer-gene interactions. In vertebrates, this insulator function is mostly attributed to clusters of CTCF sites located at topologically associating domain (TAD) boundaries. However, TAD boundaries allow some physical crosstalk across regulatory domains, which is at odds with the specific and precise expression of developmental genes. Here we show that developmental genes and nearby clusters of CTCF sites cooperatively foster the robust insulation of regulatory domains. By genetically dissecting a couple of representative loci in mouse embryonic stem cells, we show that CTCF sites prevent undesirable enhancer-gene contacts (*i.e.* physical insulation), while developmental genes preferentially contribute to regulatory insulation through non-structural mechanisms involving promoter competition rather than enhancer blocking. Overall, our work provides important insights into the insulation of regulatory domains, which in turn might help interpreting the pathological consequences of certain structural variants.

The specific and precise expression of developmental genes during embryogenesis requires the concerted action of several types of *cis*-regulatory elements[1,2]. Among them, enhancers are able to activate gene transcription by communicating with gene promoters across large linear distances[1,3]. In contrast, insulators protect gene promoters from signals emanating from neighboring regulatory domains by either blocking the communication with non-cognate enhancers (i.e. enhancer blocking insulators) or by acting as barriers against the spreading of repressive chromatin (i.e. boundary elements)[4,5]. In order to execute their enhancer-blocking function, insulators need to be located between an enhancer and the protected promoter. Thus, enhancer activity is typically constrained within discrete regulatory domains containing cognate gene promoters and demarcated by

insulators at both ends[4,6,7]. Insulators are best understood in *Drosophila*, where the combinatorial binding of several architectural proteins (i.e. Cp190, CTCF, Su(Hw), BEAF-32, GAF) to these elements enables their enhancer-blocking activity[8–10]. In mammals, the repertoire of architectural proteins is markedly reduced and CTCF is considered the main insulator-binding protein[6,11,12].

More recently, the emergence of novel methodologies, such as Hi-C, has resolved the 3D organization of genomes at high resolution. These studies revealed that, in mammals, CTCF sites often coincide with the boundaries of large self-interacting genomic regions that were termed topologically associating domains (TADs)[13]. Notably, TADs often overlap with regulatory domains, particularly those containing major developmental genes whose expression is regulated by

[1]Institute of Biomedicine and Biotechnology of Cantabria (IBBTEC), CSIC/Universidad de Cantabria, Santander, Spain. [2]Service of Neurology, University Hospital Marqués de Valdecilla, Universidad de Cantabria and IDIVAL, Santander, Spain. [3]Centro de Biología Molecular Severo Ochoa (CBMSO), CSIC-UAM, Madrid, Spain. [4]These authors contributed equally: Thais Ealo, Victor Sanchez-Gaya. ✉e-mail: victor.sanchezgaya@unican.es; endika.haro@unican.es; alvaro.rada@unican.es

long-range enhancers[14]. Furthermore, these developmental genes are often located within TADs whose boundaries display particularly strong evolutionary conservation[14]. In mammals, the formation of TAD boundaries involves the stalling of loop-extruding cohesin complexes by CTCF[11,15,16]. In contrast, cohesin and loop extrusion might not play a major role in TAD boundary formation in *Drosophila*, which might instead involve pairwise looping between insulator elements due to the dimerization of architectural proteins[10,17,18]. Furthermore, although architectural proteins play a preponderant role in TAD boundary formation, with, for example, over 80% of mammalian TAD boundaries being CTCF dependent, they seem to be dispensable for or even counteract the organization of chromatin compartments[11,15,16]. Interestingly, TAD boundaries that are not dependent on CTCF are often located proximal to active transcription start sites (TSS) and housekeeping genes[13,19]. Recent reports indicate that RNA Pol2 can participate in the formation of contact domains and physical boundaries[20–26], but whether this can globally contribute to the insulation of regulatory domains and enhancer-blocking remains controversial[27–29]. In fact, due to the intermingling of chromatin domains, it has been shown that, at least at certain loci, the location of genes near or within boundaries might facilitate, rather than block, the communication with enhancers located along neighbouring TADs[30–32]. Similarly, strong enhancers might be able to bypass boundaries and activate genes across TADs[33,34]. These observations suggest that, rather than impenetrable walls, TAD boundaries might act as dynamic and partially permeable barriers, allowing certain level of physical crosstalk across regulatory domains[35,36]. On the other hand, in mammals, the CTCF clusters that often overlap with TAD boundaries might not only act as insulators but also as tethering elements that facilitate the physical communication of nearby genes with distal enhancers located within the same TAD[37–41]. Overall, TAD boundaries seem to play a dual regulatory role, as they can facilitate enhancer-gene communication within TADs while preventing, albeit partly, undesired enhancer-gene contacts across TADs[42].

The permeability of TAD boundaries is at odds with the remarkable tissue specificity and spatiotemporal precision with which most developmental genes are expressed, even in the absence of CTCF[11,18] or other architectural proteins[8,10,15]. Therefore, besides boundary elements, additional mechanisms might ensure the robust insulation of developmental regulatory domains. In this regard, work mostly based on the genetic dissection of the mammalian alpha and beta globin loci, as well as reporter assays in *Drosophila*, suggests that promoters can also regulate enhancer-gene communication through either promoter competition or enhancer blocking[24,43–50]. When located within the same domain, several promoters can, in principle, share and get activated by a common enhancer[51–53]. However, depending on the presence of distinct promoter elements, the activation of a preferred gene can preclude the expression of its neighbours independently of insulators. Promoter competition occurs when the preferred gene gets specifically activated regardless of its relative position with respect to the shared enhancer/s and neighbouring gene/s[47,54]. It has been previously suggested that the competition between promoters for a shared enhancer might involve mutually exclusive enhancer-promoter contacts (flip-flop model)[44,55]. In addition, recent observations suggest that, at certain loci, promoter competition could also entail non-structural mechanisms whereby promoters and enhancers share transcriptional hubs/condensates[51,52], within which promoters might compete for rate-limiting factors (e.g. TFs, GTFs, RNA Pol2) required for gene transcription[56,57]. In contrast, promoter-driven enhancer blocking occurs when the preferred gene prevents the expression of its neighbours only if placed between the shared enhancer/s and the other gene/s, thus resembling how insulators work[24,48]. Recent studies suggest that promoter-driven enhancer blocking might involve structural mechanisms, whereby protein complexes present at promoters (e.g. RNA Pol2) act as weak barriers against cohesin-mediated loop

extrusion[21–23,25]. However, other studies based on the depletion or inhibition of RNA Pol2 have reported small or no effects in 3D chromatin architecture and formation of TAD boundaries[27–29]. Overall, it is currently unclear how prevalent these two promoter-dependent mechanisms are within endogenous loci and whether they significantly contribute to gene expression specificity, particularly in mammals. Similarly, it is largely unknown whether insulators, architectural proteins and promoters might somehow crosstalk in order to modulate the insulation of regulatory domains[26,49].

In the present study, we uncovered that a significant fraction of developmental genes in both mice and humans are located near TAD boundaries that contain clusters of CTCF binding sites. Furthermore, at these boundaries, developmental genes and CTCF sites are often organized in a sequential and evolutionary conserved manner, with the genes preceding the CTCF clusters. In contrast, genes previously reported as capable of bypassing boundaries[30–32] are typically flanked by CTCF sites and, thus, located within CTFC clusters. Most importantly, through the exhaustive genetic dissection of a couple of representative developmental loci (i.e. *Gbx2/Asb18*, *Six3/Six2*), we show that the positioning of developmental genes close to boundaries does not facilitate their own expression. Instead, we found that developmental genes and CTCF sites strengthen the regulatory insulation capacity of the nearby TAD boundaries. Finally, we show that developmental genes seem to preferentially contribute to regulatory insulation through non-structural mechanisms that involve promoter competition rather than enhancer blocking. Overall, our work provides important insights into the mechanisms contributing to the robust and specific expression of developmental genes during mammalian embryogenesis.

## Results
### TADs containing developmental genes display distinctive features
While dissecting the regulatory landscapes of a few representative developmental genes characterized by the presence of large CpG island (CGI) clusters and Polycomb-Group (PcG) protein domains at their promoters (e.g. *TFAPA*, *Six3*, *Lhx5*)[58,59], we noticed that these genes were often located within TADs displaying rather unique features. Namely, these developmental TADs (i) showed low gene density and (ii) displayed a skewed gene distribution, as the developmental genes were often located near TAD boundaries (Figs. S1–2). To evaluate whether these features are prevalent among developmental TADs, we used TAD maps previously generated in either mouse or human cells and defined developmental genes based on the presence of broad PcG domains around their TSS (See Methods)[60–62]. This definition does not include all developmental genes according to Gene Ontology (GO) terms, but rather selects a subset of major developmental genes whose promoters include large CGI clusters and display strong enhancer responsiveness[63–65]. Nevertheless, for the sake of simplicity, and unless stated otherwise, we will refer to this subset of genes with broad PcG domains around their TSS as developmental genes. Using this definition of developmental genes, the following analyses were performed:

(i) Gene density: We first classified TADs previously identified in mESC and hESC according to gene density (High, Medium or Low) and performed Gene Ontology analyses for the genes contained within each TAD category. Interestingly, TADs with low gene density were strongly and specifically enriched in developmental genes, while High and Medium gene density TADs showed milder gene ontology enrichments and were not particularly enriched in developmental terms (Fig. 1A, Fig. S3, Data S1–2). Next, to evaluate whether these observations could be generalized, we considered TAD maps previously generated in additional human (n = 37) and mouse (n = 11) cell types. Notably, developmental genes (i.e. genes with broad PcG domains) were significantly enriched within TADs with low gene

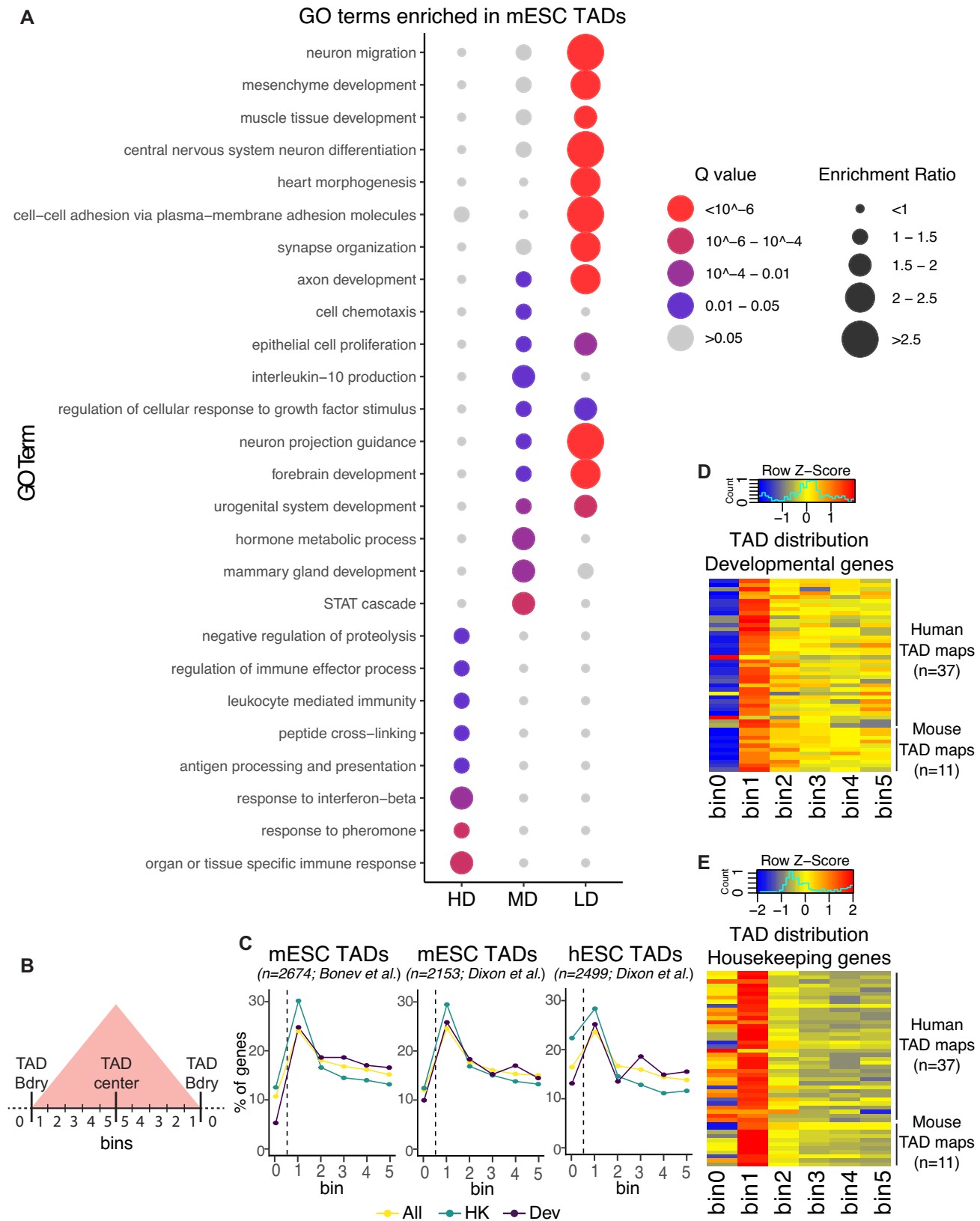

**A** GO terms enriched in mESC TADs

**B**

**C** mESC TADs *(n=2674; Bonev et al.)* · mESC TADs *(n=2153; Dixon et al.)* · hESC TADs *(n=2499; Dixon et al.)*

**D** TAD distribution Developmental genes

**E** TAD distribution Housekeeping genes

density in all the considered mouse and human cell types (Fisher test p-values obtained were smaller than 4.2e-05 and 1.6e-10 in humans and mice, respectively, for all tests performed, with an average odds ratio of 2.0 (95% confidence interval: 1.9 to 2.1) in humans and 2.1 in mice (95% confidence interval: 2.0 to 2.2). In contrast, housekeeping genes were not enriched within low gene density TADs in any of the analyzed

TAD maps (Fisher test p-value = 1 for all tests, average odds ratio of 0.36 and 0.65 in humans and mice, respectively).

(ii) Gene distribution: It was previously shown that TAD boundaries are enriched for housekeeping genes in comparison to tissue-specific genes[66]. However, from a regulatory standpoint, developmental genes (as defined here based on the presence of broad PcG

**Fig. 1 | TADs containing developmental genes have distinctive features. A** TADs previously identified in mESC[13] were classified based on their gene density in three different groups: High Density (HD), Medium Density (MD) and Low Density (LD). Then, the genes present within each TAD group were subject to GO enrichment analysis. For the MD (n = 17 enriched GO terms) and LD (n = 135 enriched GO terms) groups, only the top 10 most significantly enriched GO terms are highlighted (i.e. Q value ≤ 0.05), while for the HD group all the significantly enriched GO terms (n = 8) are presented. **B** The distribution of different groups of genes within TADs was investigated using previously generated TAD maps. TADs were divided in 10 bins of equal sizes, which were then grouped in five bin pairs based on their distance to the nearest boundary (e.g. Bin 1 is the closest to TAD boundaries; Bin 5 is the most distal from TAD boundaries) (see Methods). Bin 0 indicates genes located at inter-TAD regions (i.e. within TAD boundaries). **C** Distribution of different groups of genes (All (yellow; n = 21000 for mouse and 19288 for human; Housekeeping (HK; green; n = 3936 for mouse and 2779 for human); Developmental (Dev; purple; n = 962 for mouse and 1045 for human) within TADs according to Hi-C data previously generated in mESC[13,19] and hESC[118]. Heatmap plot (with scaling by rows) showing the distribution of developmental (**D**) and housekeeping (**E**) genes within TADs previously identified in several human (n = 37 TAD maps) and mouse (n = 11 TAD maps) cell types.

domains around their TSS) and tissue-specific genes represent fundamentally different gene categories, as they typically differ in the type of promoter (i.e. CpG-rich for developmental genes and CpG-poor for tissue-specific genes) and long-range enhancer responsiveness[65,67]. To investigate the distribution of different types of genes (i.e. housekeeping, developmental, all) with respect to TADs, we divided TADs in 10 bins of equal sizes and calculated the number of genes within each bin based on the location of their TSS (Fig. 1B). In addition, genes located outside TADs (i.e. within TAD boundaries or inter-TAD genomic regions) were assigned to a bin labelled as bin 0 (Fig. 1B). Next, we computed the percentage of genes located in each bin. Overall, both housekeeping and developmental genes were preferentially enriched near TAD boundaries (i.e. bin 1), with housekeeping genes showing slightly higher percentage values for bin 1. In addition, housekeeping genes were often found not only close to but also within TAD boundaries (i.e. enriched in bin 1 and, depending on the cell type, also in bin 0), while developmental genes were preferentially located inside TADs and near their boundaries (i.e. enriched in bin 1 and depleted in bin 0) (Fig. 1C–E). Moreover, bin 1 housekeeping genes were located closer to TAD boundaries than their developmental counterparts (Fig. S4), which, considering the resolution of Hi-C data, is also in agreement with the more frequent location of housekeeping genes within TAD boundaries (i.e. bin 0). The preferential location of developmental genes close to TAD boundaries was similarly observed in both mice and humans, suggesting that it might represent an evolutionary conserved feature (Fig. 1C–E).

## Sequential organization of developmental genes and clusters of CTCF sites at developmental boundaries

Based on the previous observations, we then evaluated whether the boundaries close to bin 1 developmental genes (i.e. developmental boundaries) display any distinctive features. First, we compared the number of CTCF peaks and CTCF ChIP-seq aggregated signals at developmental and non-developmental (i.e. other) TAD boundaries (developmental boundaries are defined as those having a developmental gene located in bin 1; Fig. 1B). For both CTCF metrics, the developmental boundaries showed higher values (Fig. 2A-B). Interestingly, despite these differences in CTCF binding, insulation scores and boundary strength (i.e. physical insulation) at developmental boundaries were similar to those observed at other TAD boundaries (Fig. 2A, B). Furthermore, similar analyses were performed by considering a broader set of developmental genes (i.e. genes included in the Gene Ontology (GO) term developmental process; GO:0032502), which were further divided in two groups depending on whether their promoter regions were covered or not by broad PcG domains (Fig. S5). These analyses showed that the insulation scores and boundary strength of the boundaries associated with developmental genes were similar regardless of whether their promoters were associated with broad PcG domains, suggesting that PcG domains do not have a major effect on the physical insulation properties of nearby TAD boundaries. Next, given previous reports indicating that paused RNA Pol2 at gene promoters can create boundaries[21,22,25,68], we used PRO-seq data from mESC[69] and GRO-seq data from hiPSC[70] to calculate the RNA Pol2

pausing index (PI) for different gene categories (i.e. housekeeping, developmental, all) depending on their transcriptional status and proximity to TAD boundaries (i.e. bin1 genes) (Fig. S6). For each gene category, the PI were quite similar regardless of whether the genes were close (i.e. bin1 genes) or not to TAD boundaries (Fig. S6). Furthermore, in contrast to previous reports in *Drosophila*[68,71], but in agreement with previous observations in mammalian cells[72], developmental genes showed generally lower PI than either housekeeping or all genes regardless of their transcriptional status or proximity to TAD boundaries.

On the other hand, upon visual inspection of several developmental boundaries we noticed that developmental genes and CTCF clusters were sequentially organized (Figs. S1–2), with the developmental genes typically preceding most of the CTCF peaks (i.e. the CTCF peaks were preferentially located in the genomic regions separating the genes from the nearby TAD boundaries). To evaluate whether this sequential organization of developmental genes and CTCF clusters is somehow characteristic of developmental boundaries, we performed a global analysis of the distribution of CTCF peaks around genes located in bin 1 (Fig. 1B). Briefly, for each gene, we considered a window of +/−100 Kb around its TSS and compared the number of CTCF peaks (and associated CTCF ChIP-seq signals) located between the gene TSS and the nearby TAD boundary (i.e. outer window) with the number of peaks (and associated CTCF ChIP-seq signals) located between the gene TSS and the center of its TAD (i.e. inner window) (ΔCTCFpeaks@Bdry and ΔCTCFsignal@Bdry; Fig. 2C). Notably, the fraction of boundary-proximal genes in which the CTCF peaks and their associated signals were higher towards the TAD boundaries than towards the TAD centre (i.e. genes displaying negative ΔCTCFpeaks@Bdry and ΔCTCFsignal@Bdry values) was significantly larger for developmental genes than for other gene types (i.e. all, housekeeping) in both mice and humans (Fig. 2D, E; Fig. S7). This indicates that boundary-proximal genes in general and housekeeping genes in particular are frequently flanked by CTCF peaks, while for developmental genes the nearby CTCF peaks tend to accumulate in the genomic regions extending from the genes towards the TAD boundaries (i.e. developmental genes preceding the CTCF peaks). Next, using the same window of +/- 100 Kb around the TSS of bin 1 genes (i.e. inner and outer windows), we calculated the orientation of the CTCF sites relative to the TAD centers, distinguishing between CTCF sites oriented either towards (i.e. inward site) or away (i.e. outward site) from the TAD center (Fig. S8). A considerable fraction of CTCF peaks (at least 35%) was observed for both orientations regardless of the type of genes or windows analysed, supporting the potential relevance of not only inward but also outward CTCF sites for the proper establishment of intra-TAD chromatin interactions[73]. The differences in the fraction of inward and outward CTCF sites were negligible when comparing the inner and outer windows associated to developmental genes (Fig. S8). However, although still minor, non-negligible differences were observed for All and Housekeeping genes (Fig. S8), with the CTCF sites in the inner windows preferentially showing an inward orientation (~60%) and the CTCF sites in the outer window preferentially showing an outward orientation (>50%) (Fig. S8). Together with the results

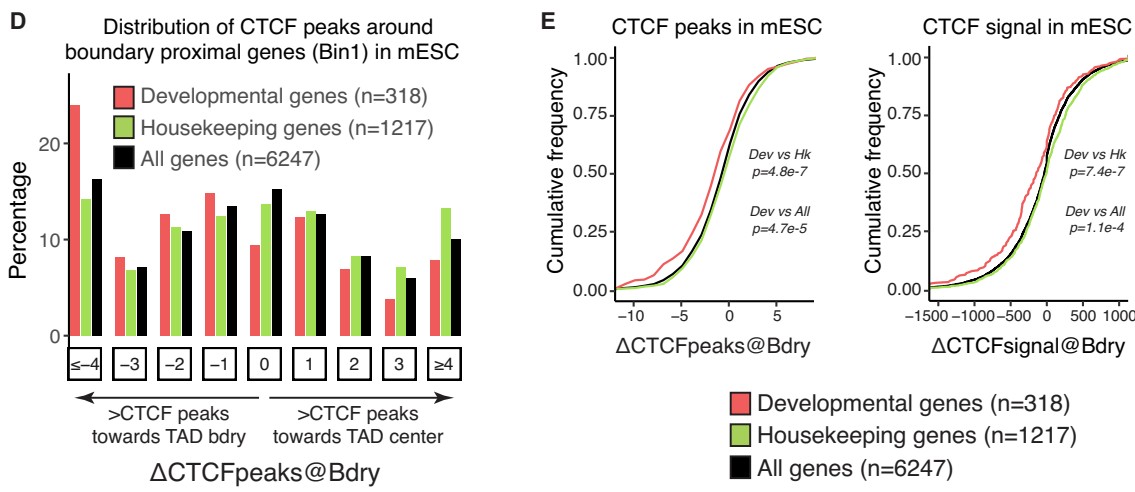

**C** Distribution of CTCF peaks & signals around genes located close to TAD boundaries (Bin1)

**ΔCTCFpeaks@Bdry=**

*#CTCF peaks towards TAD Center - #CTCF peaks towards TAD Bdry*

**ΔCTCFsignal@Bdry=**

*∑CTCF signals towards TAD Center - ∑CTCF signals towards TAD Bdry*

**D** Distribution of CTCF peaks around boundary proximal genes (Bin1) in mESC

**E** CTCF peaks in mESC    CTCF signal in mESC

presented in Fig. 1D-E and Fig. S4, these results further suggest that housekeeping genes are often embedded within TAD boundaries where they are flanked by CTCF sites with divergent orientations[74]. In contrast, developmental genes tend to be located inside TADs but close to boundaries with large clusters of CTCF sites with complex motif orientations.

In summary, the sequential organization of developmental genes and CTCF clusters at developmental boundaries seems to represent a frequent and evolutionary conserved feature of mammalian genomes. To address the potential functional relevance of this sequential organization, we genetically dissected a couple of representative loci, *Gbx2* and *Six3/Six2* (Figs. S1–2, Fig. S9). These two loci were selected because

**Fig. 2 | Developmental genes and clusters of CTCF sites are sequentially organized near mouse TAD boundaries.** TAD maps, Hi-C data and CTCF ChIP-seq profiles previously generated in mESC (**A**) or hESC (**B**) were used to investigate the insulation scores, boundary strength, number of CTCF peaks and CTCF peaks aggregated signal at developmental TAD boundaries, all other TAD boundaries or random regions. Developmental boundaries were defined as those having a nearby developmental gene located within bin 1 (Fig. 1B). P-values were calculated using unpaired two-sided Wilcoxon tests with false discovery rate correction for multiple testing; Cliff's delta (Cd) effect sizes are shown as coloured numbers (green: large effect size; blue: medium effect size; orange: small effect size; red: negligible effect size). In (A-B) box plots, the upper and lower parts of the box are the upper and lower quartiles, respectively, the horizontal line that splits the box in two is the median and the upper and lower whiskers indicate the maximum and minimum, respectively. **C** To investigate the distribution of CTCF peaks around different types of genes located close to TAD boundaries (Bin1 genes), we considered the TSS of each gene as a reference point and a window of ±100 Kb to calculate: ΔCTCFpeaks@Bdry as the difference between the number of CTCF peaks located towards the TAD center and those located towards the TAD boundary (negative values: CTCF peaks more abundant towards the TAD boundary; positive values: CTCF peaks more abundant towards the TAD center); ΔCTCFsignal@Bdry as the difference between the aggregated signal of the CTCF peaks located towards the TAD center and the aggregated signal of the CTCF peaks located towards the TAD boundary (negative values: CTCF signals higher towards the TAD boundary; positive values: CTCF signals higher towards the TAD center). **D** Histogram showing the distribution of ΔCTCFpeaks@Bdry values in mESC for different types of genes (developmental, housekeeping, all) located close to TAD boundaries (bin 1 in Fig. 1B). **E** Cumulative distribution plots for ΔCTCFpeaks@Bdry (left) and ΔCTCFsignal@Bdry (right) values in mESC. P-values were calculated using unpaired two-sided Wilcoxon tests with false discovery rate correction for multiple testing.

they display the following features: (i) they contain developmental genes (i.e. *Gbx2*, *Six3* and *Six2*) with broad PcG domains and large CGI clusters around their promoter regions; (ii) the developmental genes are located within gene-poor TADs and close to a TAD boundary (i.e. bin 1 genes); (iii) the developmental genes precede clusters of strong CTCF sites; (iv) the sequential organization of developmental genes and CTCF clusters is evolutionary conserved (Fig. S9).

### The positioning of *Gbx2* near a TAD boundary does not facilitate the maintenance of its own expression in ESC

We first focused on *Gbx2*, a major developmental gene with important functions in different processes such as neural patterning or naïve pluripotency[75,76]. *Gbx2* is highly expressed in mESC (12.15 FPKM[58]), presumably due to the regulatory activity of a super-enhancer (SE) located approximately 60 Kb downstream of its TSS (Fig. 3A). Importantly, *Gbx2* and its SE are found within a gene-poor TAD, with *Gbx2* being located ~10 Kb upstream from three CTCF sites that constitute the 3' TAD boundary (Fig. 3A). Moreover, this CTCF cluster separates *Gbx2* from *Asb18*, a tissue specific gene located within the neighbouring TAD (~60 Kb downstream of the CTCF cluster) and that is inactive in mESC (0.056 FPKM[58]) (Fig. 3A). To start evaluating whether the sequential positioning of *Gbx2* and the CTCF cluster at the 3' boundary is of any functional relevance, we first generated multiple mESC clonal lines homozygous for the following re-arrangements: (i) *Δ3xCTCF* - a 10 Kb deletion that eliminates the three CTCF sites, which in principle should lead to the fusion of the *Gbx2* and *Asb18* TADs; (ii) *71Kb INV* - a 71 kb inversion that re-positions *Gbx2* and the SE with respect to the CTCF cluster, placing the enhancer in between *Gbx2* and *Asb18*; (iii) *Δ3xCTCF:71Kb INV* - both the 71 Kb inversion and the 10 Kb deletion (Fig. 3B; Fig. S10). Next, we measured *Gbx2* and *Asb18* expression in all the generated mESC lines (Fig. 3C-D). Regarding *Gbx2*, we found that neither the 71Kb inversion nor the CTCF cluster deletion significantly affected its expression (Fig. 3C). The *Gbx2* SE is located close to a CTCF site that is not included in the 71 Kb inversion (Fig. 3A, B) and that could theoretically contribute to the maintenance of proper *Gbx2* expression in *Δ3xCTCF* and *71Kb INV* cells. To directly address this possibility, we generated ESC lines with a small (256 bp) deletion that eliminates the CTCF site located upstream of the *Gbx2* SE (*ΔCTCF Gbx2SE*) (Fig. S11A–C). The deletion of this CTCF site lead to a minor and non-significant decrease in *Gbx2* expression levels (Fig. S11D). Overall, these results suggests that, in contrast to previous reports[31,32,38], neither the positioning of *Gbx2* close to the 3' boundary nor the CTCF sites play a major role in the maintenance of *Gbx2* expression.

On the other hand, both the 71Kb inversion and the Δ3xCTCF deletion significantly increased *Asb18* expression, with the inversion having a larger effect (Fig. 3D). Moreover, when both re-arrangements were combined, *Asb18* expression levels were increased to a similar extent as with the inversion alone (Fig. 3D). Considering the established function of tandem CTCF sites as insulators[77,78], the effects of the 10 Kb deletion on *Asb18* expression were somehow expected. However, the results obtained for the inversion were more surprising and suggest that, when positioned close to the 3' boundary, *Gbx2* might display enhancer blocking activity and, thus, contribute to the physical insulation of its own regulatory domain. Interestingly, the *Gbx2* gene is transcribed away from the boundary and towards the SE (Fig. 3A, B), which could potentially counteract cohesin-mediated loop extrusion[21–23] and, thus, contribute to enhancer blocking. To asses whether *Gbx2* orientation contributes to the insulation of its own regulatory domain, we generated ESC lines in which we inverted the *Gbx2* gene in the presence (*Gbx2 INV*) or absence (*Gbx2 INV:Δ3xCTCF*) of the 3XCTCF cluster (Fig. S12A–C). Analyses of the resulting ESC lines indicate that the orientation of *Gbx2* does not significantly contribute to regulatory insulation, as the expression of *Asb18* in *Gbx2 INV* and *Gbx2 INV: Δ3xCTCF* cells was rather similar to the one observed in WT and *Δ3xCTCF* cells, respectively (Fig. S12D). To more directly assess whether *Gbx2* could contribute to physical insulation through enhancer blocking, we then performed Capture-C experiments using the *Gbx2* SE or the *Asb18* promoter as viewpoints in WT, *Δ3xCTCF, 71Kb INV and Δ3xCTCF:71Kb INV* ESC. As expected, the deletion of the 3XCTCF cluster reduced the physical insulation between the *Gbx2* and *Asb18* TADs, thus increasing the contact frequency between *Asb18* and the SE (Fig. 3E, F; Fig. S13C, D) as well as between *Asb18* and *Gbx2* (Fig. S13C, E). In contrast, the 71 Kb inversion strongly increased the contact frequency between the SE and the 3XCTCF cluster (Fig. 3E, Fig. S13B), but had a minor impact on the *Asb18*-SE or *Asb18*-*Gbx2* contacts (Fig. 3E, F, Fig. S13C–E). Furthermore, both the 3XCTCF deletion and the 71 Kb inversion reduced the contact frequency between the SE and *Gbx2* (Fig. 3E, Fig. S13A), which, nevertheless, did not have major regulatory effects on *Gbx2* expression (Fig. 3C). Overall, these Capture-C experiments suggest that the increased expression of *Asb18* in cells with the 71 Kb inversion is unlikely to be caused by the loss of *Gbx2* enhancer blocking activity. One alternative explanation is that the inversion reduces the linear distance between *Asb18* and the SE (from 141 Kb to 83 Kb), which in turn might increase *Asb18* expression without significantly affecting enhancer-gene contacts[79–81]. In agreement with this non-linear relationship between gene activity and E-P contact frequency, the combination of the 71 Kb inversion and the CTCF deletion resulted in a strong loss of physical insulation between the *Gbx2* and *Asb18* TADs that was not translated into a further increase in *Asb18* expression compared to cells with the inversion alone (Fig. 3D–F, Fig. S13). Furthermore, since *Asb18* is expressed at low levels in mESC (0.056 FPKM), the upregulation of *Asb18* in absolute levels in Δ3XCTCF (~5-fold) or 71Kb INV (~16-fold) cells is still very small, suggesting that the responsiveness between the *Asb18* promoter and the *Gbx2* enhancer might be limited[67].

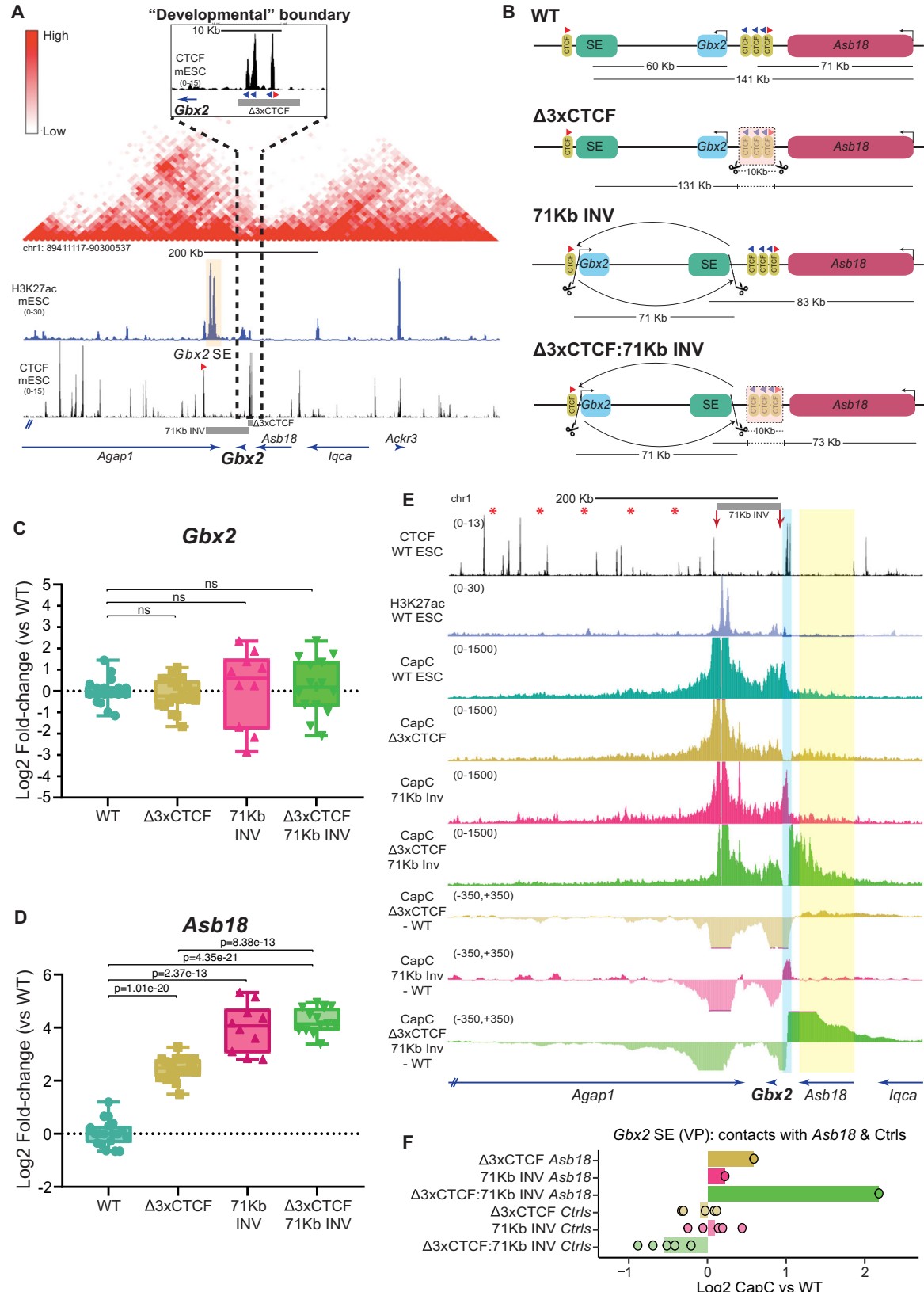

## Robust regulatory insulation between the *Gbx2* and *Asb18* TADs depends on the cooperative effects of *Gbx2* and nearby CTCF sites

The previous results indicate that the physical insulation between the *Gbx2* and *Asb18* TADs can be largely attributed to the cluster of CTCF sites. Interestingly, despite increasing *Asb18*-SE contact frequency,

the combination of the CTCF deletion and the 71 Kb inversion did not significantly change *Asb18* expression compared to the cells with the 71 Kb inversion alone (Fig. 3D–F; Fig. S13). Since promoter competition can occur regardless of the relative position of the shared enhancer(s) with respect to the competing promoters[47,48], this made us wonder whether, in the absence of the CTCF sites, the presence of

**Fig. 3 | The positioning of *Gbx2* close to its 3′ TAD boundary is not required to sustain its expression in mESC. A** mESC Hi-C data[19] shows that *Gbx2* and *Asb18* are located within neighbouring TADs separated by a cluster of three CTCF sites[122]. The orientation of key CTCF sites is illustrated with red (sense) and blue (antisense) triangles. **B** Graphical overview of genomic rearrangements generated in mESC within the *Gbx2/Asb18* locus: Δ3XCTCF; 71KbINV and Δ3XCTCF:71KbINV. **C, D** The expression of *Gbx2* and *Asb18* was measured by RT−qPCR in ESC that were either WT or homozygous for the genomic re-arrangements described in (**B**). *Gbx2* and *Asb18* expression was measured in the following number of biological replicates: WT −18 replicates; Δ3XCTCF −21 replicates using two different clonal lines; 71Kb INV −10 replicates using two different clonal lines; Δ3XCTCF:71Kb INV −14 replicates using two different clonal lines. P-values were calculated using two-sided unpaired t-tests (NS (not significant): fold-change < 2 or p > 0.05). **E** Capture-C experiments in WT, Δ3XCTCF, 71KbINV and Δ3XCTCF:71KbINV ESC using *Gbx2* SE as a viewpoint (VP). Average Capture-C signals of the two replicates performed for each mESC line are shown around the *Gbx2/Asb18* locus either individually (upper tracks) or after subtracting the WT Capture-C signals (lower tracks). The TAD boundary containing the three CTCF sites is highlighted in light blue. The red arrows indicate the 71 Kb inversion breakpoints. **F** The average Capture-C signals shown in (**E**) were measured within the *Asb18* gene (chr1:89952677-90014577 (mm10)) and within five different 30 Kb control regions (Ctrls) located within the *Gbx2* TAD (red asterisks in (**E**) indicate the midpoint of the controls regions (mm10): chr1:89803398-89833398; chr1:89753398-89783398; chr1:89703398-89733398; chr1:89653398-89683398; chr1:89603398-89633398). Capture-C signals are shown for the Δ3XCTCF, 71KbINV and Δ3XCTCF:71KbINV ESC as log2 fold-changes with respect to WT ESC. In (**C, D**) box plots, the upper and lower parts of the box are the upper and lower quartiles, respectively, the horizontal line that splits the box in two is the median and the upper and lower whiskers indicate the maximum and minimum, respectively. Source RT-qPCR data are provided as a Source Data file.

*Gbx2* could prevent a further increase in *Asb18* expression through promoter competition rather than enhancer blocking[47,48]. To test this hypothesis, we generated mESC lines with the following homozygous re-arrangements (Fig. S14): (i) *ΔPromGbx2* - a small (1 Kb) deletion spanning the *Gbx2* promoter that reduces *Gbx2* expression (Fig. 4A, B) and (ii) *Δ3XCTCF:ΔPromGbx2* - deletions spanning the *Gbx2* promoter and the CTCF cluster, respectively (Fig. 4A). Importantly, the 1 Kb promoter deletion should theoretically disrupt both the enhancer-blocking and promoter competition capacity of *Gbx2*, while barely changing the linear distance between *Asb18* and the SE (Fig. 4A). The promoter deletion alone resulted in elevated *Asb18* expression levels (Fig. 4C). Most interestingly, the combination of the *Gbx2* promoter and CTCF cluster deletions lead to a strong and synergistic increase in *Asb18* expression (61.8 fold-change in Δ3xCTCF:ΔPromGbx2 *vs* WT; 5.5 and 7.5 fold-changes in Δ3xCTCF and ΔPromGbx2 *vs* WT, respectively) (Fig. 4C), resulting in *Asb18* expression levels that were considerably higher than those observed in cells with both the CTCF deletion and the 71 Kb inversion (61.8 fold-change in Δ3xCTCF:ΔPromGbx2 *vs* WT; 17.6 fold-change in Δ3xCTCF:71 Kb INV *vs* WT) (Fig. 3D *vs* Fig. 4C). Next, we also performed Capture-C experiments in these cell lines using again the *Gbx2* SE and the *Asb18* promoters as viewpoints. Remarkably, the deletion of the *Gbx2* promoter, either alone or in combination with the CTCF cluster deletion, showed minor effects on *Asb18*-SE contact frequency in comparison to WT and Δ3xCTCF cells, respectively (Fig. 4D, E; Fig. S15B, C). Furthermore, the deletion of the *Gbx2* promoter did not have major effects on the interaction frequency between either *Gbx2* or *Asb18* and the 3XCTCF cluster (Fig. 4D, Fig. S15A, B, D). Accordingly, RAD21 ChIP-seq experiments showed that cohesin profiles within the *Asb18* locus were not significantly affected by the *Gbx2* promoter deletion (Fig. S16A). On the other hand, H3K27ac ChIP-seq signals at the *Gbx2* SE were rather similar among all the generated cell lines, albeit slightly elevated in the CTCF deletion cells (Fig. S16B), arguing that the observed *Asb18* expression changes are not caused by differences in enhancer activity.

Overall, these results support the notion of *Gbx2* preferentially contributing to regulatory (rather than physical) insulation through promoter competition. Importantly, our data also suggest that *Gbx2* and the CTCF cluster cooperatively confer the nearby 3′ TAD boundary with strong insulator capacity (Table 1).

## A large CTCF cluster prevents promoter competition between *Six3* and *Six2*

In order to test whether the previous observations could be extended to other developmental boundaries and cellular contexts, we then generated a similar set of genetic re-arrangements within the *Six3/Six2* locus (Fig. 5A). *Six3* and *Six2* are two typical developmental genes (*i.e.* with broad PcG domains and large CGI clusters around their promoters) that are located close to each other in linear space (~68 Kb between *Six3* and *Six2*). However, there is a strong and conserved TAD boundary containing seven CTCF sites and spanning ~50 Kb that separates the *Six3* and *Six2* regulatory domains (Fig. 5A)[82,83]. Accordingly, *Six3* and *Six2* display largely non-overlapping expression patterns during embryogenesis: *Six3* is expressed in the developing forebrain or the eye, while *Six2* shows high expression in the developing kidney or the facial mesenchyme[82,83]. We previously showed that within the *Six3* TAD there is a conserved SE that specifically activates *Six3*, but not *Six2*, in neural progenitors (NPC) and the developing forebrain (*Six3* expression in NPC: 10.33 FPKM; *Six2* expression in NPC: 0.21 FPKM)[58,63]. With this information at hand, we first generated mESC lines with the following homozygous re-arrangements: (i) *156Kb INV* - 156 Kb inversion between *Six3* and its SE that places the enhancer close to the TAD boundary and in between *Six3* and *Six2*; (ii) *Δ6XCTCF* - 36 Kb deletion that eliminates six out of the seven CTCF sites separating *Six3* and *Six2*[63]; (ii) *Δ6XCTCF:156Kb INV* - both the 156 Kb inversion and the 36 Kb deletion (Fig. 5B; Fig. S17). Once multiple clonal mESC lines were obtained for each of these re-arrangements, we differentiated them into NPC and measured the expression of *Six3* and *Six2* (Fig. 5C, D).

Similarly to our results for the *Gbx2* locus, the 156 Kb inversion did not significantly affect *Six3* expression (Fig. 5C). It is worth noting that this inversion places *Six3* close to a single CTCF peak located next to the SE (Fig. 5A). Although we previously showed that this CTCF site was not required for *Six3* induction in NPC[63], it could theoretically contribute to *Six3* expression in cells with the inversion. Therefore, we engineered a 226 Kb inversion that places *Six3* further away from the TAD boundary while preserving the linear distance with respect to the enhancer (Fig. 5E; Fig. S18). Notably, this 226 Kb inversion did not affect *Six3* induction in NPC either (Fig. 5F). These results further suggest that the location of developmental genes close to TAD boundaries/CTCF clusters does not universally facilitate their own expression. Furthermore, and in contrast to our findings for the *Gbx2* locus, neither the 156 Kb nor the 226 Kb inversion affected *Six2*, which remained lowly expressed in NPC (Fig. 5D, F). This could be explained by the differences between the CTCF clusters present at each locus (seven CTCF sites spanning ~50 Kb at the *Six3/Six2* locus *vs* three CTCF sites spanning ~10 Kb at the *Gbx2/Asb18* locus), which is significantly larger in the *Six3/Six2* locus and, thus, could confer stronger physical insulation.

On the other hand, and in contrast to our observations for the *Gbx2/Asb18* locus, the deletion of the CTCF cluster not only led to the ectopic activation of *Six2* (Fig. 5D), but also significantly impaired *Six3* induction in NPC (Fig. 5C). Considering the results obtained with the inversions described above, the reduced expression of *Six3* can not be simply attributed to a potential role of the CTCF sites as facilitators of enhancer-gene communication. Recent work in

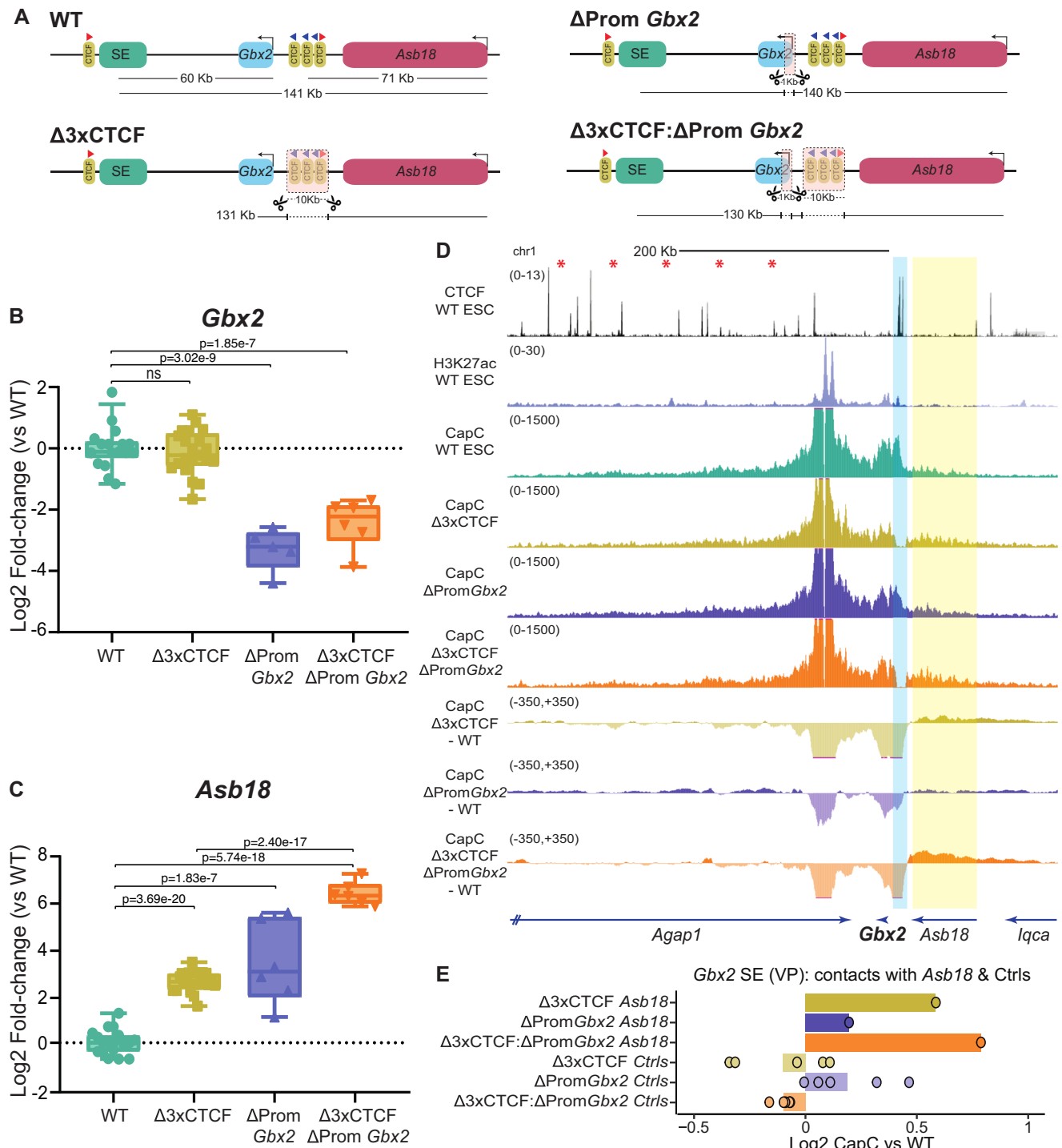

**Fig. 4 | *Gbx2* and the nearby CTCF cluster cooperatively contribute to the regulatory insulator capacity of the nearby TAD boundary. A** Graphical overview of different genomic rearrangements generated in mESC within the *Gbx2/Asb18* locus: Δ3XCTCF (same as in Fig. 3B); ΔPromGbx2; Δ3XCTCF:ΔPromGbx2. The expression of *Gbx2* (**B**) and *Asb18* (**C**) was measured by RT–qPCR in ESC that were either WT or homozygous for the genomic re-arrangements described in (**A**). For each cell line, *Gbx2* and *Asb18* expression was measured in the following number of biological replicates: WT −18 replicates (same as in Fig. 3C, D); Δ3XCTCF −21 replicates using two different clonal lines (same as in Fig. 3C, D); ΔPromGbx2 -5 replicates for *Gbx2* and 6 replicates for *Asb18* using two different clonal lines; Δ3XCTCF:ΔPromGbx2 -6 replicates using two different clonal lines. Expression differences among ESC lines were calculated using two-sided unpaired t-tests (NS (not significant): fold-change < 2 or p > 0.05). **D** Capture-C experiments in WT, Δ3XCTCF, ΔPromGbx2 and Δ3XCTCF:ΔPromGbx2 ESC using the *Gbx2* SE as a

viewpoint. The average Capture-C signals of the two replicates performed for each mESC line are shown around the *Gbx2/Asb18* locus individually (upper tracks) or after subtracting the WT signals (lower tracks). The three CTCF sites deleted within the *Gbx2/Asb18* TAD boundary are highlighted in light blue. Capture-C tracks for WT and Δ3XCTCF ESC are the same as in Fig. 3E. **E** The average Capture-C signals shown in (**D**) were measured within the *Asb18* gene (highlighted in yellow in (**D**); chr1:89952677-90014577 (mm10) as well as within five different 30 Kb control regions (Ctrls) located within the Gbx2 TAD (red asterisks in (**D**) indicate the midpoint of the same control regions as in Fig. 3F). Capture-C signals are shown for the Δ3XCTCF, ΔPromGbx2 and Δ3XCTCF:ΔPromGbx2 ESC as log2 fold-changes with respect to WT ESC. In (**B**, **C**) box plots, the upper and lower parts of the box are the upper and lower quartiles, respectively, the horizontal line that splits the box in two is the median and the upper and lower whiskers indicate the maximum and minimum, respectively. Source RT-qPCR data are provided as a Source Data file.

**Table 1 | Summary of the main results obtained for the Gbx2/Asb18 locus**

| Cell line name | Cell line description | Gene expression analyses | Capture-C experiments | Conclusions |
|---|---|---|---|---|
| **Δ3xCTCF** (Fig. 3; Fig. S13) | 10 Kb deletion that eliminates the three CTCF sites separating the Gbx2 and Asb18 TADs. | Gbx2 expression is not affected. Asb18 expression is significantly increased (~5-fold). | The contact frequency between the Gbx2 SE and Gbx2 is reduced. The contact frequency between the Gbx2 SE and Asb18 is increased. | The positioning of Gbx2 close to the 3XCTCF cluster does not play a major role in the maintenance of Gbx2 expression in ESC. The CTCF cluster strengthens the physical insulation of the Gbx2 regulatory domain, preventing spurious enhancer-gene contacts. |
| **71Kb INV** (Fig. 3; Fig. S13) | Inversion that re-positions Gbx2 and its SE with respect to the CTCF cluster, placing the SE in between Gbx2 and Asb18. | Gbx2 expression is not affected. Asb18 expression is significantly increased (~16-fold). | The contact frequency between the Gbx2 SE and Gbx2 is reduced. Minor impact on Asb18-Gbx2 SE contacts. Strong increase in the contact frequency between the Gbx2 SE and the 3XCTCF cluster. | The positioning of Gbx2 close to the 3xCTCF cluster does not play a major role in the maintenance of Gbx2 expression in ESC. Gbx2 does not contribute to physical insulation through enhancer blocking. The increased expression of Asb18 might be explained by the reduced linear distance between Asb18 and the Gbx2 SE in 71Kb INV cells. |
| **Δ3xCTCF:71Kb INV** (Fig. 3; Fig. S13) | Cells with both the Δ3xCTCF deletion and the 71 Kb inversion. | Gbx2 expression is not affected. Asb18 expression is increased to similar levels as in 71Kb INV cells. | The contact frequency between the SE and Gbx2 is reduced. The contact frequency between the Gbx2 SE and Asb18 is strongly increased. | Non-linear relationship between gene activity and E-P contact frequency: the increase in Gbx2 SE-Asb18 contacts does not lead to a further increase in Asb18 expression. |
| **ΔProm Gbx2** (Fig. 4; Fig. S15) | 1 Kb deletion spanning the Gbx2 promoter. | Asb18 expression is significantly increased (~7-fold). | Minor impact on Asb18-Gbx2 SE contacts. | Gbx2 contributes to regulatory rather than physical insulation through promoter competition. |
| **Δ3xCTCF:ΔProm Gbx2** (Fig. 4; Fig. S15) | Cells with both the Δ3xCTCF and ΔProm Gbx2 deletions. | Strong and synergistic increase in Asb18 expression (~60-fold). | The contact frequency between the Gbx2 SE and Asb18 is increased to a similar extent as in Δ3xCTCF cells. | Gbx2 contributes to regulatory rather than physical insulation through promoter competition. Gbx2 and the CTCF cluster cooperatively confer the nearby TAD boundary with strong insulator capacity. |

*Drosophila* indicates that insulators protect genes from spurious interactions with not only enhancers but also silencers located within neighbouring TADs[84]. Therefore, the CTCF cluster might protect *Six3* from the repressive effects of putative silencer elements located within the *Six2* TAD and/or from the promoter competition activity of *Six2*. Since there are no robust chromatin signatures to globally identify silencers, we decided to test the *Six2* promoter competition hypothesis by generating mESC lines with deletions spanning both the CTCF cluster and *Six2* (Fig. 5G; Fig. S19). Interestingly, the *Six2* deletion rescued, albeit partly, the impaired induction of *Six3* in NPC (Fig. 5H). This suggests that, rather than facilitating the communication between *Six3* and its cognate enhancers, the CTCF cluster protects *Six3* from promoter competition by *Six2*. However, since the *Six2* deletion led to a partial rather than total rescue of *Six3* expression levels, the CTCF cluster might also protect *Six3* from putative silencers found within the *Six2* TAD[84].

## Cooperative insulation of the *Six3* and *Six2* regulatory domains by the CTCF cluster and promoter competition

The previous results suggest that, in the absence of the CTCF cluster, *Six2* competes with *Six3* for the NPC enhancers located within the *Six3* TAD. Therefore, we wondered whether this promoter competition could be reciprocal and, thus, *Six3* could participate, together with the CTCF cluster, in the robust insulation of the *Six3* regulatory domain. To explore this possibility, we generated mESC lines with (i) a 27 Kb deletion spanning the *Six3* gene (i.e. *Six3*⁻/⁻); (ii) deletions spanning both *Six3* and the CTCF cluster (*i.e. Δ6XCTCF:Six3*⁻/⁻) (Fig. 6A; Fig. S20). All the resulting clonal ESC lines were differentiated into NPC, in which we measured *Six2* and *Six3* expression (Fig. 6A-B).

Remarkably, although the *Six3* deletion alone mildly changed (albeit in a non-statistically significant manner) *Six2* expression in NPC, *Six2* expression was strongly and synergistically increased when the *Six3* and CTCF cluster deletions were combined (Fig. 6C) (72.1 Fold-change in Δ6xCTCF:*Six3*⁻/⁻ vs WT; 8.8 fold change in Δ6xCTCF and 1.7 fold-change in *Six3*⁻/⁻ vs WT, respectively), thus in agreement with the contribution of *Six3* to regulatory insulation. Next, we performed Capture-C experiments in these cell lines using the *Six3* SE or the *Six2* promoter as viewpoints. As expected, the deletion of the CTCF cluster led to a loss of physical insulation between the *Six3* and *Six2* TADs and increased the contact frequency between *Six2* and the SE (Fig. 6D, E; Fig. S21C, D). In addition, this deletion strongly increased the contacts between *Six3* and *Six2* (Fig. S21C, E), in agreement with the formation of a shared transcriptional hub[51]. In contrast, the CTCF cluster deletion did not affect the contact frequency between the SE and *Six3* (Fig. 6D, Fig. S21A), further indicating that the 6XCTCF cluster does not play a major role as a facilitator of Six3-SE communication in NPC. Notably, the deletion of *Six3* did not change the *Six2*-SE contact frequency in comparison to WT cells (Fig. 6D, E; Fig. S21C, D) and only led to a minor increase in the interaction frequency between either the SE or *Six2* and the 6XCTCF cluster (Fig. 6D, Fig. S21B, C, F). Finally, the combined deletion of *Six3* and the CTCF cluster, which resulted in a strong induction of *Six2*, mildly increased the *Six2*-SE contact frequency in comparison to the CTCF deletion alone (Fig. 6D, E; Fig. S21C, D). Altogether, these results suggest that, as observed for the *Gbx2/Asb18* locus, *Six3* preferentially contributes to regulatory insulation through promoter competition, while *Six3*-dependent enhancer blocking seems to have a comparably smaller, albeit non-negligible, contribution. Therefore, enhancer blocking (i.e. physical insulation) is mostly dependent on the CTCF cluster, which together with *Six3*-mediated promoter competition cooperatively confer the TAD boundary with strong insulator capacity. Importantly, the contribution of *Six3* to regulatory insulation through promoter competition can also explain why the 156Kb and 226Kb inversions described above did not have any major impact on *Six2* expression either alone or in combination with the CTCF cluster deletion (Fig. 5D, F). In agreement with this model,

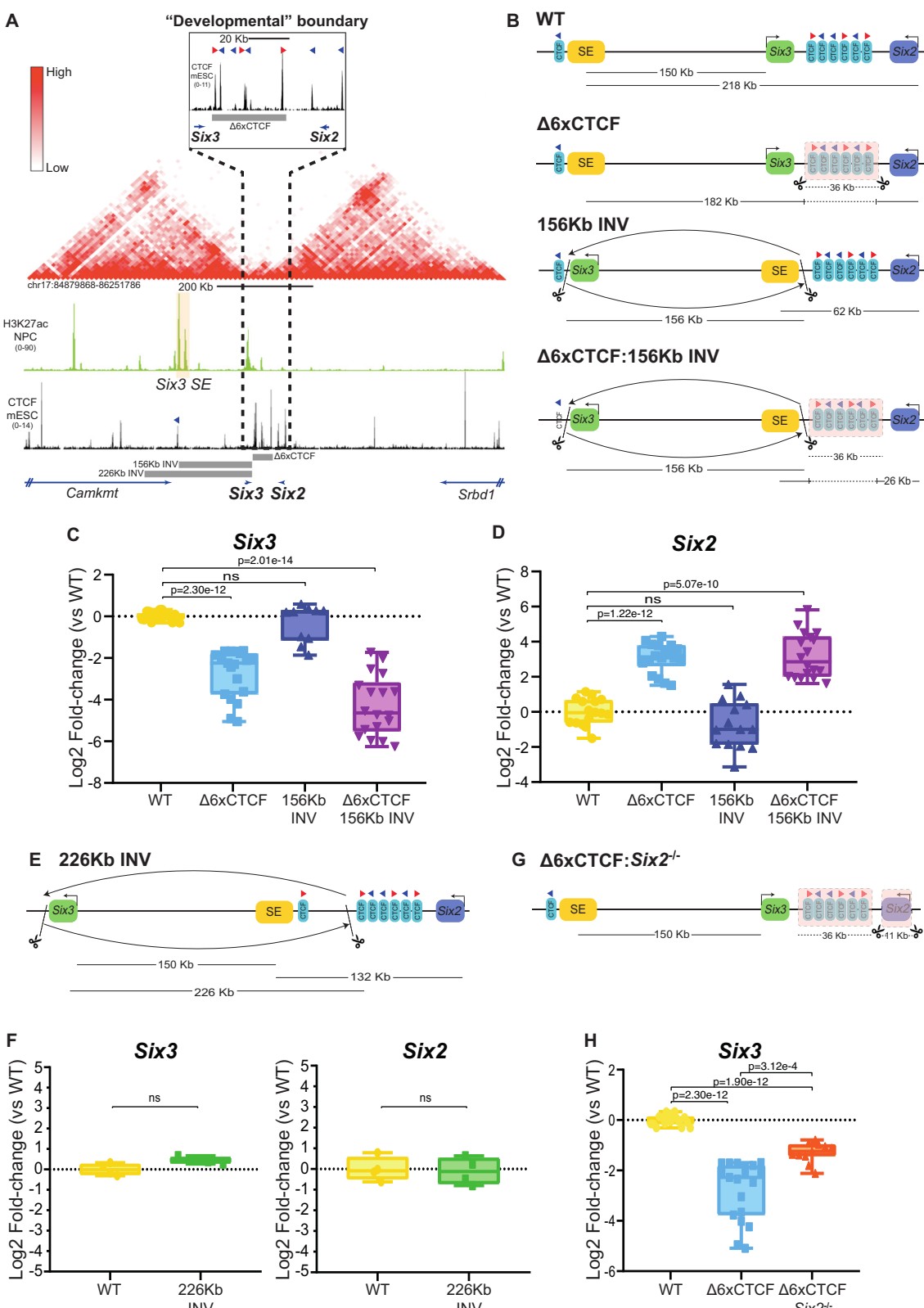

RAD21 ChIP-seq experiments showed that the CTCF cluster deletion increased cohesin levels around *Six2*, particularly at a CTCF site located upstream of this gene (red asterisk in Fig. S22A), while cohesin profiles were not affected by the *Six3* deletion (Fig. S22A). On the other hand, H3K27ac ChIP-seq signals at the *Six3* SE were rather similar among all the generated cell lines, except in the *Six3^−/−^* cells in which we

observed higher H3K27ac levels, arguing that the observed *Six2* expression changes are not caused by differences in enhancer activity (Fig. S22B).

Overall, our data suggest that, as for the *Gbx2/Asb18* locus, *Six3* and the CTCF cluster cooperatively contribute to the robust insulation of the *Six3* and *Six2* regulatory domains (Table 2).

**Fig. 5 | The large CTCF cluster separating the *Six3* and *Six2* TADs prevents promoter competition between these two genes. A** NPC Hi-C[19] data shows that *Six3* and *Six2* are located in neighbouring TADs separated by a cluster of seven CTCF sites[122]. The orientation of key CTCF sites is illustrated with red (sense) and blue (antisense) triangles. **B** Graphical overview of genomic rearrangements generated in mESC within the *Six3/Six2* locus: Δ6XCTCF; 156KbINV; Δ6XCTCF:156KbINV. **C, D** The expression of *Six3* and *Six2* was measured by RT−qPCR in NPC that were WT or homozygous for the genomic re-arrangements described in (B). For each cell line, *Six3* and *Six2* expression was measured in the following number of biological replicates: WT -16 replicates; Δ6XCTCF -19 replicates using a previously characterized clonal line[63]; 156KbINV −10 replicates for Six3 and 14 replicates for *Six2* using two different clonal lines; Δ6XCTCF:156KbINV −18 replicates using three different clonal lines. **E** Graphical overview of the 226KbINV mESC line. **F** *Six3* and *Six2* expression was measured by RT−qPCR in NPC that were

WT or homozygous for the 226KbINV. For each cell line, *Six3* and *Six2* expression was measured in the following number of biological replicates: WT -4 replicates; 226KbINV -4 replicates using two different clonal lines. **G** Graphical overview of the Δ6XCTCF:Six2[−/−] mESC line. **H** *Six3* expression was measured by RT−qPCR in NPC that were either WT or homozygous for the Δ6XCTCF and Δ6XCTCF:Six2[−/−] deletions. For each cell line, *Six3* was measured in the following number of biological replicates: WT −16 replicates (same as in C); Δ6XCTCF −19 replicates (same as in C); Δ6XCTCF:Six2[−/−] -11 replicates using three different clonal lines. Expression differences among cell lines were calculated using two-sided unpaired t-tests (NS (not significant): fold-change < 2 or p > 0.05). In (**C, D, F, H**) box plots, the upper and lower parts of the box are the upper and lower quartiles, respectively, the horizontal line that splits the box in two is the median and the upper and lower whiskers indicate the maximum and minimum, respectively. Source RT-qPCR data are provided as a Source Data file.

## Large CTCF clusters prevent competition between adjacent developmental genes

One major difference between the results obtained for the *Gbx2/Asb18* and *Six3/Six2* loci is that the deletion of the corresponding CTCF clusters impaired the induction of *Six3*, but did not significantly affect the expression of *Gbx2*. We hypothesized that this could be attributed to the fact that *Six2* and *Six3* are both typical developmental genes, which are characterized by having promoters with large clusters of CpG islands and strong long-range enhancer responsiveness (Fig. 5A)[63,65,67]. On the contrary, while *Gbx2* is also a developmental gene with a large CGI cluster around its promoter, *Asb18* is a tissue specific gene with a CpG-poor promoter, which typically show weak long-range enhancer responsiveness (Fig. 3A)[63,67]. Consequently, in the absence of CTCF sites, *Six2* and *Six3* might engage into strong promoter competition for shared enhancers, while *Asb18* might represent a weak competitor against *Gbx2*. Furthermore, the regulatory domains of developmental genes, such as *Six2*, might often contain silencers whose long-range repressive capacity needs to be properly insulated[84,85]. Therefore, we hypothesize that the boundary insulating the *Six3* and *Six2* regulatory domains might require a particularly large CTCF cluster (seven CTCF sites spanning ~50 Kb) in comparison to the one found between *Gbx2* and *Asb18* (three CTCF sites spanning ~10 Kb) (Figs. 3A and 5A). To test this prediction, we generated mESC lines with a deletion that removed four out of the seven CTCF sites separating the *Six3* and *Six2* domains (Fig. 6A; Fig. S23). In addition, we also generated mESC lines with the partial deletion of the CTCF cluster plus the *Six3* deletion (Fig. 6A). Interestingly, the partial deletion of the CTCF cluster alone did not significantly increase *Six2* expression, but strongly impaired *Six3* induction (Fig. 6F, G). Together with the results obtained for the inversions and the 6XCTCF deletion described above (Fig. 5), this further suggests that, rather than facilitating *Six3* expression, the CTCF cluster separating the *Six3* and *Six2* TADs might protect *Six3* from promoter competition by *Six2* and, potentially, from the repressive effects of putative silencers located within the *Six2* TAD[33,84], although the latter possibility remains speculative and would require further experimental evidences. Last but not least, the combination of the partial deletion of the CTCF cluster and the *Six3* deletion led to a strong and synergistic induction of *Six2* in NPC (Fig. 6G). This further supports that *Six3* contributes to regulatory insulation through promoter competition and, more generally, that the robust insulation provided by developmental boundaries requires the cooperative contribution of CTCF clusters and transcriptionally active developmental genes (Tables 1–2).

## Discussion

During embryogenesis, developmental genes are typically expressed in several spatial and/or temporal contexts, which is enabled by the presence of multiple, distinct and highly specific enhancers within regulatory domains delimited by insulators (e.g. CTCF sites)[3,86].

Moreover, within a particular cellular context, the expression of developmental genes is often controlled by enhancer clusters (i.e. SE, locus control regions (LCR)) with strong regulatory activity[87]. However, CTCF sites might not be sufficient to restrain the regulatory activity of these complex enhancer landscapes towards their target genes[33,34], suggesting that additional mechanisms might be necessary to ensure gene expression specificity and the robust insulation of developmental TADs. Here we report that developmental TADs are gene-poor and often delimited by rather unique boundaries. More specifically, we show that, within these TADs, developmental genes and CTCF clusters are often sequentially organized (with CTCF peaks being predominantly located between genes and TAD boundaries) and cooperatively strengthen the regulatory insulation capacity of nearby boundaries, thus contributing to the establishment of specific expression patterns during development.

The large size and low gene density of developmental TADs might be required to accommodate the complex enhancer landscapes responsible for the specific, yet diverse, spatiotemporal expression patterns that many developmental genes display during embryogenesis[3,86,88]. Furthermore, due to the long-range responsiveness of developmental gene promoters[63,67], their cognate enhancers can be separated by large linear distances and, thus, located anywhere within developmental TADs. In contrast, the regulatory elements controlling the expression of either housekeeping or tissue-specific genes tend to be proximal or even embedded within promoter regions[89,90]. Moreover, as housekeeping and tissue-specific genes show lower long-range enhancer responsiveness than developmental genes, they might be able to co-exist with other genes with similar promoter types within gene-rich TADs without interfering with each other's expression profiles[14,91].

Here we show that developmental genes and CTCF clusters cooperatively strengthen the insulation capacity of nearby boundaries (i.e. developmental boundaries). In agreement with previous work, the physical insulation of the investigated TADs was mostly dependent on the CTCF clusters[36,77,78,92]. In contrast, the investigated developmental genes do not seem to act as strong physical barriers preventing ectopic enhancer-gene communication (*i.e.* enhancer blocking), but preferentially contribute to regulatory insulation through promoter competition. The contribution of developmental genes to regulatory insulation through non-structural mechanisms is also in agreement with the fact that developmental boundaries show similar insulation scores (*i.e.* physical insulation) to those observed for TAD boundaries in general (Fig. 2A, B). For developmental genes located close to TAD boundaries, the majority of their cognate enhancers should be located at more central positions within the same TAD. Consequently, the linear distance between the boundary-proximal developmental genes and their cognate enhancers should be often smaller than the distance between those same enhancers and non-target genes located at the other side of the boundary within the adjacent TAD. Considering that

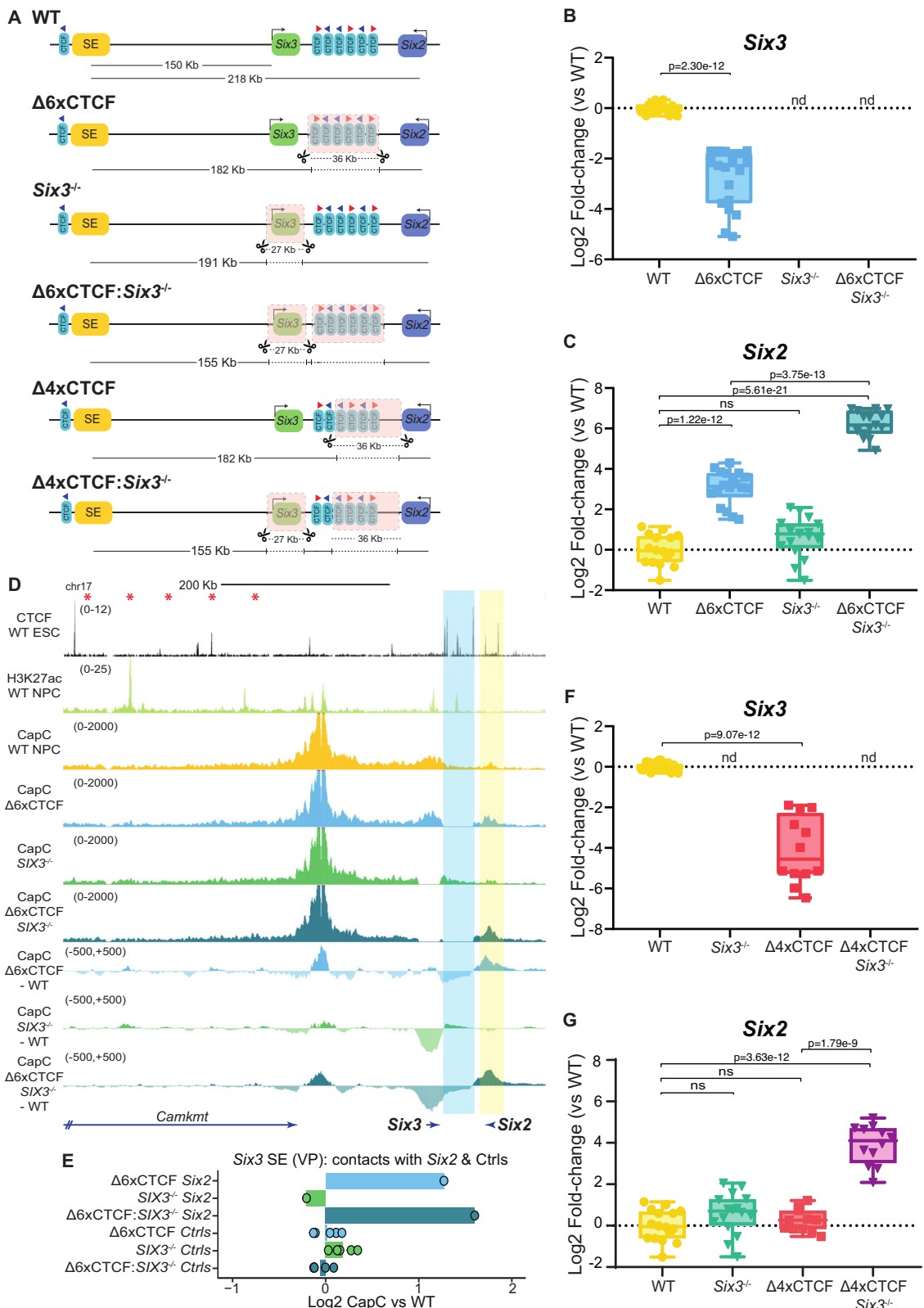

short linear distances facilitate the functional communication between genes and enhancers[41,80,81], we hypothesize that the shorter linear distances separating boundary-proximal genes and their cognate enhancers might give those developmental genes a competitive advantage over non-target genes located on the other side of the developmental boundary[93]. Recent studies indicate that the physical insulation provided by CTCF boundaries is dynamic and partial, with

strong enhancers, such as those typically controlling the expression of major developmental genes, being able to bypass those boundaries and activate genes across TADs[33–36]. At the single cell level, the permeability of CTCF boundaries might result in a small fraction of cells/alleles in which cohesin-mediated loops bypass those boundaries and enable spurious contacts between enhancers and non-target genes. In the context of developmental loci, such as those investigated in our

**Fig. 6 | *Six3* contributes to the robust insulation of the *Six3* and *Six2* regulatory domains through promoter competition. A** Genomic rearrangements generated in mESC within the *Six3/Six2* locus: Δ6XCTCF; Six3[-/-]; Δ6XCTCF: Six3[-/-]; Δ4XCTCF; Δ4XCTCF:Six3[-/-]. **B, C** *Six3* and *Six2* expression was measured by RT−qPCR in NPC homozygous for the indicated re-arrangements using the following number of biological replicates: WT −16 replicates (same as in Fig. 5C, D); Δ6XCTCF −19 replicates using a previously characterized clonal line[63] (same as in Fig. 5C, D); Six3[-/-] −16 replicates using three clonal lines; Δ6XCTCF:Six3[-/-] -15 replicates using three clonal lines. **D** Capture-C experiments in WT, Δ6XCTCF, Six3[-/-] and Δ6XCTCF:Six3[-/-] NPC using *Six3* SE as a viewpoint. Average Capture-C signals of the two replicates performed for each cell line are shown around the *Six3/Six2* locus individually (upper tracks) or after subtracting the WT signals (lower tracks). **E** Average Capture-C signals were measured around *Six2* (highlighted in yellow in (**D**); chr17:85674267-85698254 (mm10)) and within five 30 Kb control regions (Ctrls) located within the *Six3* TAD (red asterisks in (**D**)) indicate the midpoint of the controls regions (mm10): chr17:85388601-85418601;

chr17:85338601-85368601, chr17:85288601-85318601, chr17:85238601-85268601, chr17:85188601-85218601). Capture-C signals are shown for the Δ6XCTCF, Six3[-/-] and Δ6XCTCF:Six3[-/-] NPC as log2 fold-changes with respect to WT NPC. **F, G** *Six3* and *Six2* expression was measured by RT−qPCR in NPC homozygous for the indicated genomic re-arrangements using the following number of biological replicates: WT −16 replicates (same as in Fig. 5C, D); Six3[-/-] −16 replicates using three different clonal lines (same as in Fig. 6C); Δ4XCTCF −12 replicates for *Six3* and 11 replicates for *Six2* using three different clonal lines; Δ4XCTCF:Six3[-/-] −12 replicates using three different clonal lines. Expression differences among cell lines were calculated using two-sided non-paired t-tests (NS (not significant): fold-change < 2 or p > 0.05); ND (not detectable). In (**B, C, F, G**) box plots, the upper and lower parts of the box are the upper and lower quartiles, respectively, the horizontal line that splits the box in two is the median and the upper and lower whiskers indicate the maximum and minimum, respectively. Source RT-qPCR data are provided as a Source Data file.

work, this can lead to the formation of multiway contacts (hubs) in which enhancers can simultaneously interact with their target developmental genes as well as with non-target genes located in neighboring domains[51]. However, we speculate that the strong enhancer responsiveness of developmental genes[63,64] together with shorter linear distances with respect to their cognate enhancers[93] might give developmental genes a competitive advantage over non-target genes and, thus, prevent the spurious contacts between enhancers and non-target genes from being productive.

At the molecular level, it is currently unclear which mechanisms might explain how active genes contribute to insulation through promoter competition. Recent work suggests that RNA Pol2 can act as a weak physical barrier against cohesin-mediated loop extrusion[21–23,25], although the overall contribution of RNA Pol2 to chromatin architecture and physical insulation remains controversial[28,29]. In this regard, forced activation of candidate genes in ESC does not cause changes in chromatin insulation, suggesting that transcription and RNA Pol2 recruitment are not sufficient for the emergence of TAD boundaries[19]. Moreover, the potential role of RNA Pol2 as a loop extrusion barrier can not explain how developmental genes contribute to insulation through promoter competition, as this does not require the placement of developmental genes in between enhancers and non-target genes (Figs. 3 and 5)[47,48]. Instead, promoter competition might involve non-structural mechanisms whereby multiple promoters compete for rate-limiting factors (e.g. TFs, GTFs, RNA Pol2) present within shared transcriptional hubs[56,57,94]. Although these models and the putative rate-limiting factors await experimental validation, we speculate that nearby promoters with strong enhancer responsiveness and similar architecture (e.g. *Six3* and *Six2*) might be particularly sensitive to promoter competition unless insulated by large CTCF clusters. On the other hand, the positioning of developmental genes close to the boundaries should often place them in between their cognate enhancers and non-target genes located in the adjacent domain, thus potentially enabling developmental genes to display enhancer blocking function. In this regard, it is possible that the developmental genes investigated in this study (i.e. *Gbx2* and *Six3*) act as weak enhancer blockers whose effects on physical insulation can not be easily detected due to the current resolution and sensitivity of Capture-C and other 3C methods, but that, nevertheless, can still result in transcriptional changes[80,95]. Furthermore, in the context of boundaries close to strongly expressed genes (e.g. globin genes), the contribution of active transcription and RNA Pol2 complexes to physical insulation might be larger[24,43–46] in comparison to most developmental genes that, even when active, are expressed at comparably moderate levels.

It has been previously proposed that TAD boundaries and the associated CTCF clusters can (i) facilitate the communication between genes and enhancers located within the same domain and (ii) insulate

genes from the regulatory activity of enhancers located in other domains. Although CTCF clusters might facilitate ultra long-range enhancer-gene contacts (e.g. *Shh, Sox9*)[39,96], at shorter distances (such as those separating *Gbx2* and *Six3* from their SEs in ESC and NPC, respectively) CTCF sites seem to be dispensable and other types of tethering elements (e.g. CpG islands in vertebrates, GAGA elements in *Drosophila*) can mediate intra-TAD enhancer-gene communication[63,84]. However, although proper expression of *Gbx2* and *Six3* in ESC and NPC, respectively, does not seem to require the proximity of these genes to CTCF clusters, it is certainly possible that the CTCF sites facilitate the communication of *Gbx2* and/or *Six3* with more distal enhancers in other cellular contexts. Furthermore, our data as well as recent work in *Drosophila*[84] suggest that, in the context of developmental boundaries, one major role for architectural protein clusters (e.g. CTCF) is to robustly insulate the regulatory domains of major developmental genes and prevent spurious contacts with enhancers and/or silencers[84]. Accordingly, global 3D genome organization studies show that developmental genes and their cognate enhancers are typically located within the same TADs, suggesting a strong functional overlap between topological and regulatory domains[14]. Nevertheless, there are loci, with the *Hox* gene clusters being a notable example, in which genes and their cognate enhancers can be located in different TADs[30,32,97]. In these cases, the location of genes within or close to TAD boundaries might allow them to bypass those boundaries and communicate with enhancers located in more than one TAD in order to achieve proper gene expression patterns[31,32,36,98]. In contrast, for the two developmental genes that we investigated, we found no evidences indicating that CTCF clusters facilitate their expression or the communication with their cognate enhancers. Instead, we found that the sequential organization of genes and CTCF clusters at developmental boundaries is a prevalent and evolutionary conserved feature that seems to strengthen the insulation of important developmental loci.

Lastly, our findings might also have relevant medical implications, particularly in the context of structural variants (SV) causing congenital defects through long-range pathomechanisms[42]. For example, deletions spanning TAD boundaries can lead to pathological gains in gene expression through enhancer adoption mechanisms whereby enhancers can induce the expression of non-target genes[42,99]. Our data indicates that the pathological effects of this type of SVs might change (i.e. stronger or weaker induction of non-target genes) depending on whether the deletions include not only CTCF sites, but also nearby developmental genes.

## Methods
All the experiments presented in this work comply with all relevant ethical and biosafety regulations and have been approved by the IBBTEC Biosafety commission and the CSIC Ethics committee.

**Table 2 | Summary of the main results obtained for the Six3/Six2 locus**

| Cell line name | Cell line description | Gene expression analyses | Capture-C experiments | Conclusions |
|---|---|---|---|---|
| **Δ6xCTCF** (Figs. 5 and 6; Fig. S21) | 36 Kb deletion that eliminates six out of the seven CTCF sites separating the Six3 and Six2 TADs. | Six3 induction is significantly impaired (~5-fold). Six2 expression is significantly increased (~9-fold). | The contact frequency between the Six3 SE and Six3 is not affected. Increased contact frequency between Six3 and Six2. The contact frequency between the Six3 SE and Six2 is increased. | The positioning of Six3 close to the CTCF cluster might facilitate Six3 induction, although changes in Six3-SE contact frequency are not observed. The CTCF cluster strengthens the physical insulation of the Six3 regulatory domain, preventing spurious enhancer-gene contacts. |
| **156KbINV** (Fig. 5) | Inversion that re-positions Six3 and its SE with respect to the CTCF cluster, placing the SE in between Six3 and Six2. | Six3 expression is not affected. Six2 expression is not affected. | NA | The positioning of Six3 close to the CTCF cluster does not facilitate Six3 induction. The impaired induction of Six3 in Δ6xCTCF might involve alternative mechanisms. Six3 does not contribute to the insulation of its regulatory domain through enhancer blocking. |
| **Δ6xCTCF: 156KbINV** (Fig. 5) | Cells with both the Δ6xCTCF deletion and the 156 Kb inversion. | Six3 induction is significantly impaired. Six2 expression is increased to similar levels as in Δ6xCTCF. | NA | Six3 does not contribute to the insulation of its regulatory domain through enhancer blocking. |
| **Δ6xCTCF:Six2$^{-/-}$** (Fig. 5) | Cells with both the Δ6xCTCF deletion and a 11 Kb deletion spanning the Six2 gene. | Six3 induction is partially rescued (~2-fold reduction) in comparison to Δ6xCTCF cells (~5-fold reduction). | NA | The CTCF cluster might protect Six3 from promoter competition by Six2 and, potentially, from putative silencers found within the Six2 TAD (speculative). |
| **Six3$^{-/-}$** (Fig. 6; Fig. S21) | 27 Kb deletion spanning the Six3 gene. | Six2 expression is not affected. | Minor impact on Six2-Six3 SE contacts. | Six3 does not contributes to the physical insulation of its regulatory domain. |
| **Δ6xCTCF:Six3$^{-/-}$** (Fig. 6; Fig. S21) | Cells with both the Δ6xCTCF deletion and the Six3$^{-/-}$ deletion. | Strong and synergistic increase in Six2 expression (~70-fold). | The contact frequency between the Six3 SE and Six2 is mildly increased in comparison to Δ6xCTCF cells. | Six3 contributes to regulatory rather than physical insulation through promoter competition. Six3 and the CTCF cluster cooperatively confer the nearby TAD boundary with strong insulator capacity. |
| **Δ4xCTCF** (Fig. 6) | 36 Kb deletion that eliminates four out of the seven CTCF sites separating the Six3 and Six2 TADs. | Six3 induction is significantly impaired. Six2 expression is not affected. | NA | The CTCF cluster might protect Six3 from promoter competition by Six2 and, potentially, from putative silencers found within the Six2 TAD (speculative). |
| **Δ4XCTCF:Six3$^{-/-}$** (Fig. 6) | Cells with both the Δ4xCTCF and Six3$^{-/-}$ deletions. | Strong and synergistic increase in Six2 expression (~16-fold). | NA | Six3 and the CTCF cluster cooperatively confer the nearby TAD boundary with strong insulator capacity. |

## ESC maintenance and differentiation protocol

E14Tg2a (E14) mouse ESC (a kind gift from Joanna Wysocka's lab (Standford University) were cultured on gelatin-coated plates using Knock-out DMEM (Life Technologies, 10829018) supplemented with 15% FBS (Life Technologies, 10082147), leukemia inhibitory factor (LIF), antifungal and antibiotics (Sigma-Aldrich, A5955), β-mercaptoethanol (ThermoFisher Scientific, 21985023), Glutamax (ThermoFisher Scientific, 35050038) and MEM NEAA (ThermoFisher Scientific, 11140035). Cells were cultured at 37 °C with 5% $CO_2$.

For the NPC differentiation[63], ESCs were plated at 20,000 cells/$cm^2$ on geltrex-coated plates (ThermoFisher, A1413302) and grown in N2B27 medium: Advanced Dulbecco's Modified Eagle Medium F12 (Life Technologies, 21041025) and Neurobasal medium (Life Technologies, 12348017) (1:1), supplemented with 1 × N2 (R&D Systems, AR009), 1 × B27 (Life Technologies, 12587010), 2 mM L-glutamine (Life Technologies, 25030024) and 0.1 mM 2-mercaptoethanol (Life Technologies, 31350010)). During the six days of differentiation, the N2B27 medium was additionally supplemented with the following components: bFGF (ThermoFisher Scientific, PHG0368) 10 ng/ml from day 0 to day 2, Xav939 (Sigma-Aldrich, X3004-5MG) 5 μM from day 2 to day 6, BSA (ThermoFisher Scientific, 15260037) 1 mg/ml at day 0 and 40 μg/mL the remaining days.

**RNA isolation, cDNA synthesis and RT-qPCR.** Total RNA was isolated using the NZY Total RNA Isolation kit (NZYTech, MB13402) following the manufacturer's instructions. RNA was reverse transcribed into cDNA using the NZY First-Strand cDNA Synthesis Kit (NZYTech, MB13402). For each reaction, 1 μg of RNA was incubated with 10 μL of NZYRT 2x Master Mix, 2 μL of NZYRT Enzyme Mix and nuclease-free water to a total volume of 20 μL at 25 °C for 10 min, followed by 30 min at 50 °C. The enzyme was heat inactivated at 85 °C for 5 min. To digest the remaining RNA, 1 μL of NZY RnaseH was added to the reaction and incubated at 37 °C for 20 min.

RT-qPCRs were performed using the CFX 384 detection system (Bio-Rad) using NZYSpeedy qPCR Green Master Mix (2x) (NZYtech, MB224). For each sample, RT-qPCRs were performed as technical triplicates using the primers listed in Data S3. For each cell line, the number of clonal lines and biological replicates analysed in each case is indicated in the corresponding figure legends. For the investigated genes (i.e. *Gbx2*, *Asb18*, *Six3* and *Six2*), gene expression fold-changes between each cell line and the corresponding WT cells were calculated using the $2^{-\Delta\Delta CT}$ method, using *Eef1a1* and *Hprt1* as housekeeping genes. Fold-changes are shown as Log2 values and are plotted using box plots in which (i) the upper and lower parts of the box are the upper and lower quartiles, respectively, (ii) the horizontal line that splits the box in two is the median and (iii) the upper and lower whiskers indicate the maximum and minimum, respectively.

**CRISPR-Cas9 genome editing.** The design of the CRISPR/Cas9 guide sequences (gRNA) was performed using the CRISPR Benchling software tool (https://www.benchling.com/crispr/) (Data S3). In order to increase the cutting efficiency of the gRNA, a guanine nucleotide was added at the first position of the sequence and a restriction site for the BbsI enzyme (R0539L, NEB) was added to the beginning of the gRNA for cloning purposes. For each sgRNA, two oligonucleotides were synthesized (Sigma), annealed and cloned into a CRISPR-Cas9 expression vector (pX330-hCas9-long-chimeric-grna-g2p; gift from Leo Kurian's laboratory). The hybridized oligos were cloned into the pX330-hCas9-long-chimeric-grna-g2p using 50 ng of BbsI-digested vectors and 1 μl ligase (Thermo Fisher, EL0013) in a total volume of 20 μl. The ligation reaction was incubated for 1 h at room temperature. For each cell line, ESC were transfected with the corresponding pair of gRNAs-Cas9 expressing vectors using Lipofectamine following manufacturer's recommendations (Thermo Scientific, L3000001). After 24 h, transfected cells were selected by treating them with puromycin

for 48 h. Single-cell isolation of surviving cells was performed by serial dilution and seeding in 96-well plates. Next, clones with the desired genetic rearrangements (i.e. deletions or inversions) were identified by PCR using the primers listed in Data S3. DNA extraction was performed using Lysis Buffer: 25 mM KCl (SigmaAldrich, 27810.295), 5 mM TRIS (Sigma-Aldrich, 0497-5KG) pH8.3, 1.25 mM MgCl2 (VWR BDH7899-1), 0.225% IGEPAL (Sigma-Aldrich, I8896-50 ML) and 0.225% Tween20 (VWR). Proteinase K (ThermoFisher Scientific, EO0492) was added to a final concentration of 0.4 μg/μl before use.

Using E14Tg2a mESC, the following cell lines were generated in this work:

*Gbx2/Asb18* locus: Δ3xCTCF, 71 Kb INV, Δ3xCTCF:71 Kb INV, ΔProm Gbx2, Δ3xCTCF:ΔProm Gbx2, ΔCTCF SE Gbx2, Gbx2 INV, Gbx2 INV:Δ3xCTCF

*Six3/Six2* locus: Δ6xCTCF, 156 Kb INV, Δ6xCTCF:156 Kb INV, Six3$^{-/-}$, Δ6xCTCF:Six3$^{-/-}$, Δ4xCTCF, Δ4xCTCF:Six3$^{-/-}$, Δ6xCTCF:Six2$^{-/-}$, 226 Kb INV

**ChIP-Seq.** ~4 × 10$^7$ cells were used for RAD21 (abcam, ab154769) and ~1 × 10$^7$ cells for H3K27ac (Active Motif, 39133). Cells were crosslinked with 1% formaldehyde for 10 min at room temperature (RT) and quenched with 0.125 M glycine for 10 min. Cells were consecutively incubated with three different lysis buffers (Buffer 1: 50 mM HEPES, 140 mM NaCl, 1 mM EDTA, 10% glycerol, 0.5% NP-40, 0.25% TX-100; Buffer 2: 10 mM Tris, 200 mM NaCl, 1 mM EDTA, 0.5 mM EGTA; Buffer 3: 10 mM Tris, 100 mM NaCl, 1 mM EDTA, 0.5 mM EGTA, 0.1% Na-deoxycholate, 0.5% N-lauroylsarcosine) in order to isolate chromatin. Next, chromatin was sonicated for 8 min in Buffer 3 (20 s on, 30 s off, 25% amplitude) using an EpiShear probe sonicator (Active Motif).

The sonicated chromatin was mixed with 3 μg of H3K27ac antibody or 10 μg of Rad21 antibody and incubated overnight at 4 °C. Next, 50−100 μl of Dynabeads (Thermo Fisher, 10-002-D) were aliquoted into a microtube for each ChIP reaction. The dynabeads were washed with 1 ml of cold blocking solution (0,5% BSA and 1x PBS). The antibody bound chromatin was added to the washed beads and incubated for 4 h at 4 °C. Magnetic beads were washed five times with RIPA buffer (50 mM Hepes, 500 mM LiCl, 1 mM EDTA, 1% NP-40 and 0,7% Na-Deoxycholate). The chromatin was eluted and incubated at 65 °C overnight to reverse the crosslinking. Next, samples were treated with 0.2 mg/ml Rnase A and 0.2 mg/ml proteinase K. DNA purification was performed using Zymoclean Gel DNA Recovery kit (Zymo Research, D4008).

Regarding the computational processing of the ChIP-seq data, reads were subject to quality control and trimming of low quality regions and/or adapters using *fastqc* (https://www.bioinformatics.babraham.ac.uk/projects/fastqc/), *MultiQC*[100] and *trimmomatic*[101]. Next, reads were mapped to mm10 reference genome with Bowtie2[102]. After read mapping, only reads with a mapping quality above 10 were kept and duplicated reads were removed with SAMtools[103]. Afterwards, bigwig files were generated with *bamCoverage* from *deepTools*[104] applying the reads per genome coverage normalization.

**Capture-C.** Capture-C experiments were performed as previously described[105]. 5 × 10$^6$ cells were crosslinked with 2% formaldehyde for 10 min and quenched with 0.125 M glycine for 10 min. Cells were washed with PBS and resuspended in lysis buffer (10 mM Tris pH 8, 10 mM NaCl, 0.2% NP-40 and 1× protease inhibitors) during 20 min on ice. Following centrifugation, the pellet was resuspended in 215 μL 1×CutSmart buffer and transferred to a microcentrifuge tube. The resuspended pellet was mixed with 60 μl 10× CutSmart buffer, 393.5 μl water and 9.5 μl 20% (vol/vol) SDS (0.28% final concentration) (Invitrogen, cat. no. AM9820) followed by 1 h incubation at 37 °C while shaking on a thermomixer at 500 rpm (intermittent shaking: 30 s on/ 30 s off). Then, 20% vol/vol Triton X 100 was added at a final concentration of 1.67% vol/vol followed by another incubation at 37 °C for

1 h while shaking. Chromatin was digested by adding 25 μL NlaIII (250 U, R0125L) and incubating at 37 °C for several hours, followed by addition of another 25 μL of NlaIII and incubation at 37 °C overnight. The digested chromatin was ligated with 8 μl (240U) of T4 DNA ligase (Life Tech, cat. no. EL0013) for 18 h at 16 °C. Samples were treated with Proteinase K (3U; Thermo Fisher, cat. no. EO0491) and RNase A (7.5 mU; Roche: 1119915), and DNA was purified using the Qiagen kit (28506). After checking the quality of the digestion and subsequent ligation, chromatin was sonicated for 30 cycles (30 s on, 30 s off, 25% amplitude) using an EpiShear probe sonicator (Active Motif) and DNA samples were purified using AMPure XP SPRI beads (Beckman Coulter, cat. no. A63881). Libraries were prepared using the NEBNext Ultra II kit (New England Biolabs, cat. no. E7645S/L). Index primers set 1 and 2 from the NEBNext Multiplex Oligos for Illumina kit (New England Biolabs, E7335S/L E7500S/L) were incorporated using Herculase II Fusion Polymerase Kit (Agilent, cat. no. 600677). The resulting libraries were pooled (six libraries/pool) and a double Hybridization capture using ssDNA probes was performed following a modified version of the Roche HyperCapture streptavidin pull-down protocol (Roche, cat. no. 09075763001) described in ref. 105. Libraries were sequenced using Novaseq6000_150PE_2,25 Gb/lib (15 Mreads/lib). For each of the investigated cell lines, Capture-C experiments were performed as two independent biological replicates.

*Capsequm2* (http://apps.molbiol.ox.ac.uk/CaptureC/cgi-bin/CapSequm.cgi) was used to design the ssDNA probes (120 bp probe length). The sequences of the ssDNA probes are listed in Data S3.

## Analysis and quantification of Capture-C data

Capture C reads were subject to quality control and trimming of low quality regions and/or adapters using *fastqc* (https://www.bioinformatics.babraham.ac.uk/projects/fastqc/), *MultiQC*[100] and *trimmomatic*[101]. Next, the reads were processed with capC-MAP[106], considering the restriction enzyme *NlaIII* (cutting site CATG), the mm10 reference genome and *normalize = TRUE*. The coordinates (mm10) of the viewpoints (i.e. «targets» according to *capC-MAP* terminology) were:

*Gbx2* SE: chr1 89869929 89870764
*Six3* SE: chr17 85484699 85485038
*Asb18* TSS: chr1 90013014 90013383
*Six2* TSS: chr17 85688402 85689017

After running *capC-MAP*, the normalized pileup bedgraph files for intra-chromosomal contacts for each viewpoint were collected. According to *capC-MAP* documentation, the number of piled-up interactions per restriction fragment are normalized to reads per million, so that the sum of the number of reads associated to each viewpoint genome-wide is equal to one million. Subsequently, for the restriction sites without any detected interactions, a signal equal to 0 was assigned. In addition, only data from the regions *chr1 89228752 90664659* (*Gbx2-Asb18* locus) and *chr17 84890284 86289254* (*Six3-Six2* locus) was considered. Next, the bedgraph files of both replicates were averaged and the resulting bedgraph was converted to a bigwig with the usage of the *bedGraphToBigWig* UCSC tool[107]. Bigwig subtraction tracks were generated with the usage of *bigwigCompare* from *deepTools*[104]. For visualization purposes in the UCSC browser, the subtraction tracks were subject to smoothing (16 pixels window) in order to minimize the contribution of single restriction site fragments.

## Annotation of developmental and housekeeping genes

Developmental genes were annotated based on the presence of broad H3K27me3/PcG domains around their TSS[60,62] using a previously described strategy[61,63]. Briefly, H3K27me3 Chip-Seq fastq files from hESC (GSE24447; H3K27me3: SRR067372, input: SRR067371) and mESC (GSE89209; H3K27me3:SRR4453259, input: SRR4453262) were downloaded. Then, reads were mapped against mm10 and hg19 genomes with bowtie2 and peaks were called with MACS considering the

broad peak calling mode[108]. After peak calling, only peaks with a fold enrichment >3 and q value < 0.1 were kept. Next, peaks within 1 Kb of each other were merged using *bedtools*, and associated with a protein coding gene when overlapping a TSS. Subsequently, the knee of the peak size distribution was evaluated with *findiplist()* (inflection R package; https://cran.r-project.org/web/packages/inflection/vignettes/inflection.html). Upon curvature analyses, genes associated with a H3K27me3 peak>7Kb were defined as developmental genes (Mouse: n = 967; Human: n = 1045).

For the annotation of housekeeping genes, one list for humans and one for mice (Mouse: n = 3277; Human: n = 2176) were obtained from the *Housekeeping and Reference Transcript Atlas* database[109].

For the analyses presented in Fig. S5, developmental genes were defined as those associated with the Gene Ontology (GO) term developmental process; (GO:0032502) using AmiGO[110]. These developmental genes were further sub-divided in two groups based on the presence or absence of broad H3K27me3/PcG domains around their TSS as described above.

## In silico analysis of TAD gene density

TAD maps previously generated in 11 mouse and 37 human cell types were considered. To ensure consistency in TAD calling, the genomic coordinates of all TADs and their boundaries were retrieved from the 3D Genome Browser database[111], in which TAD boundaries were consistently called using the directionality index approach initially established by Dixon et al.[13]. Regarding human TAD maps, the 37 hg19 TAD maps available in the 3D Genome browser were used[111]. With respect to mice, 11 TAD maps were used: (i) eight mm10 TAD maps available in the 3D Genome browser[111], (ii) three mm9 TAD maps (CH12_Lieberman-raw_TADs.txt, cortex_Dixon2012-raw_TADs.txt and mESC_Dixon2012-raw_TADs.txt), also available in 3D genome browser and that were liftover to mm10 using the UCSC liftover tool[112]. For the eight mm10 TAD maps from the 3D Genome browser, with the exception of the files G1E-ER4.rep1-raw.domains and G1E-ER4.rep2-raw.domains, the prefix chr was added to the chromosome name, chr23 was relabeled to chrX and chr24 to chrY.

For each TAD, gene density was computed as the number of TSSs (based on hg19 and mm10 RefSeq curated annotations downloaded through the UCSC Table browser[113]) located within each TAD divided by the length of the TAD. Subsequently, for each TAD map, the TADs were sorted based on increasing gene densities, and three groups of TADs of equal size were considered: low density (LD), medium density (MD) and high density (HD) TADs. Next, for each TAD map, we computed whether developmental genes were over represented among the genes found within LD TADs using a Fisher test. Lastly, Gene Ontology functional enrichment analyses were performed for two different TAD maps (*hESC Dixon* and *mESC Dixon*, available in 3D genome browser) using the *WebGestalt* R package[114] and considering the genes located in the three different groups of TADs (LD, MD and HD) described above. The WebGestalt functional enrichment analysis were performed using the ORA (over representation analyses) method and all genes as the reference gene list (group of genes used to compare and compute enrichments).

## In silico analysis of gene distribution within TADs

When computing gene distribution within TADs, the TSSs of genes were taken as a reference. TSS coordinates were obtained from RefSeq curated annotation (downloaded through the UCSC Table browser[113]) for both human and mice. Gene distribution within TADs was computed in multiple TAD maps independently (37 TAD maps in humans and 11 in mice, see previous methods section).

Each TAD was divided in ten bins of equal sizes. Therefore, the size of each bin is 10% of the size of its corresponding TAD. Regarding the labeling of the bins, when moving from the boundaries of the TAD (TAD start or end coordinates) towards its interior, the first bin

was labeled as bin 1, the next one as bin 2 and so on, until reaching bin 5, which is the closest to the center of the TAD (Fig. 1B). The genomic regions located outside TADs (inter-TAD) were assigned to bin 0 (Fig. 1B). For each of the considered TAD maps, the TSSs of genes were assigned to bins attending to their location in the genome. Furthermore, for the analyses presented in Fig. S4, the distance between each TSS located in bin 1 and its closest TAD boundary was computed.

### Global analysis of insulation strength and CTCF binding at TAD boundaries

The CTCF data used in these analyses were obtained from mESC (GSE36027, CTCF replicates: SRR489719 and SRR489720, input replicates: SRR489731 and SRR489732) and hESCs (GSE116862, CTCF replicates: SRR7506641 and SRR7506642, input replicates: SRR7506652 and SRR7506653). After downloading the corresponding fastq files, reads from both replicates were merged and subject to quality control and trimming of low quality regions and/or adapters using *fastqc*, *MultiQC*[100] and *trimmomatic*[101]. Next, reads were mapped to either mm10 or hg19 genomes with *Bowtie2*[102]. After read mapping, only reads with a mapping quality above 10 were kept and duplicated reads were removed with *SAMtools*[103]. Afterwards, CTCF bigwig files were generated with *bamCoverage* from *deepTools* applying the reads per genome coverage normalization[104]. CTCF peaks were called with MACS2[108] and only those peaks with a fold change > 4 and q value < 0.01 were considered. Lastly, using the coordinates of the CTCF peaks and the CTCF bigwig files, the maximum intensity of each CTCF peak was calculated using the *bigWigAverageOverBed* UCSC binary tool.

The insulation score and boundary strength datasets used for our analyses were obtained from different sources and through various procedures. Regarding mESC data, a bigwig file (file ID: 4DNFIMVJ2YV3) with mm10 insulation scores obtained from previously generated Hi-C data[19] was downloaded from the 4D Nucleome data portal[115]. In addition, a bed file (file id: 4DNFI1S7FI1U) with boundary strength values as computed by *cooltools*[116] for the bins mapped as TAD boundaries was also downloaded from the 4D Nucleome data portal[115]. Regarding hESC data, *.hic* files from two replicates (GSM3262956 and GSM3262957[20]) were downloaded from GEO. Next, *.hic* files were converted to *.cool* format with the *hic2cool* software (https://github.com/4dn-dcic/hic2cool) considering the 10 Kb contact resolution matrix. Afterwards both replicates were merged with *cooler merge* and normalised with *cooler balance*[117]. Subsequently, insulation scores were computed with *cooltools* (based on the diamond insulation score technique) using a window size of 100 Kb. In addition a bigwig file was also created in this step with *cooltools* by specifying the –*bigwig* option. Moreover, boundary strength metrics were also computed by *cooltools* for the bins considered as TAD boundaries. In order to make these boundary strength values more comparable to those obtained from the 4D Nucleome data portal, only TAD boundaries with a strength larger than 0.2 were considered, while a strength value = 0 was assigned to the remaining boundaries

Once ready, the insulation score, boundary strength and CTCF datasets were used to compute several metrics for TAD boundaries in both mice and humans. TAD boundaries were defined using the start and end coordinates of TADs previously identified in mESC[19] and hESC[118]. Next, each TAD boundary was expanded by ±50 Kb and the following metrics were calculated within the resulting 100 Kb window: (i) the number of overlapping CTCF peaks, (ii) the CTCF aggregated signal (sum of the CTCF bigwig signal for all the peaks overlapping with the 100 kb window), (iii) the insulation score (minimum insulation score value in the 100 kb window computed with *bigWigAverageOverBed*) and (iv) the boundary strength (maximum boundary strength value computed by *cooltools* for any bin in the 100 kb

window). In addition, TAD boundaries were classified as Developmental or Other: Developmental boundaries were defined as those associated with a developmental gene located in bin 1 (Fig. 1 1B); Other included all the remaining boundaries. Moreover, a set of 5000 random TAD boundaries was generated by randomly selecting 5000 regions in the mouse and human genomes.

The chicken and zebrafish CTCF ChIP-seq profiles shown in Fig. S9 were previously generated in HH18 chicken embryos (GSM5835469)[119] and 48 hpf zebrafish embryos (GSM5344494)[120], respectively.

### In silico analysis of the distribution of CTCF sites and CTCF motif orientation around genes located close to TAD boundaries

Gene located close to TAD boundaries were defined as those assigned to bin 1 (Fig. 1B) according to TAD maps previously generated in mESC[19] and hESC[118]. Then, for each bin 1 gene, the number of CTCF peaks (obtained as described in the previous section) located within a ±100 Kb window around its TSS was calculated. The 100 Kb window extending from the gene TSS towards the TAD boundary was defined as the outer window, while the 100 Kb window extending from the gene TSS towards the center of the TAD was defined as the inner window (Fig. 2C). After calculating the number of CTCF peaks in the inner and outer windows, we computed the $\Delta CTCFpeaks@Bdry$ metric for each gene as the number of CTCF peaks located in the inner window minus the number of CTCF peaks located in the outer window. In addition, the CTCF bigwig signals associated with the CTCF peaks were used to calculate the $\Delta CTCFsignal@Bdry$ metric for each gene as the aggregated signal for all the CTCF peaks located in the inner window minus the aggregated signal for all the CTCF peaks located in the outer window.

In addition, for the CTCF peaks located in the inner and outer windows of bin 1 genes described above, their motif orientation was obtained using the *CTCF package* (https://github.com/dozmorovlab/CTCF). Briefly, JASPAR 2022 CTCF motif predictions for CTCF motif MA0139.1 were considered: (i) mm10.JASPAR2022_CORE_vertebrates_non_redundant_v2 data for mouse analyses, and (ii) hg19.JASPAR2022_CORE_vertebrates_non_redundant_v2 data for human analyses. First, the CTCF peaks were intersected with the motif coordinates (as provided by the *CTCF package*) using bedtools. Next, for each CTCF peak, the orientation associated with the overlapping CTCF motif showing the lowest q-value was defined as the CTCF peak orientation. Finally, taking into account the position of the center of the TADs, the CTCF peaks orientation was defined as +1 when oriented towards the center of the TAD (inward site) or as −1 when oriented away from the center of the TAD (outward site). To compare the distribution of inward and outward CTCF sites between the inner and outer window of the different bin1 gene categories (i.e. All, Developmental and Housekeeping), Chi-squared tests were calculated. The Cramér's V effect size estimator[121] was calculated with *rcompanion*.

### Analysis of RNA Pol2 Pausing Index (PI)

The pausing index (PI) was calculated as the ratio of the average PRO-Seq signal (from mESC[69]) or GRO-Seq (from hiPSC[70]) between the gene promoter (from 50 bp upstream of the TSS to 250 bp downstream of the TSS) and the gene body (from 250 bp downstream of the TSS to the Transcription Termination Site (TTS)). TSS and TTS coordinates were obtained from RefSeq curated annotation, as described above.

For mouse analyses, strand-specific PRO-Seq signal bedgraph files from mESC were retrieved from GEO (GSE178230). The downloaded files were GSE178230_mm10_mESC_PROseq_mESC_N4_F.bedGraph and GSE178230_mm10_mESC_PROseq_mESC_N4_R.bedGraph. Bedgraph files were converted to bigwigs with the *bedGraphToBigWig* tool. Next, for each transcript, the average bigwig signal for the promoter and gene body regions were obtained with the *bigWigAverageOverBed* tool and used for the PI calculation. For each transcript, only the PRO-Seq data of its corresponding strand was considered.

For human analyses, GRO-Seq bedgraph files from induced pluripotent stem cells (iPSCs), derived of two different donors, were retrieved from GEO (GSE117086). The downloaded files were: GSM3271001_iPSC_11104_A.bedGraph and GSM3271002_iPSC_11104_B.bedGraph. Next, the two bedgraph files were combined and converted to a single bigwig with the usage of the *bedGraphToBigWig* and *bigWigMerge* UCSC binary tools. Subsequently, for each transcript, the average bigwig signals for the promoter and gene body regions were obtained with *bigWigAverageOverBed* tool and used for the PI calculation.

After PIs calculation, transcripts with not finite PIs were discarded with the usage of is.finite() R function. When filtering genes by gene expression levels, those with expression levels above 5 FPKM for mESC[58] or 5 RPKM for hESC (GSE24447, file GSM602289_ESC_RPKM.txt) were considered to be active. When selecting genes based on their proximity to TAD boundaries, only transcripts whose TSS were located within bin 1 (Fig. 1B) according to mESC or hESC TAD maps (see previous Method sections) were considered. In all the comparisons, for genes with multiple transcripts, the highest pausing index computed for all the transcripts was kept as a reference.

### Reporting summary
Further information on research design is available in the Nature Portfolio Reporting Summary linked to this article.

## Data availability
All the generated ChIP-seq and Capture-C data are publically available through GEO under accession codes: GSE252218, GSE252080. The following publically available and previously generated datasets were used in this study: GSE24447, GSE89209, GSE36027, GSE116862, GSE178230, GSE117086. Source data are provided with this paper.

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

## Acknowledgements

We would like to thank all the Rada-Iglesias lab members for insightful comments and suggestions. Work in the Rada-Iglesias laboratory is supported by grant PID2021-123030NB-I00 (AR-I) funded by MCIN/AEI/10.13039/501100011033 and by ERDF A way of making Europe, grant RED2022-134100-T (AR-I) (REDEVNEURAL 3.0) funded by MCIN/AEI/10.13039/501100011033, grant ERC CoG PoisedLogic (862022) (A.R.-I.) funded by the European Research Council, grant ENHPATHY H2020-MSCA-ITN-2019-860002 (A.R.-I.) funded by the European Commission, grant Chrom_rare HORIZON-MSCA-2021-DN-01-101073334 (A.R.-I.) funded by the European Commission. E.M.B was granted with a fellowship from the network PIE-202120E047-Conexiones-Life.

## Author contributions

T.E. performed most of the experimental studies. V.S-G. carried out the computational analysis and contributed to the conceptual design of the project. P.R., M.M-S. and E.M-B. participated in the generation some of

the transgenic mESC lines. E.H. contributed to the design of the project and supervised the experimental work. A.R-I. supervised the work. T.E., V.S-G. and A.R.-I. wrote the manuscript and prepared the figures.

## Competing interests

The authors declare no competing interests.

## Inclusion and ethics

The authors have adhered to the inclusion & ethics standards of the Nature portfolio journals.
