## [Peer Review File · Nature Communications]

Cooperative insulation of regulatory domains by CTCF-dependent physical insulation and promoter competitionREVIEWER COMMENTS

Reviewer #1 (Remarks to the Author):

Mammalian enhancers and their target promoters are grouped together in TADs, but the question why has remained difficult to unambiguously settle. In this study, Ealo and colleagues address a part of this question, by focusing on developmental genes that are enriched at TAD boundaries where multiple CTCF peaks cluster together. After a genome-wide characterization, they focus on 2 gene loci where a TAD boundary separates nearby developmentally-regulated genes. They combine a large number of genome-edited cell lines with the analysis of transcription changes and chromosome conformation to identify an intriguing mechanism that combines insulation and promoter competition.

Although this study does not provide a final answer to the above question, it adds a valuable contribution to the ongoing discussion. Some open ends about key observations and their explanations remain to be explored (see below), but otherwise I support the publication of this manuscript.

Major comment.

1. The major question that I had after reading this manuscript, and that remains mostly undiscussed, is why developmental genes often localize next to TAD boundaries. At both loci that were studied, *Gbx2-Asb18* and *Six3-Six2*, the issue of enhancer competition would be resolved if one or both genes in the pairs would localize away from the boundary. Yet, the linking of these gene pairs, and possibly the array of CTCF binding sites in-between, appears strictly conserved. Could it be that the presence of CTCF binding sites / TAD boundary is an integral part of the regulation at these loci? Instead of (only) acting as boundaries, could these arrays (also) bring the promoters of *Gbx2* and *Six3* in contact with the more distal enhancers (e.g. Vian et al, 2018 Cell and Kubo et al, 2021 NSMB)? This possibility is mentioned in passing (“...the reduced expression of *Six3* can not be simply attributed to a potential role of the CTCF sites as facilitators of enhancer-gene communication”) but not further explored. It’s noticeable that the SEs at both loci are close to CTCF binding sites. Whereas I believe that promoter competition is involved (see below), I’m under the impression that the incorporation of the CTCF arrays as both permeable blockers (for *Asb18* and *Six2*) and as loop facilitators (*Gbx2* and *Six3*) helps to better explain observations like those in Fig. 5C.

A number of questions arise:

- a. Could the authors indicate the orientation of the CTCF peaks, both in the arrays at the boundary and near the SEs? Are these peaks convergently oriented? Could the arrays also facilitate loops between genes and enhancers on the other side of the boundary?
- b. Are the CTCF binding sites near the SEs included in the inversions (Fig. 3B, Fig. 5B)? See my comment about scales and annotations of the figures.
- c. The blocking and facilitator function can be separated by removing the CTCF binding sites at the SEs, followed by gene expression and CapC experiments. Although I’m conscious of the additional work this requires, adding this result (at least for one domain) will considerably increase the impact of the study.
- d. Could the authors confirm the evolutionary conservation of the CTCF arrays at both the *Gbx2-Asb18* and *Six3-Six2* domains? Addition of a figure panel with CTCF binding in different vertebrate species would be insightful. CTCF binding has been published for many vertebrate species, at least up to zebrafish (e.g. Kaaij et al, 2018 Cell Rep).

2. I consider the promoter competition aspect convincing, but the mechanism remains unclear. Whereas the competition for rate-limiting factors may be possible, it seems unlikely to me that a single promoter can exert such an effect. The authors state “... as this does not require the placement of developmental genes in between enhancers and non-target genes (Fig. 3 and 5) 36,37”, but these cited studies are from *Drosophila*, where loop extrusion and blocking by insulators appears not so important for enhancer promoter looping. Whereas the *Six3* deletion alone has a minor impact on *Six2*-SE contacts, the combined deletion of *Six3* and the CTCF array quite strongly increases the contacts (and *Six2* activity). Recent work, both cited and non-cited, has confirmed that TSS can create strong boundaries, particularly when enriched for paused PolII (Zhang D et al, 2020 Nat Genet; Barshad et al, 2023 Nat Genet; Zhang S et al, Nat Genet 2023).

Question:

Could the authors analyze GRO-seq/PRO-seq/... data to determine if the Gbx2 and Six3 genes have a strong pausing index? Would it be worth it to include this analysis for the genome-wide characterization of paused genes at boundaries (Fig. 1)?

Minor issues:

- Introduction (page 3 in Word document): "Nevertheless, the binding of architectural proteins can only explain the formation of a fraction of TAD boundaries, suggesting that alternative mechanisms should exist 8,10,11". This may be true in Drosophila cells, but in mammalian cells over 80% of TAD boundaries are lost upon CTCF depletion (Nora et al, 2017 Cell), with the remainder possibly explained by PolII-mediated boundaries (see also above).
- Introduction (page 4): citations 8,20-24: the authors should include Barshad et al, 2023 Nat Genet. Zhang S et al, 2021 Sci Adv could be replaced by Zhang S et al, 2023 Nat Genet.
- Introduction (page 4): citations 40-41: the authors should add Allayar et al, 2018 Nat Genet.
- Introduction (page 4): "Promoter competition occurs when the preferred gene gets specifically activated regardless of its relative position with respect to the shared enhancer/s and neighbouring gene/s 36". Leading insights into promoter competition, in mammalian cells, come from imprinted domains and particularly the Igf2-H19 domain. The authors could cite Barlow, 1997 Embo J.
- Figure 1: I am surprised by the difference for bin 0 and 1 between panels C and F. Could this be due to how the boundaries were called? The manuscript provides very limited information about this. Depending on the strategy (directionality index, insulation score, arrowhead algorithm), the span of inter-TAD domains can considerably differ. It would be useful if the strategy could be harmonized.
- Results (page 12) and Fig. 3B: clonal lines (i) and (ii) are inversed in the figure panel as compared to the text.
- Figure 3A, 3B and all similar panels: the interpretation of these panels is complicated by a lack of genomic coordinates and consistent scaling. Fig. 3A: what is the size of this region? Fig. 3B: these panels will be improved if they are drawn to the correct scale. I particularly have problems understanding how the configurations in Fig. 5B and 5E compare. The interpretation could be further facilitated if the span of panel 3B is added to panel 3A (and similar for other panels).
- Figure 3C and other RT-qPCR panels: I'm surprised to see that the WT data, which should be normalized to itself, is showing variation between replicates. How was this normalization performed? In a conventional $\Delta\Delta$ CT analysis, the normalization is performed for each PCR individually, thereby setting the WT value consistently to 1 for each experiment (or 0 after log transformation).
- Figure 3C and other RT-qPCR panels: why did the authors include the two-fold cut-off for significance? This seems arbitrary.
- Figure 6: Panel 6B is a repeat of panel 5C and could be removed. Fig. 5H and 6C are partially redundant with Fig. 5C and 5D, and may thus be reorganized as well.

Reviewer #2 (Remarks to the Author):

Ealo et al. provide evidence that the robust insulation of regulatory domains occurs through the synergistic activities of CTCF insulation and promoter competition. Using genome-wide analysis of publicly available Hi-C and CTCF ChIP-seq data from mouse and human embryonic stem cells (ESCs), they found that developmental genes tend to exist in gene poor TADs and that they are generally found near CTCF clusters at TAD boundaries.

Next, the authors focused on two specific loci: 1) the developmental gene Gbx2 and its upstream enhancer, separated by a CTCF cluster from the tissue-specific Abs18 gene in the neighboring TAD; and 2) the Six3 and Six2 genes which are separated by a strong and conserved TAD boundary and displaying largely non-overlapping expression patterns during embryogenesis. Through the generation of multiple mESC homozygous cell lines they investigate the functional relevance of the sequential positioning of developmental genes, CTCF clusters and TAD boundaries. Their main conclusions are that the CTCF cluster and the promoter of the developmental gene itself act together to insulate the gene in the neighboring TAD. Furthermore, they suggest that this insulation by the promoter is due to competition for the enhancer, rather than physical insulation. This conclusion is supported by the finding that deletion of the developmental gene promoter had

only minor effects on the contact frequency of the neighboring gene and the enhancer in the developmental TAD, but robust increases in gene expression.

Overall, this is a very well executed study that addresses mechanisms of chromatin boundary formation. One of the interesting results is that the experiments make the distinction between enhancer/promoter competition vs boundary activity of a strong promoter. The molecular dissection of two boundaries is very well done, thoroughly controlled, and described in a clear and logical manner. An issue for me is that at the very end a new concept is introduced, i.e. that of silencer action at a distance, without providing any experimental evidence, and with no attempt to integrate this separate concept from the beginning.

The list of comments is long but a lot of it can be addressed textually.

Major points

- 1) The generalizing computational binary division of housekeeping vs developmental genes and their differential enrichment at various distances from boundaries is not entirely convincing because the differences between the two groups are quite modest or absent depending on data sets. Also, what about the converse question: are genes near TAD boundaries more enriched for developmental genes?
- 2) A possible limitation of the premise of this paper is that it remains an open question whether the binning of genes within TADs still applies when considering that TADs vary between tissues. In other words, developmental genes might distribute differently if the TAD analysis was done in the tissue in which they are actually expressed.
- 3) In Figs. 1 c and f, housekeeping and developmental genes seems to behave differently when it comes to boundary proximity (bins 0 and 1). This makes me wonder about the significance of this categorization. Moreover, the Y-axes do not start at zero giving the impression of a larger effect size.
- 4) The distinction between developmental genes and tissue-specific genes is too vague. Developmental genes can also be tissue-specific.
- 5) Does promoter competition occur more frequently in genes near TAD boundaries? In other words, is this phenomenon of promoter competition more related to simply any genes near TAD boundaries, or is there something specific about developmental genes being involved in the promoter competition?
- 6) Related to point 5: I don't understand this idea: "We hypothesize that, by being close to boundaries, developmental genes are more likely to be located at shorter linear distances from their cognate enhancers than the potential competing genes located at the other side of the boundaries." Please clarify.
- 7) Is there any role for the orientation of CTCF sites? The authors show that developmental boundaries have more CTCF peaks and CTCF ChIP-seq aggregated signal, but the insulation scores and boundary strengths (i.e. physical insulation) are similar to "other boundaries". Is this because of any distinct features of the directions of these CTCF binding sites near "developmental" boundaries than "non-developmental" boundaries? I suggest labeling the orientation of the CTCF sites under the CTCF tracks (Fig. 3-6, S1, S2, S5, S6, S8, S10, S11, etc.).
- 8) Clonal variation: In Fig. 3C, 71Kb INV seems to decrease the expression of Gbx2 in three clones, whereas the expression of Gbx2 increased in two clones, and was unchanged in four clones. What is the basis for this variability? Which clone was used in Fig. 3E? What were the fold changes of the expression of Gbx2 and Asb18? Would the conclusion of Fig. 3E be different in clones with different Gbx2 expression level?
- 9) All gene expression changes are shown as fold-change. To better understand the competition model absolute expression levels would be helpful to know. For example, is Six3 expressed highly enough to be able to compete?
- 10) It would be nice to know whether the Six2/Six3 TAD boundary is as strong in NPGs.
- 11) Fig.5H: can the incomplete rescue of Six3 expression by the Six2 deletion be explained that the Six3 enhancer acts on other genes in the Six2 TAD? What is the level of Srbd1?
- 12) Line 659: Pls make this statement more precise: "Six3 preferentially contributes to regulatory insulation through promoter competition, while structural mechanisms, such as enhancer blocking, seem to have a comparably smaller, albeit non-negligible, contribution." To me it seems they have synergistic effects. Δ CTCF upregulates Six2 ~8-fold but when combined with Δ Six3 ~64-fold.
- 13) A small point: comparing Fig.6C and Fig.6G: The Δ Six3 (green data points) should be identical but are not. Where these clones re-analyzed separately?
- 14) Line 709: At the very end the authors propose an all new concept: spreading of repressive chromatin or silencers from Six3 towards Six2 in Δ 4xCTCF cells, but apparently not in Δ 6xCTCF.

Why? This is a major problem because it really confounds the story. The authors need to test this idea experimentally. What is the evidence that there is a silencer element in the 1kb Six3 region that was deleted? This is not even considered in the Discussion.

15) Related to the previous point: Line 85: "compartmental domains" and "topologically associating domains" (TADs) can be overlapping but are not the same. This needs to be considered especially when invoking the spreading of repressive chromatin or long-range repression.

16) This also means that "developmental genes" need to be sub-categorized. developmental genes were defined as those containing broad polycomb domains. While the reason for this and references are provided in the methods, perhaps a short sentence explaining their reasoning (with references) in line 175-176 might help. Regardless, this definition may cause bias in the analyses since findings may reflect the roles of PcG domains in regulatory insulation near TAD boundaries rather than developmental gene promoter competition. The authors should consider comparing the insulation of genes near TAD boundaries: a) developmental genes with polycomb domains; b) developmental genes without polycomb domains; c) non-developmental/non-lineage specific genes (if any) with polycomb domains; d) non-developmental/non-lineage specific genes without polycomb domains.

17) Can the author discuss/speculate how promoter competition works across a CTCF boundary?

Additional points

There are a number of citation- and subject definition-related issues that need to be addressed (points 1-6).

1) Line 69: Insulators can be divided into enhancer blocking insulators and boundary elements. I would not necessarily equate the two, especially in light of the invocation of silencer spreading at the end.

2) It is important to textually clarify at the outset that in some of the experiments the authors use the term "insulation" in the context of 4C contacts, not in terms of function.

3) Line 94: "Nevertheless, the binding of architectural proteins can only explain the formation of a fraction of TAD boundaries". Again, this depends on the definition. Loop domains are lost upon cohesin depletion, but compartments remain intact and are even strengthened.

4) Line 96 and subsequent: "many architectural protein independent boundaries are located proximal to active transcription start sites (TSS) and housekeeping genes". I suggest not mixing citations from flies and mammals because some of the concepts don't translate well between organisms.

5) Line 97: "Recent reports indicate that transcription and/or active promoters can participate in the formation of contact domains and physical boundaries". Several studies also show that transcription has minimal impact on architecture (e.g. PMID: PMC7331254 and PMID: PMC8397779). Ref 21 could also be explained by enhancer-promoter competition.

6) Same for line 138: There are other studies that use acute pol2 degradation or inhibition with little or no effect on architecture. These should be considered for a balanced presentation of the literature.

7) Line 125: "It has been previously suggested that the competition between promoters for a shared enhancer might involve mutually exclusive enhancer-promoter contacts ("flip-flop" model). However, more recent observations support non-structural mechanisms whereby promoters and enhancers share transcriptional hubs/condensates". I suggest not phrasing these as mutually exclusive scenarios. The cited papers simply suggest that both mechanisms can work but at different loci.

8) There is a typo in line 119, remove "no".

9) Line 167 – please define CGI clusters (write out in full before using the abbreviation).

10) Figure 1A: even if non-significant (since this will be reflected in the size of the bubbles anyway), show the results of the enrichment analysis of each GO term in all three populations.

11) In Fig. 3E, it is not clear if the Capture-C maps with 71Kb INV displayed on reference genome track (non-inverted), or displayed as actual genome track (71Kb inverted)? Why is the enhancer still strongly interacting with the upstream regions outside the inverted regions even when the viewpoint is distal to it after inversion? In addition, is the increased interaction between the enhancer and the 3xCTCF sites in 71Kb INV is because the enhancer is now closer to the 3xCTCF sites? If the 71Kb INV is displayed on reference genome track (non-inverted), it might be better to show it on the actual genome track (71Kb inverted) as well to avoid confusion.

12) Line 524 – PcG has not been defined before this point – write it out in full.

Reviewer #3 (Remarks to the Author):

In this manuscript, Ealo et al, investigate the synergies between CTCF clusters and developmental genes in topological insulation and gene expression. They initially used published HiC data from various human and mouse cell lines to determine unique characteristics of TADs containing developmental genes compared to TADs with housekeeping or other genes, with a focus on the relative position to TAD boundaries, the strength and number of CTCF sites around the genes of interest and the boundary insulation strength.

Then, they perform a number of elegant genetic perturbation experiments on two specific developmental loci to uncover distinct mechanisms of how genes function synergistically with CTCF clusters to ensure functional insulation from neighboring genes. Specifically, they provide evidence for two distinct mechanism, one supporting enhancer-blocking and another showing promoter competition. Although, both mechanisms have been proposed and illustrated in previous studies, the value and quality of this study is still very high.

Overall, this is an elegant and well-controlled study, with many interesting findings which support previous hypothesized or illustrated mechanisms of gene regulation and functional insulation. Although the perturbation studies are quite robust, I found the first two figures not very relevant for the study and rather weak both regarding the depth of analysis and the significance of the conclusions. Specific comments:

1. One of the author conclusions is that developmental genes have CTCF clusters in between them and the boundaries, while other genes are flanked on both directions. First, the sequential orientation of boundary∅CTCF cluster∅gene is fully unsurprising. However, the selective “flanking” of housekeeping genes by CTCF sites is puzzling. Does this mean that these genes are part of “nested” TADs, while developmental genes are not? What is the orientation of the CTCF sites towards the center of TAD relative to the boundary cluster? What are the relative insulation scores of the CTCF sites on either direction?

2. The authors also mention that genes that have been previously shown to bypass boundaries are actually flanked by CTCF sites. Not sure, how is this observation relevant to the story. It seems - based on CTCF peaks- that both developmental loci (gene and SE) that they focus on, are also flanked by CTCF.

3. A better justification of why they selected these particular example loci would be very helpful. Some discussion of their special features and how could be used to ultimately predict other loci in the genome with either enhancer blocking or promoter competition would be very useful.

4. As mentioned above, the genetic experiments are very elegant and interesting. Their results indeed support that Gbx2 contributes to the functional insulation through enhancer blocking. Do they authors believe that the transcriptional orientation of the gene, transcribing away of the boundary and towards the SE contribute to phenotype, potentially by counteracting extrusion? Inversion of the gene -especially in the context of CTCF deletions- should directly examine their insulating ability regardless transcriptional orientation.

5. A better representation and higher resolution of the Capture data would be helpful, especially for the inversion experiments. Did they reconstruct the genome for reference? What is the new strong peak downstream of the SE peak represents in Figure 3E?

6. Overall, both for the Gbx2 and the Six2/3 experiments, more clear statements and quantitation of all specific interactions between SE-promoter, promoter-promoter, promoter-boundary, SE-boundary etc, for each locus and perturbation would be very useful to better appreciate the conformational changes that associate and further support the enhancer-blocking and promoter competition models.

7. Many figure panels need better annotation. For example, in Figure 1D, it is unclear of the numbers shown on the right represent distinct TADs or cell types. Overall, the numbers of TADs and genes interrogated in each analysis should be clearly stated.

REVIEWER COMMENTS

Reviewer #1 (Remarks to the Author):

Mammalian enhancers and their target promoters are grouped together in TADs, but the question why has remained difficult to unambiguously settle. In this study, Ealo and colleagues address a part of this question, by focusing on developmental genes that are enriched at TAD boundaries where multiple CTCF peaks cluster together. After a genome-wide characterization, they focus on 2 gene loci where a TAD boundary separates nearby developmentally-regulated genes. They combine a large number of genome-edited cell lines with the analysis of transcription changes and chromosome conformation to identify an intriguing mechanism that combines insulation and promoter competition.

Although this study does not provide a final answer to the above question, it adds a valuable contribution to the ongoing discussion. Some open ends about key observations and their explanations remain to be explored (see below), but otherwise I support the publication of this manuscript.

We would like to thank the reviewer for his/her constructive and useful suggestions, which we address below in more detail.

Major comment.

1. The major question that I had after reading this manuscript, and that remains mostly undiscussed, is why developmental genes often localize next to TAD boundaries. At both loci that were studied, Gbx2-Asb18 and Six3-Six2, the issue of enhancer competition would be resolved if one or both genes in the pairs would localize away from the boundary. Yet, the linking of these gene pairs, and possibly the array of CTCF binding sites in-between, appears strictly conserved. Could it be that the presence of CTCF binding sites / TAD boundary is an integral part of the regulation at these loci? Instead of (only) acting as boundaries, could these arrays (also) bring the promoters of Gbx2 and Six3 in contact with the more distal enhancers (e.g. Vian et al, 2018 Cell and Kubo et al, 2021 NSMB)? This possibility is mentioned in passing (“...,the reduced expression of Six3 can not be simply attributed to a potential role of the CTCF sites as facilitators of enhancer-gene communication”) but not further explored. It’s noticeable that the SEs at both loci are close to CTCF binding sites. Whereas I believe that promoter competition is involved (see below), I’m under the impression that the incorporation of the CTCF arrays as both permeable blockers (for Asb18 and Six2) and as loop facilitators (Gbx2 and Six3) helps to better explain observations like those in Fig. 5C.

We fully agree with the reviewer in that, based on previous reports, the localization of developmental genes near TAD boundaries/CTCF arrays could facilitate enhancer-gene loops. Therefore, following the reviewer’s suggestions, we have included a new paragraph in the introduction in which we describe the role of CTCF arrays as facilitators of enhancer-gene communication and provide the references indicated by the reviewer together with a few additional ones (Vian, L. et al., 2018; Kubo, N. et al., 2021; Paliou, C. et al.; 2019; Long, H. K. et al., 2020; Rinzema, N. J. et al., 2022). In fact, the role of CTCF arrays as loop facilitators was one of our main hypotheses as we first observed the preferential location of developmental genes near TAD boundaries and started this project. Accordingly, the different genomic re-arrangements (e.g.

deletion of CTCF arrays, inversions moving genes away from CTCF arrays and towards the TAD center) that we engineered were designed to address both the competition and facilitator models. However, when considered together, our results do not support a major role for the CTCF arrays in facilitating the expression of *Gbx2* or *Six3*. Briefly, for *Gbx2* neither the deletion of the CTCF array nor the 71Kb inversion that moves *Gbx2* away from the boundary seem to significantly affect *Gbx2* expression (Fig. 3). Similarly, for *Six3* the 156 and 226 Kb inversions that move *Six3* away from the TAD boundary do not affect *Six3* induction in NPC (Fig. 5C,F). Furthermore, although the deletion of the CTCF array impairs *Six3* induction (Fig 5C), this is rescued (albeit partly) by deleting *Six2* (Fig 5h). Overall, these results suggest that, rather than facilitating *Six3* induction in NPC, the main role of the CTCF array is to protect *Six3* from promoter competition by *Six2* and, potentially, from silencers located within the *Six2* TAD. Regarding the CTCF sites located close to both SEs, we would like to emphasize that, as already mentioned in the manuscript (Page 15), we previously deleted the CTCF site located close to the *Six3* SE and showed that it is not required for proper *Six3* induction in NPC (see Extended Data Fig 9 in Pachano, T. et al., 2021). Similarly, the 226 Kb inversion, which places *Six3* far from both the CTCF arrays and the SE CTCF site, does not affect *Six3* induction. On the other hand, and followings the reviewer's suggestion (see below), we have generated ESC lines in which we deleted the CTCF site located close to the *Gbx2* SE (Fig. S11A-C). The analyses of these new ESC lines showed that the deletion of the SE CTCF site lead to a small (<2-fold) and non-significant decrease in *Gbx2* expression (Fig. S11D). Overall, these results suggest that the CTCF sites located close to the SE do not play a major role in the control of *Six3* and *Gbx2* expression in NPC and ESC, respectively.

So, although we agree with the reviewer in that one could have expected that the CTCF arrays facilitate enhancer-gene loops and, thus the expression of *Gbx2* and *Six3*, our data does not support that model at least in the investigated loci and cellular contexts. However, as mentioned in the “*Discussion*” section, CTCF arrays might be particularly relevant as facilitators of “*ultra long-range enhancer-gene contacts (e.g. Shh, Sox9) (Paliu, C. et al., 2019; Chen, L.-F. et al., 2023)*”, while being dispensable for enhancer-gene communication across shorter distances (such as those separating *Gbx2* and *Six3* from their SEs in ESC and NPC, respectively). In this regard, and as now mentioned in the updated “*Discussion*”, “*although proper expression of Gbx2 and Six3 in ESC and NPC, respectively, does not seem to require the nearby CTCF arrays, it is certainly possible that these CTCF sites facilitate the communication of Gbx2 and/or Six3 with more distal enhancers in other cellular contexts*”.

A number of questions arise:

a. Could the authors indicate the orientation of the CTCF peaks, both in the arrays at the boundary and near the SEs? Are these peaks convergently oriented? Could the arrays also facilitate loops between genes and enhancers on the other side of the boundary?

Following the reviewer's advice, the orientation of the CTCF peaks in the arrays at the boundary and near the SEs have been added for both the *Gbx2/Asb18* (Fig 3) and *Six3/Six2* (Fig 5) loci. For *Gbx2*, but not for *Six3*, the peak at the SEs is convergently oriented with respect to peaks present at the arrays/boundary. However, as stated above, the deletion of either the SE-associated CTCF peaks or the CTCF arrays does not seem to have major effects on the expression of either *Gbx2* or *Six3* within the investigated cellular contexts. As the reviewer suggests, the arrays could theoretically facilitate loop formation between genes and enhancers on the other side of the

boundary (i.e. *Asb18* and *Six2* TADs). However, the *Asb18* and *Six2* TADs do not contain active enhancers (based on the presence of H3K27ac peaks) in the investigated cell types (ESC and NPC, respectively; Fig. 3A and Fig. 5A), thus precluding us from investigating enhancer-gene loops within those TADs. Although this could be interesting, we feel that is currently outside the main scope of the manuscript.

b. Are the CTCF binding sites near the SEs included in the inversions (Fig. 3B, Fig. 5B)? See my comment about scales and annotations of the figures.

We apologize for not making this information more explicit in the previous version of the indicated figures. The 71Kb inversion in the *Gbx2/Asb18* locus and the 156Kb inversion in the *Six3/Six2* locus do not span the CTCF sites near their corresponding SE, but rather place *Gbx2* and *Six3* next to those CTCF sites. In contrast, the 226Kb inversion in the *Six3/Six2* locus spans the CTCF site, which is moved together with the nearby SE closer to the TAD boundary/CTCF array. Following the reviewer's advice, the CTCF sites near the SEs and their orientation are now depicted in Fig. 3B and Fig. 5B (and all other similar Figure panels).

c. The blocking and facilitator function can be separated by removing the CTCF binding sites at the SEs, followed by gene expression and CapC experiments. Although I'm conscious of the additional work this requires, adding this result (at least for one domain) will considerably increase the impact of the study.

As mentioned above, we previously deleted the CTCF site located close to the *Six3* SE and showed that it is not required for proper *Six3* induction in NPC (see Extended Data Fig 9 in Pachano, T. et al., 2021). Similarly, the 226 Kb inversion, which spans the *Six3* SE and the nearby CTCF site and places *Six3* away from both the SE-CTCF site and the CTCF array/TAD boundary does not affect *Six3* induction in NPC (Fig. 5F). These results suggest that, at least in NPC, the CTCF sites do not play a major role in facilitating the communication between *Six3* and its SE. Instead, the CTCF array displays a clear blocking/insulation function, preventing the physical communication between *Six2* and the SE. In addition, we have now deleted the CTCF site near the *Gbx2* SE (Fig. S11A-C), which, as shown in Fig. S11D, lead to a small (<2-fold) and not significant decrease in *Gbx2* expression (these results are described in page 11 of the "Results" section). Given this minor contribution of the SE CTCF sites to the expression of either *Gbx2* or *Six3*, we have decided not to perform Capture-C experiments to further investigate the potential regulatory function of those CTCF sites.

d. Could the authors confirm the evolutionary conservation of the CTCF arrays at both the *Gbx2-Asb18* and *Six3-Six2* domains? Addition of a figure panel with CTCF binding in different vertebrate species would be insightful. CTCF binding has been published for many vertebrate species, at least up to zebrafish (e.g. Kaaij et al, 2018 Cell Rep).

Following the reviewer's advice, we have combined the ChIP-seq data from mESC and hESC used in our manuscript with ChIP-seq data previously generated in HH18 chicken embryos (GSM5835469) and 48 hpf zebrafish embryos (GSM5344494) in order to evaluate the

evolutionary conservation of the CTCF arrays found at the *Gbx2-Asb18* and *Six3-Six2* domains. As shown in Fig. S9, for the *Six3-Six2* domain, the CTCF array appears to be conserved across vertebrates, albeit the number of CTCF sites found between the two genes seem to differ among the evaluated species and/or cell types. For the *Gbx2-Asb18* domain, the CTCF array appears to be conserved among amniotes, while in zebrafish the *gbx2* and *asb18* genes are located in different chromosomes.

2. I consider the promoter competition aspect convincing, but the mechanism remains unclear. Whereas the competition for rate-limiting factors may be possible, it seems unlikely to me that a single promoter can exert such an effect. The authors state “... as this does not require the placement of developmental genes in between enhancers and non-target genes (Fig. 3 and 5) 36,37”, but these cited studies are from *Drosophila*, where loop extrusion and blocking by insulators appears not so important for enhancer promoter looping. Whereas the *Six3* deletion alone has a minor impact on *Six2*-SE contacts, the combined deletion of *Six3* and the CTCF array quite strongly increases the contacts (and *Six2* activity). Recent work, both cited and non-cited, has confirmed that TSS can create strong boundaries, particularly when enriched for paused PolII (Zhang D et al, 2020 Nat Genet; Barshad et al, 2023 Nat Genet; Zhang S et al, Nat Genet 2023).

We agree with the reviewer in that in the current manuscript we do not provide evidences regarding the molecular basis of promoter competition. As mentioned in the Discussion, one possibility supported mostly by theoretical studies (Deng, H. & Lim, B., 2022; Sabi, R. & Tuller, T., 2019; Waymack, R. et al., 2021) is that promoters might compete locally from some rate limiting factor(s) necessary for gene expression, but whether this type of mechanism can actually occur requires extensive additional work that goes beyond the current scope of our study.

Regarding previous studies in *Drosophila*, in which promoter competition and enhancer blocking have been evaluated, it is worth mentioning that some recent studies have shown, using Micro-C, that insulators and tethering elements are indeed important to either block or facilitate enhancer-gene contacts in *Drosophila* (e.g. Batut, P. J. et al., 2022). However, as the reviewer states, it is unclear whether loop extrusion is involved in such contacts.

Regarding the effects of the combined deletion of *Six3* and the CTCF array, we would like to point out that the contact frequency between *Six2* and the SE is only slightly higher than the contact frequency observed in cells in which the CTCF array was deleted alone (Fig 6. D-E; Fig. S21C-D). This together with the minor impact of the *Six3* deletion on *Six2*-SE contacts suggest that *Six3* does not act as a strong physical barrier. In agreement with this, the 156Kb inversion, either alone or combined with the CTCF array deletion, does not significantly affect *Six2* expression despite the fact that in those cell lines the *Six3* is no longer placed in between the SE and *Six2*. Overall, we think that, when considered together, our results support a model whereby *Six3* preferentially contributes to regulatory insulation through promoter competition rather than enhancer blocking, while the CTCF array provides physical insulation and blocks undesired *Six2*-SE contacts. However, and as stated in the “Discussion” section, we can not completely dismiss the possibility that the investigated developmental genes contribute to insulation through enhancer blocking (i.e. as physical barriers) (page 22: “*In this regard, it is possible that the developmental genes investigated in this study (i.e. Gbx2 and Six3) act as weak enhancer blockers whose effects on physical insulation can not be easily detected due to the current resolution and sensitivity of Capture-C and other 3C methods, but that, nevertheless, can still result in transcriptional*”).

changes”). In addition, it is also possible that genes with very strong expression levels and, thus, high accumulation of RNA Pol2 complexes (e.g. alpha and beta globin genes in blood cells) can display stronger barrier effects (e.g. Harrold, C. L. et al., 2020; Topfer, S. K. et al., 2022). than most developmental genes that, even when active, are expressed at comparably lower levels. This possibility has also been included in the “Discussion” section (page 22): “Furthermore, in the context of boundaries close to strongly expressed genes (e.g. globin genes), the contribution of active transcription and RNA Pol2 complexes to physical insulation might be larger in comparison to most developmental genes that, even when active, are expressed at comparably moderate levels.”

Following the reviewer’s suggestion, we have incorporated additional references supporting the role of TSS and RNA Pol2 as boundaries (Zhang, Y. et al., 2019; Zhang, S. et al., 2021; Zhang, S. et al., 2023; Banigan, E. J. et al., 2023; Bozhilov, Y. K. et al., 2021; Barshad, G. et al., 2023; Zhang, D. et al., 2020). However, in order to provide a more balanced view of the literature and following the suggestions of reviewer#2, we have also incorporated additional references indicating that the contribution of Pol2 to 3D chromatin architecture and physical insulation is rather minor (Hsieh, T.-H. S. et al., 2020; Zhang, H. et al., 2021; Jiang, Y. et al., 2020).

Question:

Could the authors analyze GRO-seq/PRO-seq/... data to determine if the Gbx2 and Six3 genes have a strong pausing index? Would it be worth it to include this analysis for the genome-wide characterization of paused genes at boundaries (Fig. 1)?

Considering previous reports in mammalian cells mentioned by the reviewer (Zhang, D. et al., 2020; Barshad, G. et al., 2023; Zhang, S. et al., 2023) as well as previous work in *Drosophila* (Chopra, V. S. et al., 2009), according to which paused RNA Pol2 at gene promoters can contribute to boundary formation, we decided to follow the reviewer’s suggestion and investigate the pausing index (PI) for the gene categories defined in Fig. 1 (i.e. All, Housekeeping and Developmental). Briefly, we used PRO-seq data from mESC (Vlaming, H. et al., 2022) and GRO-seq from hiPSC (Viiri, L. E. et al., 2019) to calculate the RNA Pol2 pausing index (PI) for Housekeeping, Developmental and All genes depending on their transcriptional status and proximity to TAD boundaries (i.e. bin1 genes). As depicted in Fig. S6, for each gene category, the genes located close to TAD boundaries (i.e. bin1 genes) showed similar PI to those observed for all genes within the same category. Furthermore, developmental genes showed lower PI than either housekeeping or all genes regardless of their transcriptional status or proximity to TAD boundaries. These observations are in agreement with previous reports showing that, in contrast to what has been described in *Drosophila* (Gaertner, B. & Zeitlinger, J., 2014), in mammalian cells developmental genes show lower PI than genes with more housekeeping functions (e.g. cell cycle, signal transduction) (Williams, L. H. et al., 2015). These analyses are now described in page 9 of the “Results” section.

Minor issues:

- Introduction (page 3 in Word document): “Nevertheless, the binding of architectural proteins can only explain the formation of a fraction of TAD boundaries, suggesting that alternative mechanisms should exist 8,10,11”. This may be true in *Drosophila* cells, but in mammalian cells

over 80% of TAD boundaries are lost upon CTCF depletion (Nora et al, 2017 Cell), with the remainder possibly explained by PolII-mediated boundaries (see also above).

The reviewer is right and our previous sentence could be misleading, as it could imply a rather moderate contribution of architectural proteins to TAD boundary formation. Therefore, we have rephrased this sentence in the introduction and also included the previously mentioned references regarding the role of RNA Pol2 in boundary formation in the same paragraph. Moreover, we have also incorporated a small sentence regarding the comparable minor role of architectural proteins for the organization of chromatin compartments based on the suggestions from Reviewer#2 (page 4): *“Furthermore, although architectural proteins play a preponderant role in TAD boundary formation, with, for example, over 80% of mammalian TAD boundaries being CTCF dependent, they seem to be dispensable for or even counteract the organization of chromatin compartments^{11,15,16}. Interestingly, TAD boundaries that are not dependent on CTCF are often located proximal to active transcription start sites (TSS) and housekeeping genes^{13,19}”*.

- Introduction (page 4): citations 8,20-24: the authors should include Barshad et al, 2023 Nat Genet. Zhang S et al, 2021 Sci Adv could be replaced by Zhang S et al, 2023 Nat Genet.

We have added the suggested references to highlight the role of RNA Pol2 in boundary formation. However, and as already mentioned, following reviewer#2 suggestions, we have also incorporated additional references indicating that the contribution of RNA Pol2 to 3D chromatin architecture are rather moderate (page 4): *“Recent reports indicate that RNA Pol2 can participate in the formation of contact domains and physical boundaries²⁰⁻²⁶, but whether this can globally contribute to the insulation of regulatory domains and enhancer-blocking remains controversial²⁷⁻²⁹”*.

- Introduction (page 4): citations 40-41: the authors should add Allayar et al, 2018 Nat Genet.

The suggested reference has been added.

- Introduction (page 4): “Promoter competition occurs when the preferred gene gets specifically activated regardless of its relative position with respect to the shared enhancer/s and neighbouring gene/s³⁶”. Leading insights into promoter competition, in mammalian cells, come from imprinted domains and particularly the Igf2-H19 domain. The authors could cite Barlow, 1997 Embo J.

The suggested reference has been added.

- Figure 1: I am surprised by the difference for bin 0 and 1 between panels C and F. Could this be due to how the boundaries were called? The manuscript provides very limited information about this. Depending on the strategy (directionality index, insulation score, arrowhead algorithm), the span of inter-TAD domains can considerably differ. It would be useful if the strategy could be harmonized.

We would like to thank the reviewer for this important comment. Upon closer inspection, and as the reviewer suspected, the TAD maps used in Fig. 1F were the only ones in which boundaries were defined using insulation scores according to Bonev et al., 2017, while all other boundaries were called using the directionality index approach initially established by Dixon et al., 2012 and systematically applied in the 3D Genome Browser database (Wang et al., 2018). Therefore, in order to harmonize the strategy, the boundaries of the TAD maps previously considered in Fig. 1F have now been defined using the directionality index approach. As a result, the discrepancy for the mESC TADs previously observed between Fig. 1C (Dixon et al.) and Fig. 1F (Bonev et al.), especially for Housekeeping genes, are considerably more subtle. Accordingly, panels C, D and E in Fig.1 have been modified as well as the accompanying text in the results section to better reflect the distribution of genes within TADs. Moreover, while revising the data presented in Fig. 1D-E, we noticed that three of the mouse TAD maps were duplicated, so the total number of analyzed mouse TAD maps is 11 rather than 14. Finally, in order to provide more information to the reader, additional text has been added to the “Methods” section (pages 30) explaining how TAD boundaries were called: *“To ensure consistency in TAD calling, the genomic coordinates of all TADs and their boundaries were retrieved from the 3D Genome Browser database¹¹², in which TAD boundaries were consistently called using the directionality index approach initially established by Dixon et al.¹³.”*

- Results (page 12) and Fig. 3B: clonal lines (i) and (ii) are inversed in the figure panel as compared to the text.

The text has been modified.

- Figure 3A, 3B and all similar panels: the interpretation of these panels is complicated by a lack of genomic coordinates and consistent scaling. Fig. 3A: what is the size of this region? Fig. 3B: these panels will be improved if they are drawn to the correct scale. I particularly have problems understanding how the configurations in Fig. 5B and 5E compare. The interpretation could be further facilitated if the span of panel 3B is added to panel 3A (and similar for other panels).

Regarding Fig.3A (and Fig.5A), we would like to bring to the reviewer’s attention that a size scale was included just beneath the Hi-C data. Moreover, and following the reviewer’s advice, we have added the genomic coordinates to Fig.3A (and Fig.5A), which further facilitates calculating the size of the regions shown in Fig. A (889 Kb) and Fig.5A (1,37 Mb).

On the other hand, in Fig. 3B and other similar panels (4A, 5B, 5E, 5G, 6A) we have made changes to improve the scaling. Moreover, in Fig. 3A and Fig. 5A we have also indicated the span of the corresponding deletions and inversions.

- Figure 3C and other RT-qPCR panels: I’m surprised to see that the WT data, which should be normalized to itself, is showing variation between replicates. How was this normalization performed? In a conventional $\Delta\Delta\text{CT}$ analysis, the normalization is performed for each PCR individually, thereby setting the WT value consistently to 1 for each experiment (or 0 after log transformation).

For all RT-qPCR panels the normalization with respect to WT data was performed using multiple biological replicates as indicated in the corresponding figure legends. Briefly, for the normalizations we calculated the mean value for the biological replicates of the WT cells and normalize the expression of each replicate (including the WT replicates and the replicates of the transgenic cell lines) against that average value. As result, the average fold-change for the WT cells is equal to 1 (0 in log2), but each replicate can show some variation around that mean value. This way of normalizing and presenting the WT data is frequently used in the scientific literature (see some examples below) and we think it better illustrates the variation in the expression of the investigated genes.

Funato et al., Am J Hu Genet, 2024

Li et al., Nature, 2023

Drizak et al., Nature Communications, 2023

- Figure 3C and other RT-qPCR panels: why did the authors include the two-fold cut-off for significance? This seems arbitrary.

The reason to include the two-fold cut-off was to use a measurement of effect size in addition to a measurement of statistical significance (i.e. $p \leq 0.05$ in two-sided unpaired t-tests), which we think helps in not overinterpreting p-values when assessing the significance of expression differences. With this said, the 2-fold cut-off, as well as the $p \leq 0.05$, are indeed arbitrary.

- Figure 6: Panel 6B is a repeat of panel 5C and could be removed. Fig. 5H and 6C are partially redundant with Fig. 5C and 5D, and may thus be reorganized as well.

It is true that some of the data presented in panel 6B is already showed in panel 5C (as well as for the other panels mentioned by the reviewer) and this is explicitly mentioned in the corresponding Figure legends. However, the redundancy is only partial (WT and $\Delta 6XCTCF$ cell lines), as each panel also includes unique data (156Kb Inv and $\Delta 6XCTCF:156Kb\ inv$ for Fig. 5C and $Six3^{-/-}$ and $\Delta 6XCTCF:Six3^{-/-}$ for Fig. 6B). We think that including the WT and $\Delta 6XCTCF$ in both Fig. 5C and Fig. 6B helps the reader to better appreciate the differences with respect to the other cell lines included in each panel, while keeping the order in which the different cell lines are presented in the main text. Therefore, we have decided to keep those panels as presented in the original manuscript version.

Reviewer #2 (Remarks to the Author):

Ealo et al. provide evidence that the robust insulation of regulatory domains occurs through the synergistic activities of CTCF insulation and promoter competition. Using genome-wide analysis of publicly available Hi-C and CTCF ChIP-seq data from mouse and human embryonic stem cells (ESCs), they found that developmental genes tend to exist in gene poor TADs and that they are generally found near CTCF clusters at TAD boundaries.

Next, the authors focused on two specific loci: 1) the developmental gene *Gbx2* and its upstream enhancer, separated by a CTCF cluster from the tissue-specific *Abs18* gene in the neighboring TAD; and 2) the *Six3* and *Six2* genes which are separated by a strong and conserved TAD boundary and displaying largely non-overlapping expression patterns during embryogenesis. Through the generation of multiple mESC homozygous cell lines they investigate the functional relevance of the sequential positioning of developmental genes, CTCF clusters and TAD boundaries. Their main conclusions are that the CTCF cluster and the promoter of the developmental gene itself act together to insulate the gene in the neighboring TAD. Furthermore, they suggest that this insulation by the promoter is due to competition for the enhancer, rather than physical insulation. This conclusion is supported by the finding that deletion of the developmental gene promoter had only minor effects on the contact frequency of the neighboring gene and the enhancer in the developmental TAD, but robust increases in gene expression. Overall, this is a very well executed study that addresses mechanisms of chromatin boundary formation. One of the interesting results is that the experiments make the distinction between enhancer/promoter competition vs boundary activity of a strong promoter. The molecular dissection of two boundaries is very well done, thoroughly controlled, and described in a clear and logical manner. An issue for me is that at the very end a new concept is introduced, i.e. that of silencer action at a distance, without providing any experimental evidence, and with no attempt to integrate this separate concept from the beginning.

We would like to sincerely thank the reviewer for the overall positive evaluation of our work and for the very thorough and constructive revision, which has allowed us to improve our manuscript.

Regarding the silencer action at a distance, we would like to point out that this concept is already introduced in page 16 of the “Results” section where we try to address why the deletion of the 6XCTCF cluster at the *Six3/Six2* locus impairs *Six3* induction in NPC. Since the potential role of the CTCF sites as facilitators of enhancer-gene communication is not supported by the inversions generated in this locus, we hypothesized that other mechanisms could contribute to the reduced induction of *Six3* upon deletion of the 6XCTCF cluster. We initially considered two possibilities: (i) the CTCF cluster could protect *Six3* from promoter competition by *Six2*; (ii) the CTCF cluster could protect *Six3* from the repressive effects of putative silencers located within the *Six2* TAD. Although we originally provided a reference (Batut et al., Science, 2022) indicating that TAD boundaries might protect genes from spurious interactions with silencers, the reviewer is correct in that we did not properly describe these previous evidences to better motivate our hypothesis. Therefore, in the revised manuscript, we have now added the following text in page 16: “Recent work in *Drosophila* indicates that insulators protect genes from spurious interactions with not only enhancers but also silencers located within neighbouring TADs⁶⁵. Therefore, the CTCF cluster might protect *Six3* from the repressive effects of putative silencer elements located within the *Six2* TAD and/or from the promoter competition activity of *Six2*”.

In any case, given the lack of robust chromatin signatures to globally identify silencers, we decided to focus on the *Six2* promoter competition hypothesis by generating mESC lines with deletions spanning both the CTCF cluster and *Six2* (Fig. 5G). As described in the “Results” section (page 16; Fig. 5H), the *Six2* deletion rescued, albeit partly, the impaired induction of *Six3* in NPC. In agreement with our hypothesis, these results suggests that, rather than facilitating the communication between *Six3* and its cognate enhancers, the CTCF cluster protects *Six3* from promoter competition by *Six2*. However, since the *Six2* deletion led to a partial rather than total rescue of *Six3* expression levels, it is still possible that the CTCF cluster could also protect *Six3* from silencers found within the *Six2* TAD. Given the lack of predictive chromatin signature for silencers mentioned above, we have not attempted to experimentally address whether silencers within the *Six2* TAD contribute to the impaired induction of *Six3* in cells with the CTCF cluster deletion and, based on recent observations (Batut et al., Science, 2022), we have simply stated this as a possibility (page 16): “This suggests that, rather than facilitating the communication between *Six3* and its cognate enhancers, the CTCF cluster protects *Six3* from promoter competition by *Six2*. However, since the *Six2* deletion led to a partial rather than total rescue of *Six3* expression levels, the CTCF cluster might also protect *Six3* from putative silencers found within the *Six2* TAD⁶⁵.” Overall, we believe that, given the main focus of our manuscript on promoter competition together with the lack of chromatin signatures to identify putative silencers with the *Six2* TAD, experimentally testing whether the CTCF cluster protects *Six3* from silencers should be address in future studies.

Finally, the concept that CTCF clusters could protect genes from the spurious effects of silencers located in neighbouring TADs initially introduced in page 16 appears again at the end of the manuscript when assessing the effects of a smaller deletion that partially removes the CTCF cluster separating the *Six3* and *Six2* domains (page 19, Fig. 6F-G). In this case, we observed that the partial deletion of the CTCF cluster (4XCTCF deletion) did not significantly increase *Six2* expression, but strongly impaired *Six3* induction. Together with the results obtained for the 6XCTCF deletion and the inversions generated within the *Six3* locus we surmise that (page 19) “Together with the results obtained for the inversions and the 6XCTCF deletion described above (Fig. 5), this further suggests that, rather than facilitating *Six3* expression, the CTCF cluster

separating the Six3 and Six2 TADs might protect Six3 from the repressive effects of Six2 and, potentially, of putative silencers located within the Six2 TAD^{33,83}, although the latter possibility remains speculative and would require further experimental evidences”.

The list of comments is long but a lot of it can be addressed textually.

Major points

1) The generalizing computational binary division of housekeeping vs developmental genes and their differential enrichment at various distances from boundaries is not entirely convincing because the differences between the two groups are quite modest or absent depending on data sets. Also, what about the converse question: are genes near TAD boundaries more enriched for developmental genes?

It is true that the differences between housekeeping and developmental genes regarding their distribution within TADs are very subtle or even absent in most cell types, as the % of both housekeeping and developmental genes located in “bin 1” is quite similar (Fig 3B-E). However, in the original text we did not state that there were any major differences between these two gene categories regarding their distribution within TADs, but rather that both were enriched near TAD boundaries (i.e. enriched in “bin 1” in comparison to other bins). In addition, while housekeeping genes are often located within TAD boundaries (i.e. “bin 0”), developmental genes are depleted in “bin 0”. Moreover, we have performed additional analyses showing that “bin 1” housekeeping genes are located at shorter distances from TAD boundaries than “bin 1” developmental genes. We apologize if the previous text was unclear and we have tried to improve it in the revised version of the manuscript (pages 7-8):

“Overall, both housekeeping and developmental genes were preferentially enriched near TAD boundaries (i.e. bin 1), with housekeeping genes showing slightly higher percentage values for bin 1. In addition, housekeeping genes were often found not only close to but also within TAD boundaries (i.e. enriched in bin 1 and, depending on the cell type, also in bin 0), while developmental genes were preferentially located inside TADs and near their boundaries (i.e. enriched in bin 1 and depleted in bin 0) (Fig. 1C-E). Moreover, “bin 1” housekeeping genes were located closer to TAD boundaries than their developmental counterparts (Fig. S4), which, considering the resolution of Hi-C data, is also in agreement with the more frequent location of housekeeping genes within TAD boundaries (i.e. bin 0). The preferential location of developmental genes close to TAD boundaries was similarly observed in both mice and humans, suggesting that it might represent an evolutionary conserved feature (Fig. 1C-E).

Regarding the question of whether the genes near TAD boundaries are enriched for developmental genes, the answer is no. As mentioned above, the % of housekeeping and developmental genes located in “bin 1” is rather similar. However, as stated in the “Methods” section (page 30), since the total number of housekeeping genes (Mouse: n=3277; Human: n=2176) is higher than that of developmental genes (Mice: n=967; Humans: n=1045), then there are more housekeeping genes than developmental ones located near TAD boundaries (i.e. bin 1). In any case, the enrichment near TAD boundaries has been previously reported for housekeeping genes but not for

developmental ones, which together with the sequential organization of developmental genes and CTCF cluster near boundaries made us focus on a couple of representative developmental loci. We acknowledge that it could be also interesting to investigate whether housekeeping genes can contribute to regulatory insulation through enhancer blocking and/or promoter competition, but we feel that this is outside the current scope of our manuscript and, thus, should be addressed in future studies.

Finally, we would like to point out that, based on suggestions from Reviewer#1, Fig. 1 has been modified in order to use TAD maps in which boundaries are consistently called using the same method (i.e. directionality index approach initially established by Dixon et al., 2012). Therefore, panel 1F has been eliminated since the TAD maps used in that panel were the only ones in which boundaries were defined using a different method (i.e. insulation scores according to Bonev et al., 2017). The boundaries of the TAD maps previously considered in Fig. 1F have now been defined using the directionality index approach and incorporated into Fig. 1C-E.

2) A possible limitation of the premise of this paper is that it remains an open question whether the binning of genes within TADs still applies when considering that TADs vary between tissues. In other words, developmental genes might distribute differently if the TAD analysis was done in the tissue in which they are actually expressed.

We agree with the reviewer in that the analysis could be even more informative if the TAD analysis was done in tissues in which the developmental genes are expressed. However, since the starting number of developmental genes is already not very high (Mice: n=967; Humans: n=1045), additional filtering based on expression levels results in very low gene numbers (e.g. 22 active genes (>5 fpkm) in hESC) that preclude any robust downstream analysis. On the other hand, and to somehow minimize the possibility that our results are affected by the TAD maps generated in a particular cell type, in the analyses presented in Fig. 1D-E we have used TAD maps generated in 37 human cell types and 11 mouse cell types. Finally, as now mentioned in the “Introduction” (page 3), previous studies suggest that major developmental genes are often located within TADs whose boundaries display particularly strong evolutionary conservation (Harmston, N. et al., 2017). In this regard, the two developmental loci that we have genetically dissected in our manuscript display a highly conserved TAD boundary/CTCF cluster. As shown in the new Fig S9, for the *Six3-Six2* domain, the CTCF cluster appears to be conserved across vertebrates (i.e. mouse, human, chicken and zebrafish). For the *Gbx2-Asb18* domain, the CTCF cluster appears to be conserved among amniotes (i.e. mouse, human and chicken), while in zebrafish the *gbx2* and *asb18* genes are located in different chromosomes.

3) In Figs. 1 c and f, housekeeping and developmental genes seems to behave differently when it comes to boundary proximity (bins 0 and 1). This makes me wonder about the significance of this categorization. Moreover, the Y-axes do not start at zero giving the impression of a larger effect size.

As mentioned above and following the suggestions of Reviewer#1, Fig. 1C-F have been modified. Briefly, upon closer inspection and as the reviewer#1 suspected, the TAD maps used in Fig. 1F

were the only ones in which boundaries were defined using insulation scores according to Bonev et al., 2017, while all other boundaries were called using the directionality index approach initially established by Dixon et al., 2012 and systematically applied in the 3D Genome Browser database (Wang et al., 2018). Therefore, in order to harmonize the strategy, the boundaries of the TAD maps previously considered in Fig. 1F have now been defined using the directionality index approach. As a result, the discrepancy for the mESC TADs previously observed between Fig. 1C (Dixon et al.) and Fig. 1F (Bonev et al.), especially for Housekeeping genes, are considerably more subtle. Accordingly, panels C, D and E in Fig.1 have been modified as well as the accompanying text in the results section to better reflect the distribution of genes within TADs. Finally, to provide more information to the reader, additional text has been added to the “Methods” section (pages 30) explaining how TAD boundaries were called: *“To ensure consistency in TAD calling, the genomic coordinates of all TADs and their boundaries were retrieved from the 3D Genome Browser database¹¹², in which TAD boundaries were consistently called using the the directionality index approach initially established by Dixon et al.¹³.”*

On the other hand, and following the reviewer’s suggestion, Fig. 1C has been modified so that the same Y-axes starting at 0 are shown for the three included cell types/TAD maps.

4) The distinction between developmental genes and tissue-specific genes is too vague. Developmental genes can also be tissue-specific.

We apologize for not including a proper description of how “developmental genes” were defined in our study, which was only described in the “Methods” section (page 29). We have now added the following text in the first paragraph of the “Results” section (page 6): *“To evaluate whether these features are prevalent among “developmental” TADs, we used TAD maps previously generated in either mouse or human cells and defined developmental genes based on the presence of broad polycomb domains around their TSS (See Methods)^{47–49}. This definition does not include all developmental genes according to Gene Ontology (GO) terms, but rather selects a subset of major developmental genes whose promoters include large CpG island clusters and display strong enhancer responsiveness^{50–52}. Nevertheless, for the sake of simplicity, and unless stated otherwise, we will refer to this subset of genes with broad polycomb domains around their TSS as “Developmental genes”. We hope that these sentences help clarifying the distinction we made later in the manuscript between developmental and tissue-specific genes (Page 7: “However, from a regulatory standpoint, developmental genes (as defined here based on the presence of broad PcG domains around their TSS) and tissue-specific genes represent fundamentally different gene categories, as they typically differ in the type of promoter (i.e. CpG-rich for developmental genes and CpG-poor for tissue-specific genes) and long-range enhancer responsiveness^{52,56}”) and that is mostly related to differences in promoter architecture and enhancer responsiveness.*

5) Does promoter competition occur more frequently in genes near TAD boundaries? In other words, is this phenomenon of promoter competition more related to simply any genes near TAD boundaries, or is there something specific about developmental genes being involved in the promoter competition?

These are highly relevant questions, which, unfortunately, our data can not conclusively answer as this would require the genetic dissection of additional loci in which non-developmental genes are located close to TAD boundaries as well as loci in which developmental genes are located far from boundaries.

With that said, the experiments performed in cells in which CTCF clusters were deleted (Fig. 3-4 and Fig. 5-6) and in which the relevant genes (*Gbx2* and *Six3*) are no longer close to a TAD boundary suggest that genes can engage into promoter competition even when located far from boundaries. Furthermore, the deletion of the CTCF cluster in the *Gbx2/Asb18* locus results in *Asb18* upregulation without affecting *Gbx2* expression levels (Fig. 3-4). This suggests that promoter competition between *Gbx2* (developmental gene according to our definition) and *Asb18* (tissue-specific gene) is not reciprocal, with *Gbx2* outcompeting *Asb18*. In contrast, in the *Six3/Six2* locus, the deletion of the CTCF cluster leads to *Six2* induction and *Six3* repression (Fig. 5-6), which suggest that these two developmental genes (according to our definition) display strong and reciprocal promoter competition. Together with previous reports indicating that developmental genes display strong long-range enhancer responsiveness, these results could suggest that developmental genes might display stronger promoter competition capacity than other gene types. However, and as stated above, in the absence of additional data we feel that these ideas are too speculative and, thus, we have not discussed them in the manuscript.

6) Related to point 5: I don't understand this idea:” We hypothesize that, by being close to boundaries, developmental genes are more likely to be located at shorter linear distances from their cognate enhancers than the potential competing genes located at the other side of the boundaries.” Please clarify.

We apologize for not properly explaining this idea. For developmental genes located close to TAD boundaries, the majority of their cognate enhancers should be located at more central locations within the same TAD. Consequently, the linear distance between the boundary proximal developmental genes and their cognate enhancers will be often smaller than the distance between those same enhancers and non-target genes located at the other side of the boundary within the adjacent TAD. Considering that short linear distances facilitate the functional communication between genes and enhancers, we hypothesize that the shorter distances separating boundary proximal genes and their cognate enhancers might give those developmental genes a competitive advantage over non-target genes located on the other side of the boundary within the neighboring TAD. On the other hand, the positioning of developmental genes close to the boundaries should often place them in between their cognate enhancers and non-target genes located in the adjacent TAD, thus potentially enabling developmental genes to display enhancer blocking function, which, depending on the locus, could be stronger than for *Gbx2* or *Six3*.

In order to make these ideas more clear to the reader, we have modified the corresponding text in the “Discussion” section (page 21).

7) Is there any role for the orientation of CTCF sites? The authors show that developmental boundaries have more CTCF peaks and CTCF ChIP-seq aggregated signal, but the insulation scores and boundary strengths (i.e. physical insulation) are similar to “other boundaries”. Is this

because of any distinct features of the directions of these CTCF binding sites near “developmental” boundaries than “non-developmental” boundaries? I suggest labeling the orientation of the CTCF sites under the CTCF tracks (Fig. 3-6, S1, S2, S5, S6, S8, S10, S11, etc.).

Following the request of reviewers#2 and #3, we used the window of +/- 100 Kb around the TSS of “bin 1” genes (as described in Fig. 2C) to calculate the orientation of the CTCF sites relative to the TAD centers, distinguishing between CTCF peaks located towards either the TAD center (inner window) or the TAD boundary (outer window) (Fig. S8A). Overall, for “all” and “housekeeping” genes the CTCF sites in the inner window show a slight preference for being oriented towards the TAD centers, while the CTCF sites in the outer window show a slight preference for being oriented towards the nearby boundaries (Fig. S8B-C). These moderate biases in the orientation of the CTCF sites are even less pronounced around developmental genes (Fig. S8B-C). Together with the results presented in Fig. 1D-E and Fig. S4, these results further suggest that housekeeping genes are often embedded within TAD boundaries where they are flanked by CTCF sites with divergent orientations (Nanni, L. et al., 2020). These results are presented in Fig. S8 and are briefly described in page 10.

On the other hand, following the reviewer’s suggestion, we have labelled the orientation of the key CTCF sites found within the *Gbx2/Asb18* and *Six3/Six2* loci in all figures except those showing very large genomic regions (e.g. Fig. S13, S15 and S21).

8) Clonal variation: In Fig. 3C, 71Kb INV seems to decrease the expression of *Gbx2* in three clones, whereas the expression of *Gbx2* increased in two clones, and was unchanged in four clones. What is the basis for this variability? Which clone was used in Fig. 3E? What were the fold changes of the expression of *Gbx2* and *Asb18*? Would the conclusion of Fig. 3E be different in clones with different *Gbx2* expression level?

As indicated in the Fig.3 legend, the expression data for the 71Kb INV was generated using 10 biological replicates using two different clonal lines (five replicates for each). Therefore, the variability noticed by the reviewer is not clonal but seems to arise between biological replicates (i.e. RNA was extracted for each clonal lines on different days following consecutive passages) and, thus, should not have a major effect on the conclusions of Fig. 3E. Furthermore, despite the variability in *Gbx2* expression levels, the effects of the 71Kb INV on *Asb18* expression are quite robust, which is most relevant for the main conclusion of Fig. 3E (pages 12): “*In contrast, the 71 Kb inversion strongly increased the contact frequency between the SE and the 3XCTCF cluster (Fig. 3E, Fig. S13B), but had a minor impact on the Asb18-SE or Asb18-Gbx2 contacts (Fig. 3E-F, Fig. S13C-E). Furthermore, both the 3XCTCF deletion and the 71 Kb inversion reduced the contact frequency between the SE and Gbx2 (Fig. 3E, Fig. S13A), which, nevertheless, did not have major regulatory effects on Gbx2 expression (Fig. 3C). Overall, these Capture-C experiments suggest that the increased expression of Asb18 in cells with the 71 Kb inversion is unlikely to be caused by the loss of Gbx2 enhancer blocking activity.*”

9) All gene expression changes are shown as fold-change. To better understand the competition

model absolute expression levels would be helpful to know. For example, is *Six3* expressed highly enough to be able to compete?

We agree with the reviewers in that providing more absolute expression levels for the investigated genes could be helpful. Therefore, we have now included in the “Results” sections the expression values (in FPKMs) for *Gbx2* and *Asb18* in mESC as well as for *Six3* and *Six2* in NPC as previously measured by RNA-seq (Cruz-Molina, S. et al., 2017):

Gbx2 in mESC: 12.15 FPKM

Asb18 in mESC: 0.056 FPKM

Six3 in NPC: 10.33 FPKM

Six2 in NPC: 0.21 FPKM

10) It would be nice to know whether the *Six2/Six3* TAD boundary is as strong in NPGs.

The Hi-C data shown in Fig. 5A for the *Six3/Six2* locus was generated in NPCs (Bonev et al., 2017), while the Hi-C data shown for the same locus in Supplementary Fig. 1B was generated in mESC, as indicated in the corresponding figure legends. The Hi-C data shown in these two figure panels indicates that the strength of the *Six3/Six2* boundary is similar in both cell types. Furthermore, all the Capture-C experiments for the *Six3/Six2* locus (e.g. Fig. 6D) were performed in NPC.

11) Fig.5H: can the incomplete rescue of *Six3* expression by the *Six2* deletion be explained that the *Six3* enhancer acts on other genes in the *Six2* TAD? What is the level of *Srbd1*?

Although this could be an interesting possibility, *Six2* is the only gene located within its TAD, with the *Srbd1* TSS/promoter already being located within the next TAD (Fig. S1B, Fig. S2B). Therefore, it is unlikely that the *Six3* enhancer can regulate *Srbd1* expression upon deletion of the CTCF cluster. Furthermore, the *Srbd1* gene can be classified as a housekeeping gene with similar expression levels in mESC (11 FPKM) and NPC (16 FPKM) (Cruz-Molina, S. et al., 2017), which based on work from us and others (Pachano, T. et al., 2021; Kraft, K. et al., 2019) typically show poor enhancer responsiveness.

12) Line 659: Pls make this statement more precise: “*Six3* preferentially contributes to regulatory insulation through promoter competition, while structural mechanisms, such as enhancer blocking, seem to have a comparably smaller, albeit non-negligible, contribution.” To me it seems they have synergistic effects. Δ CTCF upregulates *Six2* ~8-fold but when combined with Δ *Six3* ~64-fold.

We apologize for not being clear with the statement mentioned by the reviewer, since the idea that we wanted to put forward was that, based on our data, *Six3* contribution to regulatory insulation preferentially entails promoter competition rather than enhancer blocking, which instead seems to

be mostly dependent on the CTCF cluster. In order to be more precise and better describe our ideas, we have made the following changes to the text (page 17-18): *“Altogether, these results suggest that, as observed for the Gbx2/Asb18 locus, Six3 preferentially contributes to regulatory insulation through promoter competition, while Six3-dependent enhancer blocking seems to have a comparably smaller, albeit non-negligible, contribution. Therefore, enhancer blocking (i.e. physical insulation) is mostly dependent on the CTCF cluster, which together with Six3-mediated promoter competition synergistically confer the TAD boundary with strong insulator capacity”*

Furthermore, we fully agree with the reviewers in that the effects of the CTCF cluster deletion and the Six3 deletion are synergistic, as stated in page 17: *“Remarkably, although the Six3 deletion alone mildly changed (albeit in a non-statistically significant manner) Six2 expression in NPC, Six2 expression was strongly and synergistically increased when the Six3 and CTCF cluster deletions were combined (Fig. 6C) (72.1 Fold-change in $\Delta 6xCTCF:Six3^{-/-}$ vs WT; 8.8 fold change in $\Delta 6xCTCF$ and 1.7 fold-change in $Six3^{-/-}$ vs WT, respectively), thus in agreement with the contribution of Six3 to regulatory insulation.”*

13) A small point: comparing Fig.6C and Fig.6G: The $\Delta Six3$ (green data points) should be identical but are not. Where these clones re-analyzed separately?

The data for the $\Delta Six3$ cells is actually the same in Fig. 6C and Fig. 6G, but due to the usage of different type of triangles in the two panels, the distribution of each data point within the two panels is somehow different. To minimize this problem and avoid confusions, we have now used the same type of triangles in both panels. Furthermore, in the Fig.6 legend we indicate that the data for the $\Delta Six3$ cells is the same in Fig. 6C and Fig. 6G.

14) Line 709: At the very end the authors propose an all new concept: spreading of repressive chromatin or silencers from Six3 towards Six2 in $\Delta 4xCTCF$ cells, but apparently not in $\Delta 6xCTCF$. Why? This is a major problem because it really confounds the story. The authors need to test this idea experimentally. What is the evidence that there is a silencer element in the 1kb Six3 region that was deleted? This is not even considered in the Discussion.

Based on this comment, we believe the reviewer has misunderstood our interpretation of the results shown for the $\Delta 4xCTCF$ and $\Delta 6xCTCF$ cells, which we will try to explain below. Firstly, the spreading of repressive chromatin and/or silencers occurs from the Six2 TAD towards the Six3 TAD and not the other way around and, importantly, this is consistently observed in both $\Delta 4xCTCF$ and $\Delta 6xCTCF$ cells (i.e. impaired induction of Six3 in $\Delta 6xCTCF$ and $\Delta 4xCTCF$ can be observed in Fig. 5C and Fig. 6F, respectively). Secondly, we did not generate any 1Kb deletion within the Six3 region, since the $\Delta 6xCTCF$ and $\Delta 4xCTCF$ deletions both span 36 Kb that remove either six or four CTCF sites within the Six3/Six2 boundary, respectively (Fig. 6A). In order to more clearly describe the results associated with the $\Delta 4xCTCF$ deletion (Fig. 6F-G), we have now slightly modified the corresponding text in the “Results” section (page 19): *“Interestingly, the partial deletion of the CTCF cluster alone did not significantly increase Six2 expression, but strongly impaired Six3 induction (Fig. 6F-G). Together with the results obtained for the inversions and the 6XCTCF deletion described above (Fig. 5), this further suggests that, rather than facilitating Six3 expression, the CTCF cluster separating the Six3 and Six2 TADs might protect*

Six3 from the repressive effects of Six2 and, potentially, of putative silencers located within the Six2 TAD^{33,83}, although the latter possibility remains speculative and would require further experimental evidences. Last but not least, the combination of the partial deletion of the CTCF cluster and the Six3 deletion led to a strong and synergistic induction of Six2 in NPC (Fig. 6G). This further supports that Six3 contributes to regulatory insulation through promoter competition and, more generally, that the robust insulation provided by “developmental” boundaries requires the synergistic contribution of CTCF clusters and transcriptionally active developmental genes.”

On the other hand, and as already mentioned at the very beginning of our response to the reviewer, the concept of CTCF clusters acting as barriers that prevent silencer action at a distance is already introduced in page 16 when we describe the results obtained for the 6XCTCF cluster deletion (Fig. 5). Briefly, since the potential role of the CTCF sites as facilitators of enhancer-gene communication is not supported by the inversions generated in the *Six3* locus, we hypothesized that other mechanisms could contribute to the reduced induction of *Six3* upon deletion of the 6XCTCF cluster, including protection of *Six3* from the repressive effects of putative silencers located within the *Six2* TAD. Although we originally provided a reference (Batut et al., 2022) indicating that TAD boundaries might protect genes from spurious interactions with silencers, the reviewer is correct in that we did not properly describe these previous evidences to better motivate our hypothesis. Therefore, in the revised manuscript, we have now added the following text in page 16: *“Recent work in Drosophila indicates that insulators protect genes from spurious interactions with not only enhancers but also silencers located within neighbouring TADs⁸³. Therefore, the CTCF cluster might protect Six3 from the repressive effects of putative silencer elements located within the Six2 TAD and/or from the promoter competition activity of Six2”*. In any case, given the lack of robust chromatin signatures to globally identify silencers, we decided to focus on whether *Six2* could directly contribute to the impaired induction of *Six3* upon deletion of the 6XCTCF cluster (Fig. 5G). As described in the results section (page 16; Fig. 5H), the *Six2* deletion rescued, albeit partly, the impaired induction of *Six3* in NPC. Since the *Six2* deletion led to a partial, rather than total, rescue of *Six3* expression levels, it is conceptually possible that the CTCF cluster could also protect *Six3* from silencers found within the *Six2* TAD. Given the lack of predictive chromatin signature for silencers mentioned above, we have not attempted to experimentally address whether silencers within the *Six2* TAD contribute to the impaired induction of *Six3* in cells with the CTCF cluster deletion and, based on recent observations (Batut et al., 2022), we have simply stated this as a possibility (page 16): *“This suggests that, rather than facilitating the communication between Six3 and its cognate enhancers, the CTCF cluster protects Six3 from promoter competition by Six2. However, since the Six2 deletion led to a partial rather than total rescue of Six3 expression levels, the CTCF cluster might also protect Six3 from putative silencers found within the Six2 TAD⁶⁵.”*

Overall, we believe that, given the main focus of our manuscript on promoter competition together with the lack of chromatin signatures to identify putative silencers within the *Six2* TAD, experimentally testing whether the CTCF cluster protects *Six3* from silencers should be addressed in future studies. In any case, since the results obtained with the Δ 4XCTCF deletion cells are not essential to the main conclusions of our study, if the reviewer still finds these results confounding, we would agree to remove them from the revised manuscript.

15) Related to the previous point: Line 85: “compartmental domains” and “topologically

associating domains” (TADs) can be overlapping but are not the same. This needs to be considered especially when invoking the spreading of repressive chromatin or long-range repression.

We have modified this sentence and focused on mammalian TADs, which is the main focus of our study. Moreover, in doing so we have also avoid mixing references from *Drosophila* and mammals: (page 3) *“These studies revealed that, in mammals, CTCF sites often coincide with the boundaries of large self-interacting genomic regions that were termed “topologically associating domains” (TADs) ¹³.”*

16) This also means that “developmental genes” need to be sub-categorized. developmental genes were defined as those containing broad polycomb domains. While the reason for this and references are provided in the methods, perhaps a short sentence explaining their reasoning (with references) in line 175-176 might help. Regardless, this definition may cause bias in the analyses since findings may reflect the roles of PcG domains in regulatory insulation near TAD boundaries rather than developmental gene promoter competition. The authors should consider comparing the insulation of genes near TAD boundaries: a) developmental genes with polycomb domains; b) developmental genes without polycomb domains; c) non-developmental/non-lineage specific genes (if any) with polycomb domains; d) non-developmental/non-lineage specific genes without polycomb domains.

We agree with the reviewer in that our definition of “developmental genes” should be better described already in the “Results” section. Therefore, we have added the following sentence in page 6: *“...and defined developmental genes based on the presence of broad polycomb domains around their TSS (See Methods) ⁶⁰⁻⁶². This definition does not include all developmental genes according to Gene Ontology (GO) terms, but rather selects a subset of major developmental genes whose promoters include large CGI clusters and display strong enhancer responsiveness ⁶³⁻⁶⁵. Nevertheless, for the sake of simplicity, and unless stated otherwise, we will refer to this subset of genes with broad polycomb domains around their TSS as “Developmental genes”.*

Moreover, in order to perform the analyses suggested by the reviewer, we have used a broader set of “Developmental genes” as defined by Gene Ontology (GO) annotations (GO term “developmental process”; GO:0032502). Then, all boundaries associated with a nearby “bin 1” gene were divided in the following four groups:

1. Dev+ PcG+ (n=145 for mouse and n=153 for humans): boundaries associated with “bin 1” genes that are considered “GO-Developmental genes” and whose promoters have broad PcG domains.
2. Dev+ PcG- (n=952 for mouse and n=967 for humans): boundaries associated with “bin 1” genes that are considered “GO-Developmental genes” but whose promoters do not have broad PcG domains.
3. Dev- PcG+ (n=68 for mouse and n=90 for humans): boundaries associated with “bin 1” that are not considered “GO-Developmental genes”, but whose promoters have broad PcG domains.

4. Dev- PcG- (n=1890 for mouse and n=1810 for humans): boundaries associated with “bin 1” genes that are not considered as “GO-developmental genes” and whose promoters do not have broad PcG domains.

Next, we have calculated insulation scores, insulation strength, number of CTCF sites and the intensity of CTCF peaks for the four groups of boundaries as previously performed in Fig. 2A-B. These new analyses are presented in Fig. S5. Overall, these analyses show that the four types of boundaries display rather similar insulation scores and boundary strengths, while the number of CTCF peaks and CTCF ChIP-seq aggregated signals are slightly higher for boundaries associated with PcG genes (Dev+ PcG+ and Dev- PcG+ groups). These results indicate that the presence of PcG domains does not have a major effect on the physical insulation properties of nearby TAD boundaries. These new results are now briefly described in page 8: “*Furthermore, similar analyses were performed by considering a broader set of developmental genes (i.e. genes included in the Gene Ontology (GO) term “developmental process”; GO:0032502), which were further divided in two groups depending on whether their promoter regions were covered or not by broad PcG domains (Fig. S5). These analyses showed that the insulation scores and boundary strength of the boundaries associated with developmental genes were similar regardless of whether their promoters were associated with broad PcG domains, suggesting that PcG domains do not have a major effect on the physical insulation properties of nearby TAD boundaries*”.

17) Can the author discuss/speculate how promoter competition works across a CTCF boundary?

We have added the following paragraph to the “Discussion” section in order to discuss how promoter competition might contribute to regulatory insulation across CTCF boundaries (page 21): “*Recent studies indicate that the physical insulation provided by CTCF boundaries is dynamic and partial, with strong enhancers, such as those typically controlling the expression of major developmental genes, being able to bypass those boundaries and activate genes across TADs³³⁻³⁶. At the single cell level, the permeability of CTCF boundaries might result in a small fraction of cells/alleles in which cohesin-mediated loops bypass those boundaries and enable spurious contacts between enhancers and non-target genes. In the context of developmental loci, such as those investigated in our work, this can lead to the formation of multiway contacts (hubs) in which enhancers can simultaneously interact with their target developmental genes as well as with non-target genes located in neighboring domains⁵¹. However, we speculate that the strong enhancer responsiveness of developmental genes^{63,64} together with shorter linear distances with respect to their cognate enhancers⁹³ might give developmental genes a competitive advantage over non-target genes and, thus, prevent the spurious contacts between enhancers and non-target genes from being productive*”.

Additional points

There are a number of citation- and subject definition-related issues that need to be addressed (points 1-6).

- 1) Line 69: Insulators can be divided into enhancer blocking insulators and boundary elements. I

would not necessarily equate the two, especially in light of the invocation of silencer spreading at the end.

This sentence has been modified (page 3): *“In contrast, insulators protect gene promoters from signals emanating from neighbouring regulatory domains by either blocking the communication with non-cognate enhancers (i.e. enhancer blocking insulators) or by acting as barriers against the spreading of repressive chromatin (i.e. boundary elements) ^{4,5}.”*

2) It is important to textually clarify at the outset that in some of the experiments the authors use the term “insulation” in the context of 4C contacts, not in terms of function.

We agree with the reviewer in that it is important to distinguish between “insulation” in terms of contacts and in terms of function. To do so, in the revised text we have consistently used the term “physical insulation” when referring to any changes in enhancer-gene contacts as measured by Capture-C. In contrast, we have used the term “regulatory insulation” when evaluating insulation in functional terms (i.e. changes in gene expression).

The term “physical insulation” in the context of enhancer-gene contacts is already introduced in the Abstract.

3) Line 94: “Nevertheless, the binding of architectural proteins can only explain the formation of a fraction of TAD boundaries”. Again, this depends on the definition. Loop domains are lost upon cohesin depletion, but compartments remain intact and are even strengthened.

This sentence has been modified to also describe the role of architectural proteins in the organization of chromatin compartments. Moreover, and attending to a request from Reviewer#1, we have also highlighted the major role of architectural proteins in the formation of TAD boundaries (page 4): *“Furthermore, although architectural proteins play a preponderant role in TAD boundary formation, with, for example, over 80% of mammalian TAD boundaries being CTCF dependent, they seem to be dispensable for or even counteract the organization of chromatin compartments ^{11,15,16}. Interestingly, TAD boundaries that are not dependent on CTCF are often located proximal to active transcription start sites (TSS) and housekeeping genes ^{13,19}.”*

4) Line 96 and subsequent: “many architectural protein independent boundaries are located proximal to active transcription start sites (TSS) and housekeeping genes”. I suggest not mixing citations from flies and mammals because some of the concepts don’t translate well between organisms.

Following the reviewer’s advice, in both the indicated sentence as well as in the following one, we have removed the citations from flies and focused on those from mammals, which are more relevant for our work.

5) Line 97: “Recent reports indicate that transcription and/or active promoters can participate in the formation of contact domains and physical boundaries”. Several studies also show that

transcription has minimal impact on architecture (e.g. PMID: PMC7331254 and PMID: PMC8397779). Ref 21 could also be explained by enhancer-promoter competition.

We have now included the references suggested by the reviewer as well as additional references suggested by reviewer#1 in order to give a more balanced view of the potential role of transcription on chromatin architecture (page 4): *“Recent reports indicate that RNA Pol2 can participate in the formation of contact domains and physical boundaries²⁰⁻²⁶, but whether this can globally contribute to the insulation of regulatory domains and enhancer-blocking remains controversial²⁷⁻²⁹.”*

6) Same for line 138: There are other studies that use acute pol2 degradation or inhibition with little or no effect on architecture. These should be considered for a balanced presentation of the literature.

Following the reviewer’s suggestion, the indicated sentence has been modified and the references suggested by the reviewer have been incorporated (page 5): *“Recent studies suggest that promoter-driven enhancer blocking might involve structural mechanisms, whereby protein complexes present at promoters (e.g. RNA Pol2) act as weak barriers against cohesin-mediated loop extrusion^{21-23,25}. However, other studies based on the depletion or inhibition of RNA Pol2 have reported small or no effects in 3D chromatin architecture and formation of TAD boundaries²⁷⁻²⁹.”*

7) Line 125: “It has been previously suggested that the competition between promoters for a shared enhancer might involve mutually exclusive enhancer-promoter contacts (“flip-flop” model). However, more recent observations support non-structural mechanisms whereby promoters and enhancers share transcriptional hubs/condensates”. I suggest not phrasing these as mutually exclusive scenarios. The cited papers simply suggest that both mechanisms can work but at different loci.

The indicated sentences have been rephrased (page 5): *“It has been previously suggested that the competition between promoters for a shared enhancer might involve mutually exclusive enhancer-promoter contacts (“flip-flop” model)^{44,55}. In addition, recent observations suggest that, at certain loci, promoter competition could also entail non-structural mechanisms whereby promoters and enhancers share transcriptional hubs/condensates^{51,52}, within which promoters might compete for rate-limiting factors (e.g. TFs, GTFs, RNA Pol2) required for gene transcription^{56,57}.”*

8) There is a typo in line 119, remove “no”.

The typo has been removed.

9) Line 167 – please define CGI clusters (write out in full before using the abbreviation).

CGI has been defined (“...characterized by the presence of large CpG island (CGI) clusters...”)

10) Figure 1A: even if non-significant (since this will be reflected in the size of the bubbles anyway), show the results of the enrichment analysis of each GO term in all three populations.

The enrichment analyses of each GEO term are now shown in all three population in both Fig.1A and Fig. S3.

11) In Fig. 3E, it is not clear if the Capture-C maps with 71Kb INV displayed on reference genome track (non-inverted), or displayed as actual genome track (71Kb inverted)? Why is the enhancer still strongly interacting with the upstream regions outside the inverted regions even when the viewpoint is distal to it after inversion? In addition, is the increased interaction between the enhancer and the 3xCTCF sites in 71Kb INV is because the enhancer is now closer to the 3xCTCF sites? If the 71Kb INV is displayed on reference genome track (non-inverted), it might be better to show it on the actual genome track (71Kb inverted) as well to avoid confusion.

We apologize for not making this clear in the corresponding figure legend. We have now indicated in Fig. 3 and Fig. S13 legends that all Capture-C tracks are shown according to the mm10 reference genome. Although we considered showing the 71Kb INV data also with its actual genome track, we think it might complicate the overall interpretation of the figure as it would require additional items to indicate the location of the genes and enhancers in each track and, most importantly, it would complicate the interpretation of the subtraction tracks. In any case, and following the suggestions from reviewer#1, we have tried to improve the annotation of Fig.3 in general by more clearly indicating the span of the 71 Kb inversion and the 3xCTCF deletion.

On the other hand, the interaction peak outside the inverted region corresponds with a CTCF peak located upstream of the enhancer. Upon inversion, the interaction signal between the enhancer and this CTCF peaks decreases in comparison to WT or 3XCTCF cells, but it is still clearly detectable. In addition, as suspected by the reviewer, the increased interaction between the enhancer and the 3XCTCF sites in the 71Kb INV cells could be largely attributed to the close linear proximity between the enhancer and the CTCF cluster in those cells.

12) Line 524 – PcG has not been defined before this point – write it out in full. PcG has been defined where it first appears in the revised text (page 6): “...by the presence of large CpG island (CGI) clusters and Polycomb-Group (PcG) protein domains at their promoters ...”)

Reviewer #3 (Remarks to the Author):

In this manuscript, Ealo et al, investigate the synergies between CTCF clusters and developmental genes in topological insulation and gene expression. They initially used published HiC data from various human and mouse cell lines to determine unique characteristics of TADs containing developmental genes compared to TADs with housekeeping or other genes, with a

focus on the relative position to TAD boundaries, the strength and number of CTCF sites around the genes of interest and the boundary insulation strength.

Then, they perform a number of elegant genetic perturbation experiments on two specific developmental loci to uncover distinct mechanisms of how genes function synergistically with CTCF clusters to ensure functional insulation from neighboring genes. Specifically, they provide evidence for two distinct mechanisms, one supporting enhancer-blocking and another showing promoter competition. Although, both mechanisms have been proposed and illustrated in previous studies, the value and quality of this study is still very high.

Overall, this is an elegant and well-controlled study, with many interesting findings which support previous hypothesized or illustrated mechanisms of gene regulation and functional insulation. Although the perturbation studies are quite robust, I found the first two figures not very relevant for the study and rather weak both regarding the depth of analysis and the significance of the conclusions. Specific comments:

We would like to thank the reviewer for the overall positive evaluation of our work and for the constructive suggestions.

Regarding the two first figures, based on the suggestions from the three reviewers, we have performed additional analyses that complement and improve what was originally presented in Fig. 1-2:

- Considering previous reports in mammalian cells mentioned by the reviewer (Zhang, D. et al., 2020; Barshad, G. et al., 2023; Zhang, S. et al., 2023) as well as previous work in *Drosophila* (Chopra, V. S. et al., 2009), according to which paused RNA Pol2 at gene promoters can contribute to boundary formation, we decided to follow the reviewer's suggestion and investigate the pausing index (PI) for the gene categories defined in Fig. 1 (i.e. All, Housekeeping and Developmental). Briefly, we used PRO-seq data from mESC (Vlaming, H. et al., 2022) and GRO-seq from hiPSC (Viiri, L. E. et al., 2019) to calculate the RNA Pol2 pausing index (PI) for Housekeeping, Developmental and All genes depending on their transcriptional status and proximity to TAD boundaries (i.e. bin1 genes). As depicted in Fig. S6, for each gene category, the genes located close to TAD boundaries (i.e. bin1 genes) showed similar PI to those observed for all genes within the same category. Furthermore, developmental genes showed lower PI than either housekeeping or all genes regardless of their transcriptional status or proximity to TAD boundaries. These observations are in agreement with previous reports showing that, in contrast to what has been described in *Drosophila* (Gaertner, B. & Zeitlinger, J., 2014), in mammalian cells developmental genes show lower PI than genes with more housekeeping functions (e.g. cell cycle, signal transduction) (Williams, L. H. et al., 2015). These analyses are now described in page 9 of the "Results" section.
- Upon closer inspection of the TAD maps used in Fig. 1C-F, we noticed that the TAD maps used in former Fig. 1F were the only ones in which boundaries were defined using insulation scores according to Bonev et al., 2017, while all other boundaries were called using the directionality index approach initially established by Dixon et al., 2012 and systematically applied in the 3D Genome Browser database (Wang et al., 2018). Therefore,

in order to harmonize the strategy used in Fig. 1, and as suggested by Reviewer#1, the boundaries of the TAD maps previously considered in Fig. 1F have now been defined using the directionality index approach. As a result, the discrepancy for the mESC TADs previously observed between Fig. 1C (Dixon et al.) and Fig. 1F (Bonev et al.), especially for Housekeeping genes, are considerably more subtle. Accordingly, panels C, D and E in Fig.1 have been modified as well as the accompanying text in the results section to better reflect the distribution of genes within TADs.

- As requested by Reviewer#2, we have extended the analyses performed in Fig. 2A-B to additional set of genes. Briefly, we have used a broader set of “Developmental genes” as defined by Gene Ontology (GO) annotations (GO term “developmental process”; GO:0032502). Then, all boundaries associated with a nearby bin 1 gene were divided in the following four groups:
 1. Dev+ PcG+ (n=145 for mouse and n=153 for humans): boundaries associated with “bin 1” genes that are considered “GO-Developmental genes” and whose promoters have broad PcG domains.
 2. Dev+ PcG- (n=952 for mouse and n=967 for humans): boundaries associated with “bin 1” genes that are considered “GO-Developmental genes” but whose promoters do not have broad PcG domains.
 3. Dev- PcG+ (n=68 for mouse and n=90 for humans): boundaries associated with “bin 1” that are not considered “GO-Developmental genes”, but whose promoters have broad PcG domains.
 4. Dev- PcG- (n=1890 for mouse and n=1810 for humans): boundaries associated with “bin 1” genes that are not considered as “GO-developmental genes” and whose promoters do not have broad PcG domains.

Next, we have calculated insulation scores, insulation strength, number of CTCF sites and the intensity of CTCF peaks for the four groups of boundaries described above as previously performed in Fig. 2A-B. These new analyses are presented in Fig. S5. Overall, these analyses show that the four types of boundaries display rather similar insulation scores and boundary strengths, while the number of CTCF peaks and CTCF ChIP-seq aggregated signals are slightly higher for boundaries associated with PcG genes (Dev+ PcG+ and Dev- PcG+ groups). These results indicate that the presence of PcG domains does not have a major effect on the physical insulation properties of nearby TAD boundaries.

- Following the suggestions from reviewer#2 and #3, we have calculated the orientation of the CTCF sites located close to “bin 1” genes with respect to the center of the TADs. The results of these analyses are shown in Fig. S8. Briefly, we used a window of +/- 100 Kb around the TSS of “bin 1” genes (as described in Fig. 2C) to calculate the orientation of the CTCF sites relative to the TAD centers, distinguishing between CTCF peaks located towards either the TAD center (inner window) or the TAD boundary (outer window) (Fig. S8A). Overall, for “all” and “housekeeping” genes the CTCF sites in the inner window were preferentially oriented towards the TAD centers, while the CTCF sites in the outer window were preferentially oriented towards the nearby boundaries (Fig. S8B-C). These moderate biases in the orientation of the CTCF sites were even less pronounced around developmental genes (Fig. S8B-C). Together with the results presented in Fig. 1D-E and Fig. S4, these results further suggest that housekeeping genes are often embedded within

TAD boundaries where they are flanked by CTCF sites with divergent orientations (Nanni, L. et al., 2020). These results are presented in Fig. S8 and are briefly described in page 10.

1. One of the author conclusions is that developmental genes have CTCF clusters in between them and the boundaries, while other genes are flanked on both directions. First, the sequential orientation of boundary \diamond CTCF cluster \diamond gene is fully unsurprising. However, the selective “flanking” of housekeeping genes by CTCF sites is puzzling. Does this mean that these genes are part of “nested” TADs, while developmental genes are not? What is the orientation of the CTCF sites towards the center of TAD relative to the boundary cluster? What are the relative insulation scores of the CTCF sites on either direction?

Firstly, we would like to point out that, as shown in Fig. 2D-E and Fig. S7, the distribution of CTCF peaks around “all” and “housekeeping” genes located close to TAD boundaries (i.e. “bin 1” genes) is rather similar. Therefore, the “flanking” of housekeeping genes by CTCF sites does not seem to be selective but rather a frequent feature of genes located close to TAD boundaries. In contrast, what seems to be more selective is the sequential organization of developmental genes and CTCF clusters near TAD boundaries.

Secondly, we do not think that the “flanking” of housekeeping genes by CTCF sites indicates that these genes are part of “nested” TADs. Instead, if housekeeping genes are located closer to or even within TAD boundaries in comparison to developmental genes, this could explain why housekeeping genes are more often flanked by CTCF sites. In agreement with this possibility, Fig. 1D-E show that housekeeping genes are more often located in “bin 0” (within TAD boundaries) than developmental genes. In addition, we have calculated the distance between “bin 1” genes and their nearby boundaries. These new analyses (Fig. S4) show that “bin1” housekeeping genes are located significantly closer to TAD boundaries than “bin 1” developmental genes.

Furthermore, as already mentioned above, we used a window of +/- 100 Kb around the TSS of “bin 1” genes (as described in Fig. 2C) to calculate the orientation of the CTCF sites relative to the TAD centers, distinguishing between CTCF peaks located towards either the TAD center (inner window) or the TAD boundary (outer window) (Fig. S8A). Overall, for “all” and “housekeeping” genes the CTCF sites in the inner window were preferentially oriented towards the TAD centers, while the CTCF sites in the outer window were preferentially oriented towards the nearby boundaries (Fig. S8B-C). These biases in the orientation of the CTCF sites were moderate, but interestingly they were even less pronounced around developmental genes (Fig. S8B-C). Together with the results presented in Fig. 1D-E and Fig. S4, these results further suggest that housekeeping genes are often embedded within TAD boundaries where they are flanked by CTCF sites with divergent orientations (Nanni, L. et al., 2020). These results are presented in Fig. S8 and are briefly described in page 10.

Finally, we did not calculate the insulation scores for individual CTCF sites depending on their orientation, as the limited spatial resolution of Hi-C data would not allow us to distinguish between CTCF sites that are often part of clusters in which each site is separated by small distances from other CTCF sites.

2. The authors also mention that genes that have been previously shown to bypass boundaries are actually flanked by CTCF sites. Not sure, how is this observation relevant to the story. It seems - based on CTCF peaks- that both developmental loci (gene and SE) that they focus on, are also flanked by CTCF.

We agree with the reviewer in that those statements (and the associated figures; former Fig. S5-6) are not essential for our story and could be confusing, so they have been removed from the revised manuscript.

3. A better justification of why they selected these particular example loci would be very helpful. Some discussion of their special features and how could be used to ultimately predict other loci in the genome with either enhancer blocking or promoter competition would be very useful.

We have added the following text when we first mention the *Gbx2* and *Six3/Six2* loci in the “Results” section (page 10): “*To address the potential functional relevance of this sequential organization, we genetically dissected a couple of representative loci, Gbx2 and Six3/Six2 (Fig. S1-2, Fig. S9). These two loci were selected because they display the following features: (i) they contain developmental genes (i.e. Gbx2, Six3 and Six2) with broad PcG domains and large CGI clusters around their promoter regions; (ii) the developmental genes are located within gene-poor TADs and close to a TAD boundary (i.e. “bin 1” genes); (iii) the developmental genes precede clusters of strong CTCF sites; (iv) the sequential organization of developmental genes and CTCF clusters is evolutionary conserved (Fig. S9).*”

4. As mentioned above, the genetic experiments are very elegant and interesting. Their results indeed support that *Gbx2* contributes to the functional insulation through enhancer blocking. Do they authors believe that the transcriptional orientation of the gene, transcribing away of the boundary and towards the SE contribute to phenotype, potentially by counteracting extrusion? Inversion of the gene -especially in the context of CTCF deletions- should directly examine their insulating ability regardless transcriptional orientation.

Firstly, when considered together, our results suggest that both *Gbx2* and *Six3* contribute to the functional insulation of their regulatory domains preferentially through promoter competition rather than enhancer blocking. More specifically, for *Gbx2*, the deletion of its promoter, especially when combined with the deletion of the 3XCTCF cluster, lead to a strong increase in *Asb18* expression without having any major impact on neither the contact frequency between the *Gbx2* SE and *Asb18* (Fig. 4, Fig S13) nor the distribution of cohesin within the *Asb18* TAD (Fig. S16). These results are difficult to reconcile with *Gbx2* contributing to functional insulation through enhancer blocking. On the other hand, although *Gbx2* is transcribed towards the SE, *Six3* is transcribed in the opposite direction (towards the boundary) and yet, both genes seem to contribute to functional/regulatory insulation preferentially through promoter competition. Therefore, it seems unlikely that transcriptional orientation could play a major role in functional insulation. In any case, to directly address this possibility we have generated cell lines in which we inverted the

Gbx2 gene alone (i.e. in WT cells; *Gbx2 INV*) or together with the 3XCTCF cluster deletion (i.e. in Δ 3XCTCF cells; *Gbx2 INV: Δ 3xCTCF*) (Fig. S12A-C). As can be seen in Fig. S12D, the inversion of the *Gbx2* gene did not have any major effect on the expression of *Asb18* in the context of the 3XCTCF deletion. These results are described in page 12 of the “Results” section. Please note that, despite our efforts and as indicated in the Fig. S12 legend, for the *Gbx2 INV* cells we were only able to obtain one clonal line, while two clonal lines were obtained for the *Gbx2 INV: Δ 3xCTCF* cells.

5. A better representation and higher resolution of the Capture data would be helpful, especially for the inversion experiments. Did they reconstruct the genome for reference? What is the new strong peak downstream of the SE peak represents in Figure 3E?

6. Overall, both for the *Gbx2* and the *Six2/3* experiments, more clear statements and quantitation of all specific interactions between SE-promoter, promoter-promoter, promoter-boundary, SE-boundary etc, for each locus and perturbation would be very useful to better appreciate the conformational changes that associate and further support the enhancer-blocking and promoter competition models.

We address points #5 and #6 together as they are both related to the visualization and analysis of the Capture-C data.

All the Capture-C data, including the data generated in cells with the inversions, is shown using the reference mm10 genome rather than a reconstructed “custom” genome. We apologize for not making this clear in the corresponding figure legends. We have now indicated in the legends of Fig. 3, Fig. 4, Fig. 6 and the associated supplementary figures that all Capture-C tracks are shown according to the mm10 reference genome. Although we considered showing the *Gbx2* 71Kb INV using a reconstructed “custom” genome track, we think it might complicate the overall interpretation of the figure as it would require additional items to indicate the location of the genes and enhancers in each track and, most importantly, it would complicate the interpretation of the subtraction tracks. In any case, and following the suggestions from reviewer#1, we have tried to improve the annotation of Fig.3 in general by more clearly indicating the span of the 71 Kb inversion and the 3xCTCF deletion.

The strong peak downstream of the *Gbx2* SE in Fig. 3E is located within the 3’UTR of the *Agap1* gene and, thus, its potential relevance for *Gbx2* or *Asb18* expression is unclear. Moreover, the peak is restricted to a single restriction site fragment and, as a result, in the subtraction tracks and following smoothing of the data for visualization purposes, the difference between the 71Kb INV and WT cells does not appear to be very pronounced.

Following the reviewer’s suggestion we have now performed additional quantifications of the Capture-C data, including: (i) interactions between SE *Gbx2* and *Gbx2* (Fig. S13A); (ii) interactions between SE *Gbx2* and the 3XCTCF cluster (Fig. S13B, Fig. S15A); (iii) interactions between *Asb18* and *Gbx2* (Fig. S13E); (iv) Interactions between *Asb18* and the 3XCTCF cluster (Fig. S13F, Fig. S15D); (v) interactions between SE *Six3* and *Six3* (Fig. S21A); (vi) interactions between SE *Six3* and the 6XCTCF cluster (Fig. S21B); (vii) interactions between *Six2* and *Six3* (Fig. S21E); (viii) interactions between *Six2* and 6XCTCF cluster (Fig. S21F). These quantifications are now briefly described in the results section. Please note that quantifications

with regions that have been deleted in some of the cell lines have not been considered in those cell lines (e.g. interaction between SE *Gbx2* and the 3XCTCF cluster in $\Delta 3xCTCF$ or $\Delta 3xCTCF:71Kb$ *INV* cells).

7. Many figure panels need better annotation. For example, in Figure 1D, it is unclear of the numbers shown on the right represent distinct TADs or cell types. Overall, the numbers of TADs and genes interrogated in each analysis should be clearly stated.

We apologize for not including this important information in the original manuscript. The overall number of TADs, boundaries and genes considered in the analyses presented in Fig. 1 and Fig. 2 (and associated supplementary figures; Fig. S4-8) are now stated directly in the figures.

REVIEWERS' COMMENTS

Reviewer #1 (Remarks to the Author):

I thank the authors for their efforts to respond to my questions and to the questions raised by the other reviewers as well. The added analyses have helped to solve a number of open questions, whereas the modified figures have made the results more easily accessible. Based on the comments by the authors, and also reviewer 2, I appreciate the nuanced discussion about the potential impact of PolII and active promoters in boundary formation.

I have three minor remarks, but otherwise I fully support its publication in Nature Communications.

Minor remarks:

1. Reflecting on the model that emerges from the study, would it be an idea to make the title more explicit (i.e. including physical insulation and promoter competition).
2. Line 301 and newly added fig. S8: I find this analysis and result very interesting. But considering that the differences are relatively minor, the analysis will be more convincing if the authors can add p-values for relevant pairwise differences.
3. In the same section, the preference for "outward" oriented motifs for "all" and "housekeeping" genes (i.e. away from the gene) has some overlap with the recent study by Ge et al. (Nature Communication 2023, article number 8101). The authors should include this study in their reflections. Could the authors expand on their analysis in Fig. S8, to see when clusters of peaks are present in the inner and outer windows, if specific patterns of orientations are enriched (for instance preferences for divergent, convergent or tandem orientations of pairs/triplets/etc)?

Reviewer #2 (Remarks to the Author):

The authors addressed my comments very thoroughly and thoughtfully, and the manuscript is much improved. Overall this is a really nice study. The remaining comments are relatively minor and relate to improving the clarity for the readers so they can better appreciate the findings.

1. At this point I would simply abstain from making major distinctions between developmental genes, housekeeping genes or "all genes" when it comes to their position within TADs. As shown in Fig.1C, the differences are negligible. Same really for Fig.2. This recommendation extends to the title of the paper. It could instead read: "Cooperative domain insulation by regulatory elements and clusters of CTCF sites". I would be careful when using the term "synergistic".
2. "...boundary-proximal genes in general and housekeeping genes in particular are frequently flanked by CTCF peaks". For the casual reader: Since genes and CTCF sites tend to accumulate at TAD boundaries, isn't this a self-fulfilling prophecy?
3. Thank you for providing expression data for Gbx2 and Asb18, with the former being expressed ~217 times higher. This means that the absolute level of Asb18 upregulation upon $\Delta 3XCTCF$ or the 71KbINV by ~5- to ~16-fold is still very small, suggesting that the Gbx2 enhancer may not be all that promiscuous in terms of activity (in spite of increased contacts in the $\Delta 3XCTCF/71KbINV$), and that both the Gbx2 enhancer-blocking activity and the ability to act as competitor are not that strong, consistent with the Capture-C data. Perhaps mention this more explicitly. Therefore, this statement "...suggest that developmental genes might display stronger promoter competition capacity than other gene types" may not be justified because competition would be expected to be a function of expression level.
4. Many of the results are based on very nice experiments but are complicated to interpret even when reading this a second time. Perhaps tables of the various mechanisms/concepts would help? E.g.: First, contributing factors related to gene activity: Gbx2 enhancer-Asb18 promoter proximity/weak boundary (based on the inversion), CTCF insulation (based on $\Delta 3XCTCF$), promoter competition (based on Gbx2 promoter deletion). Second, the same scenarios but related to Capture-C physical contacts which seem uncoupled from activation. Similar tables could be

designed for the Six2/3 locus perturbation, highlighting commonalities and differences between this and the Gbx2/Asb2 locus, including perhaps the protection from putative silencer elements, even though the latter is speculative.

5. Discussion: "... protect Six3 from the repressive effects of Six2". Since Six2 is expressed significantly more highly than, for example Asb18, its repressive effects on Six3 seem doubtful. If the CTCF deletion causes repression at a distance, wouldn't it then be more likely be due to regions outside of Six2. Perhaps adjust the Discussion accordingly.

Reviewer #3 (Remarks to the Author):

The authors did a great job addressing the reviewers' comments by including additional experiments, analyses, controls and textual clarifications. Importantly, they have improved significantly the discussion and presentation of the results, which in the original submission were unclear and confusing. Moreover, they have more thoroughly addressed (or at least discussed) alternative interpretations of their results in a fair manner.

I recommend publication of the study.

REVIEWERS' COMMENTS

Reviewer #1 (Remarks to the Author):

I thank the authors for their efforts to respond to my questions and to the questions raised by the other reviewers as well. The added analyses have helped to solve a number of open questions, whereas the modified figures have made the results more easily accessible. Based on the comments by the authors, and also reviewer 2, I appreciate the nuanced discussion about the potential impact of PolII and active promoters in boundary formation.

I have three minor remarks, but otherwise I fully support its publication in Nature Communications.

Minor remarks:

1. Reflecting on the model that emerges from the study, would it be an idea to make the title more explicit (i.e. including physical insulation and promoter competition).

Following the suggestions from Reviewers #1&2 we have modified the title as follows:
“Cooperative insulation of regulatory domains by CTCF-dependent physical insulation and promoter competition”.

2. Line 301 and newly added fig. S8: I find this analysis and result very interesting. But considering that the differences are relatively minor, the analysis will be more convincing if the authors can add p-values for relevant pairwise differences.

Following the reviewer’s advice, we have performed Chi-square tests in order to evaluate whether there were any significant differences in the distribution of the orientation of the CTCF sites between the Inner and Outer windows for each gene category (i.e. All, Developmental and Housekeeping). P-values and Cramér’s V values (an effect size estimator for Chi-squared tests) have now been added to Fig. S8. These Chi-square tests revealed significant, albeit weak, differences in the distribution of inward and outward CTCF sites between the Inner and Outer windows of both All and Housekeeping genes, while non-significant differences were observed for Developmental genes. Based on these additional tests, the text associated to Fig. S8 in the results section has been slightly modified (page 10): *“Next, using the same window of +/- 100 Kb around the TSS of bin 1 genes (i.e. inner and outer windows), we calculated the orientation of the CTCF sites relative to the TAD centers, distinguishing between CTCF sites oriented either towards (i.e. inward site) or away (i.e. outward site) from the TAD center (Fig. S8). A considerable fraction of CTCF peaks (at least 35%) was observed for both orientations regardless of the type of genes or windows analysed, supporting the potential relevance of not only inward but also outward CTCF sites for the proper establishment of intra-TAD chromatin interactions⁷³. The differences in the fraction of inward and outward CTCF sites were negligible when comparing the inner and outer windows associated to developmental genes (Fig. S8). However, although still minor, non-negligible differences were observed for All and Housekeeping genes (Fig. S8), with the CTCF sites in the inner windows preferentially showing an inward orientation*

(~60%) and the CTCF sites in the outer window preferentially showing an outward orientation (>50%) (Fig. S8). Together with the results presented in Fig. 1D-E and Fig. S4, these results further suggest that housekeeping genes are often embedded within TAD boundaries where they are flanked by CTCF sites with divergent orientations⁷⁴. In contrast, developmental genes tend to be located inside TADs but close to boundaries with large clusters of CTCF sites with complex motif orientations.”

3. In the same section, the preference for “outward” oriented motifs for “all” and “housekeeping” genes (i.e. away from the gene) has some overlap with the recent study by Ge et al. (Nature Communication 2023, article number 8101). The authors should include this study in their reflections. Could the authors expand on their analysis in Fig. S8, to see when clusters of peaks are present in the inner and outer windows, if specific patterns of orientations are enriched (for instance preferences for divergent, convergent or tandem orientations of pairs/triplets/etc)?

We have added the suggested reference (page 10; Ref#73): ““Next, using the same window of +/- 100 Kb around the TSS of bin 1 genes (i.e. inner and outer windows), we calculated the orientation of the CTCF sites relative to the TAD centers, distinguishing between CTCF sites oriented either towards (i.e. inward site) or away (i.e. outward site) from the TAD center (Fig. S8). A considerable fraction of CTCF peaks (at least 35%) was observed for both orientations regardless of the type of genes or windows analysed, supporting the potential relevance of not only inward but also outward CTCF sites for the proper establishment of intra-TAD chromatin interactions⁷³.”

On the other hand, since the Chi-squared tests showed that, albeit significant, the differences in the distribution of inward and outward CTCF sites between the Inner and Outer windows of both All and Housekeeping genes are rather minor, we consider that the additional analyses suggested by the reviewer, although potentially interesting, are unlikely to reveal any strong differences in the orientation of CTCF sites and could deviate the attention from our main findings regarding the sequential organization of developmental genes and CTCF clusters near TAD boundaries.

Reviewer #2 (Remarks to the Author):

The authors addressed my comments very thoroughly and thoughtfully, and the manuscript is much improved. Overall this is a really nice study. The remaining comments are relatively minor and relate to improving the clarity for the readers so they can better appreciate the findings.

1. At this point I would simply abstain from making major distinctions between developmental genes, housekeeping genes or “all genes” when it comes to their position within TADs. As shown in Fig.1C, the differences are negligible. Same really for Fig.2. This recommendation extends to the title of the paper. It could instead read: “Cooperative domain insulation by

regulatory elements and clusters of CTCF sites". I would be careful when using the term "synergistic".

Following the suggestions from Reviewers #1&2 we have modified the title as follows: *"Cooperative insulation of regulatory domains by CTCF-dependent physical insulation and promoter competition"*. Moreover, we have exchanged the term "synergistic" or "synergistically" by "cooperative" or "cooperatively" throughout the text.

2. "...boundary-proximal genes in general and housekeeping genes in particular are frequently flanked by CTCF peaks". For the casual reader: Since genes and CTCF sites tend to accumulate at TAD boundaries, isn't this a self-fulfilling prophecy?

The reviewer is right and the fact that genes near or at boundaries are flanked by CTCF sites is expected. However, the distribution of CTCF peaks with respect to developmental genes seem to be different, as developmental genes often precede CTCF sites rather than being flanked by them. Therefore, we believe it is important to state the differences between gene categories with respect to the distribution of genes and CTCF sites near TAD boundaries.

3. Thank you for providing expression data for Gbx2 and Asb18, with the former being expressed ~217 times higher. This means that the absolute level of Asb18 upregulation upon $\Delta 3XCTCF$ or the 71KbINV by ~5- to ~16-fold is still very small, suggesting that the Gbx2 enhancer may not be all that promiscuous in terms of activity (in spite of increased contacts in the $\Delta 3XCTCF/71KbINV$), and that both the Gbx2 enhancer-blocking activity and the ability to act as competitor are not that strong, consistent with the Capture-C data. Perhaps mention this more explicitly. Therefore, this statement "...suggest that developmental genes might display stronger promoter competition capacity than other gene types" may not be justified because competition would be expected to be a function of expression level.

Following the reviewer's suggestion, the following sentence has been added (page 13): *"Furthermore, since Asb18 is expressed at low levels in mESC (0.056 FPKM), the upregulation of Asb18 in absolute levels in $\Delta 3XCTCF$ (~5-fold) or 71Kb INV (~16-fold) cells is still very small, suggesting that the responsiveness between the Asb18 promoter and the Gbx2 enhancer might be limited 67."*

4. Many of the results are based on very nice experiments but are complicated to interpret even when reading this a second time. Perhaps tables of the various mechanisms/concepts would help? E.g.: First, contributing factors related to gene activity: Gbx2 enhancer-Asb18 promoter proximity/weak boundary (based on the inversion), CTCF insulation (based on $\Delta 3XCTCF$), promoter competition (based on Gbx2 promoter deletion). Second, the same scenarios but related to Capture-C physical contacts which seem uncoupled from activation. Similar tables could be designed for the Six2/3 locus perturbation, highlighting commonalities and differences between this and the Gbx2/Asb2 locus, including perhaps the protection from putative silencer elements, even though the latter is speculative.

Following the reviewer's advice, we have generated a couple of tables (Tables 1 and 2) to summarize the results obtained for the *Gbx2/Asb18* and *Six3/Six2* loci.

5. Discussion: "... protect Six3 from the repressive effects of Six2". Since Six2 is expressed significantly more highly than, for example *Asb18*, its repressive effects on Six3 seem doubtful. If the CTCF deletion causes repression at a distance, wouldn't it then be more likely be due to regions outside of Six2. Perhaps adjust the Discussion accordingly.

The sentence mentioned by the reviewer has been modified as follows to better reflect our overall observations (page 19): *"the CTCF cluster separating the Six3 and Six2 TADs might protect Six3 from promoter competition by Six2 and, potentially, from the repressive effects of putative silencers located within the Six2 TAD, although the latter possibility remains speculative and would require further experimental evidences."*

Reviewer #3 (Remarks to the Author):

The authors did a great job addressing the reviewers' comments by including additional experiments, analyses, controls and textual clarifications. Importantly, they have improved significantly the discussion and presentation of the results, which in the original submission were unclear and confusing. Moreover, they have more thoroughly addressed (or at least discussed) alternative interpretations of their results in a fair manner.

I recommend publication of the study.